# Ephemeral grounding on the Pine Island Ice Shelf, West Antarctica, from 2014 to 2023

Yite Chien[1,2,3], Chunxia Zhou[1,2,3*], Sainan Sun[4], Yiming Chen[1,2,3], Tao Wang[1,2,3], Baojun Zhang[1,2,3]

[1]Chinese Antarctic Center of Surveying and Mapping, Wuhan University, Wuhan, 430079, China
[2]Key Laboratory of Polar Environment Monitoring and Public Governance (Wuhan University), Ministry of Education, Wuhan, 430079 China
[3]School of Geodesy and Geomatics, Wuhan University, Wuhan, 430079 China
[4]Department of Geography and Environmental Sciences, Northumbria University, Newcastle upon Tyne, NE1 8ST, UK

*Correspondence to*: Chunxia Zhou (zhoucx@whu.edu.cn)

**Abstract.** The evolution of ephemeral grounding in ice shelf can affect buttressing, alter ice flow dynamics, and influence ice shelf stability. Long-term observations of ephemeral grounding sites are crucial for understanding how thickness, basal conditions, and tidal interactions evolve over time. Vertical displacement data derived from Sentinel-1A/B imagery reveals the history of ephemeral grounding events at PIIS from 2014 to 2023. Our results suggest that ephemeral grounding at an ice rumple is modulated by the interaction between tidal forcing, ice shelf thickness, and evolving sub-ice-shelf geometry. A prominent central keel, shaped by inherited bed topography, promotes repeated contact with a submarine ridge. Landsat-8 images reveal that the rifts that cause the 2020 calving event may have formed due to the ice shelf grounding at the study site. These findings provide new insights into the mechanisms driving ephemeral grounding behaviour and highlight its potential role in modulating ice shelf stability.

## 1 Introduction

Ice discharge from the Antarctic Ice Sheet is a major contributor to global sea-level rise (Shepherd et al., 2012; Bamber et al., 2018; Smith et al., 2020). This discharge is regulated in part by ice shelves, which exert a buttressing that resists upstream ice flow. However, in many regions, the buttressing capacity of ice shelves has been reduced by processes such as ice shelf thinning, calving events, grounding line retreat, unpinning from topographic highs, and the disintegration of shear margins (Fürst et al. 2016; Gudmundsson et al., 2019; Lhermitte et al., 2020; Miles and Bingham, 2024; Walker et al., 2024; Fricker et al., 2025).

A prominent example of these dynamics can be seen in the Amundsen Sea sector of West Antarctica, which accounts for over 31% of the continent's total ice loss. Within this sector, the Pine Island Glacier (PIG) basin alone contributed approximately 3.0 mm to global sea-level rise between 1979 and 2017 (Smith et al., 2020; Rignot et al., 2019). The PIG ice front has retreated approximately 26 km since 2015, with calving frequency increasing from intervals of about six years to every one to two years (Depoorter et al., 2013; Mouginot et al., 2014; Paolo et al., 2015; Arndt et al., 2018; Shepherd et al., 2018; Qi et al., 2021; Joughin et al., 2021). Following three major calving events in 2017, 2018, and 2020, the ice shelf experienced a >12% speedup relative to 2017, coinciding with a 19 km retreat of the ice front (Joughin et al., 2021).

The recent dynamic changes at PIG have been driven mostly by enhanced basal melting, caused by the intrusion of warm Circumpolar Deep Water (CDW) beneath the ice shelf (Jenkins et al., 2010; Jacobs et al., 2011; Pritchard et al., 2012; Hillenbrand et al., 2017; Smith et al., 2017; Davies et al., 2017; Shean et al., 2019). This oceanic forcing initially caused transient grounding of the central ice shelf on a submarine ridge from the 1940s through the 1970s, followed by complete ungrounding between 1973 and 1989 (Jenkins et al., 2010; Smith et al., 2017; Miles and Bingham, 2024). Continued ice shelf thinning subsequently drove an ~8.7 km retreat of the grounding line along the main trunk between 1992 and 2009, resulting in further ungrounding from an ice plain (Corr et al., 2001; Joughin et al., 2010; Dutrieux et al., 2014a; Rignot et al., 2014). Despite the grounding line retreat, the Pine Island Ice Shelf (PIIS) was observed to maintain intermittent contact with the bathymetric high when thick ice column being advected from the upstream deep trough (Joughin et al., 2016; Lowery et al., 2025). This region is referred to as ice rumple L (Figure 1) in the study by Rignot et al. (2014). This ephemeral grounding is now attributed to interactions between sub-ice keels and a submarine ridge (Graham et al., 2013; Joughin et al., 2016; Shean, 2016; Davies et al., 2017).

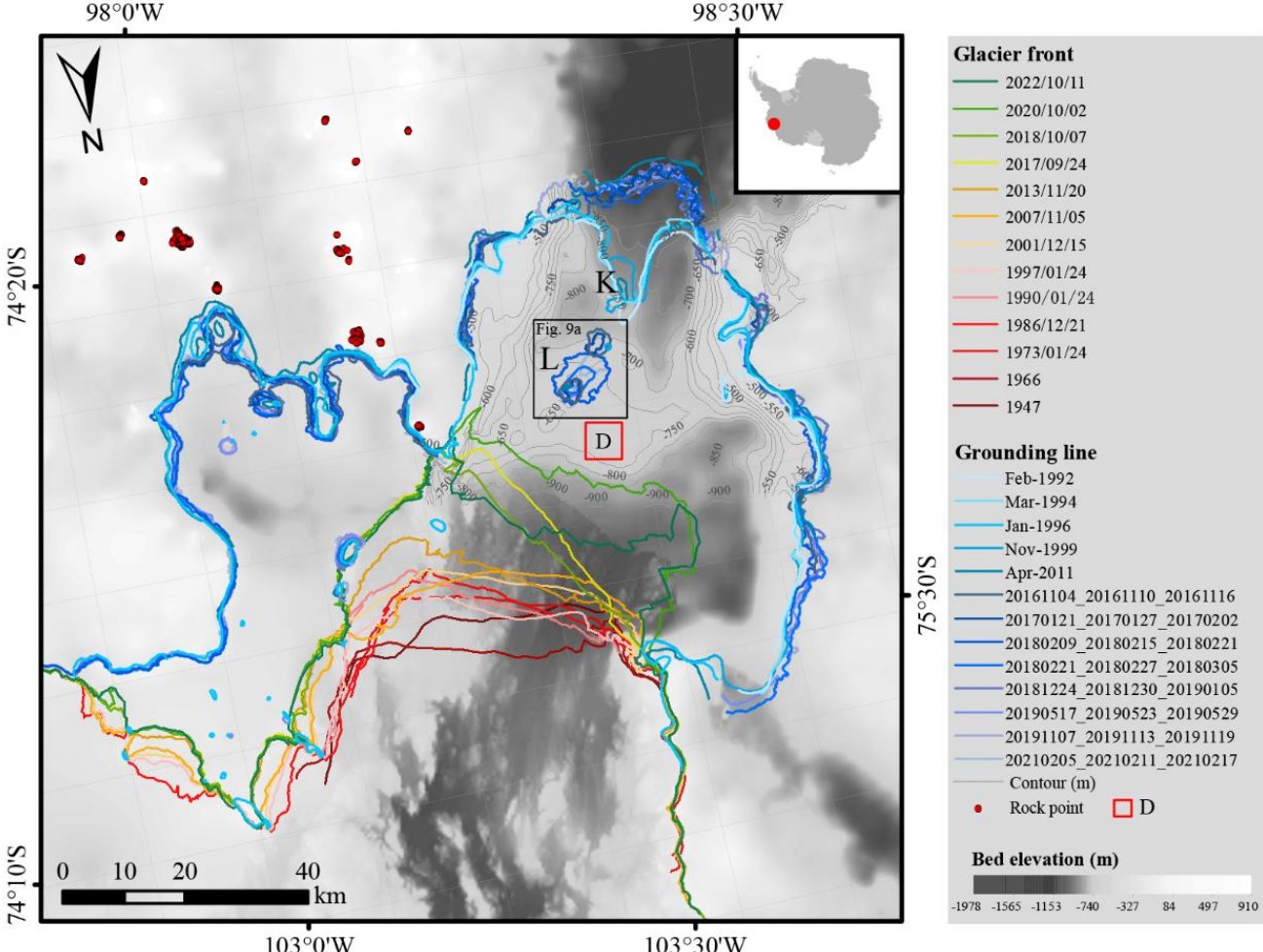

**Figure 1.** Location and geometry of the PIIS. Ice front positions, grounding line locations, and 458 non-glaciated ground control points (red points). Bed elevation (50 m contour interval, labelled between -750 m and -500 m) is from BedMachine v3 (Morlighem et al., 2020; Morlighem, 2022), showing the submarine ridge. Grounding lines are from MEaSUREs (Rignot et al., 2016) (from 1992 to 2011) and from DROT results (from 2016 to 2021). L and K mark ice rumples (Rignot et al., 2014). Ice front positions for 1947 and 1966 are from Rignot (2002); later positions (1973-2022) are from Landsat imagery (Landsat-1/4/5/7/8/9) and Sentinel-1 SAR imagery via Google Earth Engine. Red block D denote the region for calculating mean double-differential vertical displacement. The black frame denotes the zoomed-in region in Figure 9a.

Ephemeral grounding could be driven by tidal cycles, ice shelf thinning or thickening, sea-level rise, and sea-level fall—depending on prior grounding conditions (Schmeltz et al., 2001; Rignot, 2002; Matsuoka et al., 2015). The grounding of ice shelf on high bathymetry features could impact ice dynamics as an obstacle against ice flow: 1) enhance the buttressing effect by providing back stress against upstream ice; 2) facilitate fracturing and ice shelf weakening in response to stress associated with grounding (Rignot, 2002; Christianson et al., 2016; Jeong et al., 2016; Shean, 2016; Benn et al., 2022; Wang et al., 2025).

Satellite remote sensing can effectively detect transient vertical motion of ice shelves, especially tidal fluctuations that cause ephemeral grounding. Key methods include differential range offset tracking (DROT) (Marsh et al., 2013; Joughin et al., 2016; Christianson et al., 2016; Wallis et al., 2024, 2025; Lowery et al., 2025; Zhu et al., 2025), interferometric synthetic aperture radar (InSAR) (Schmeltz et al., 2001; Rignot, 2002, 2014), and satellite altimetry (Fricker and Padman, 2006). Both DROT and InSAR methods in theory indicate the landward limit of tidal flexure. While InSAR is widely used to map grounding line

migration, its effectiveness is limited in fast-flowing areas due to phase aliasing unless very short repeat intervals are available. For instance, Milillo et al. (2017) used 1-day repeat COSMO-SkyMed data to track grounding line changes at PIIS.

    In contrast, DROT provides a complementary approach that does not rely on phase information, making it useful for observing vertical tidal displacements on fast-moving ice shelves, despite being less precise than InSAR in some contexts (Marsh et al., 2013; Hogg, 2015; Joughin et al., 2016; Christianson et al., 2016; Friedl et al., 2020; Wallis et al., 2024; Lowery et al., 2025;

Zhu et al., 2025). Using TerraSAR-X data, Joughin et al. (2016) identified a vertical displacement anomaly near ice rumple L from November 2013 to November 2015. At Petermann Glacier, Friedl et al. (2020) found DROT-derived flexure limits ~2 km seaward of DInSAR results. More recently, DROT applied to Sentinel-1 IW data has proven effective for studying grounding line and pinning point dynamics on the Antarctic Peninsula (Wallis et al., 2024), Amery Ice Shelf (Zhu et al., 2025), and PIIS (Lowery et al., 2025). However, Lowery et al. (2025) focused only on the year 2017, leaving later changes unresolved.

Thus, the evolution of grounding behaviour at ice rumple L following four subsequent calving events—in 2015, 2017, 2018, and 2020—remains poorly understood.

    To address this gap, we reconstruct the grounding history of PIIS from 2014 to 2023 using DROT applied to Sentinel-1A/B SAR data. We combine these observations with a 2010–2021 time series of ice thickness change derived from Reference Elevation Model of Antarctica (REMA) DEM (Howat et al., 2022a) and ICESat-2 ATL06 data (Smith et al., 2019; Smith et

al., 2023) to examine the link between ephemeral grounding at ice rumple L and recent changes in ice shelf dynamics. This dataset provides spatially and temporally consistent coverage across the PIIS.

## 2 Methods and Data

### 2.1 Double-differential vertical displacement calculation

    Vertical displacement maps were generated for the PIG basin using the intensity offset tracking algorithm. This involved

applying the algorithm to 420 scenes of Sentinel-1A/B ascending imagery, covering periods of 6- or 12-days from October 2014 to December 2023. Details of the imagery used are provided in Table 1. Processing steps are outlined in Figure 2. We applied fine co-registration and de-ramping procedures prior to offset tracking (Wegmüller et al., 2016; Sánchez-Gámez et al., 2017; Chen et al., 2020). We used the REMA 200 m mosaic DEM (Howat et al., 2019; 2022b), which is posted on a 200 m grid, as the reference DEM for geocoding and co-registering the Sentinel-1 imagery. To compute the displacement fields from

the co-registered and de-ramped imagery, we propose a 2D cross-correlation window of 480×96 (range × azimuth) pixels with step sizes of 100 and 20 pixels in the azimuth and range directions, respectively. We used the REMA 200 m mosaic DEM to geocode the displacement maps based on bicubic-log spline interpolation. The final vertical displacement maps and velocity maps were generated on 100 m×100 m grids and geocoded to the Antarctic Polar Stereographic Projection (EPSG:3031).

**Table 1.** Sentinel-1A/B images used in this study

| Path/frame | Date | Numbers of image pairs |
|---|---|---|
| 65/909 | 2014/10/10 – 2015/11/10 | 76 |
| | 2016/05/20 – 2017/06/20 | |
| 65/908 | 2015/11/22 – 2016/07/07 | 17 |
| 65/910 and 65/911 | 2017/06/14 – 2024/01/03 | 327 |
| | Total | 420 |

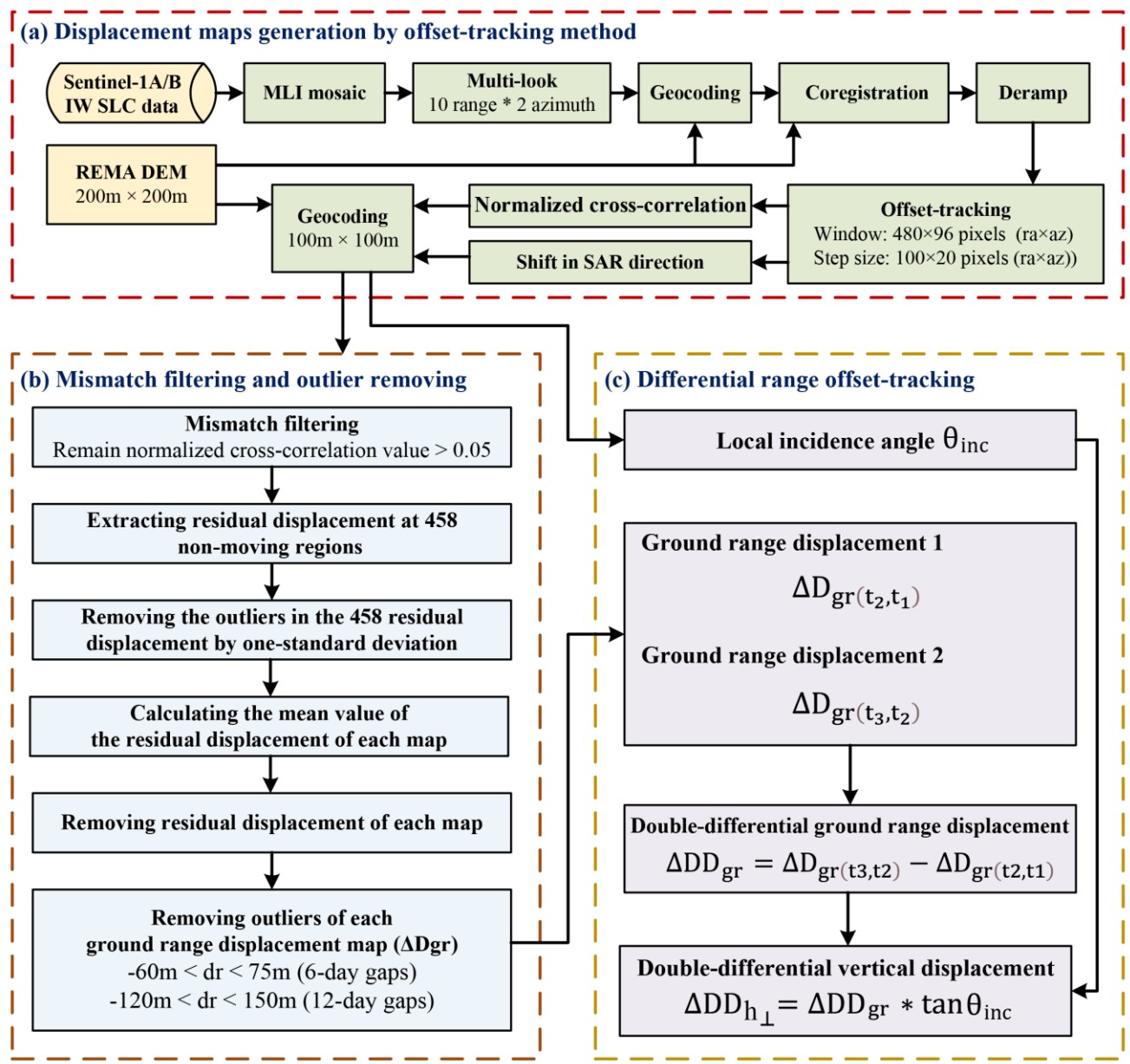

**Figure 2.** Processing steps of range displacement generation and DROT.

To reduce noise and remove outliers in the Sentinel-1 offset tracking data, we employed a multi-step filtering and calibration approach. First, we retained only pixels with a normalized cross-correlation value greater than 0.05, which also used by Solgaard et al. (2021) to ensure reliable displacement measurements. Second, we extracted the residual displacement at 458 widely distributed, non-moving points over the exposed bedrock (Figure 1). Outliers beyond one standard deviation were removed, following the same criteria that used in Chen et al. (2020), and the mean residual displacement for each time interval

was calculated and used to calibrate the displacement maps by subtracting this mean value. To further remove noise and erroneous measurements, we examined the distribution of azimuth and range displacements across the entire time series (Figure S1) and established empirical thresholds based on reasonable minimum and maximum velocities of ice movement at

85 PIIS. We invalidated pixels with slant range displacements less than -60 m or greater than 75 m for 6-day gaps, and less than -120 m or greater than 150 m for 12-day gaps, which can exclude a small portion of pixels and improved the consistency and quality of the final displacement fields.

The slant range displacement fields generated over floating ice contain both horizontal displacement and bias due to vertical ocean motion. When the SAR sensor observes an object $P_{(x,y)}$ from the same location in orbit, the SAR sensor can detect

vertical displacement in the slant range direction ($\Delta D_{sr(t_2,t_1)}$ in Figure 3):

$$\Delta D_{sr(t_2,t_1)} = D_{sr(t_2)} - D_{sr(t_1)} \tag{1}$$

where $D_{sr}$ is the distance between the object $P_{(x,y)}$ and the SAR sensor; $t_1$ and $t_2$ reflect the acquisition time of the master image and the acquisition time of the slave image, respectively. The magnitude of the observed slant range displacement depends on the local incidence angle ($\theta_{inc}$), which is defined as the angle between the incident radar signal and the local surface normal,

expressed in radians When the slant range displacement is converted to ground range displacement, the additional displacement in the ground range ($\Delta D_{gr(t_2,t_1)}$) equals the vertical change ($\Delta h_{\perp(t_2,t_1)}$) divided by $\tan\theta_{inc}$:

$$\Delta D_{gr(t_2,t_1)} = \frac{\Delta h_{\perp(t_2,t_1)}}{\tan\theta_{inc}} \tag{2}$$

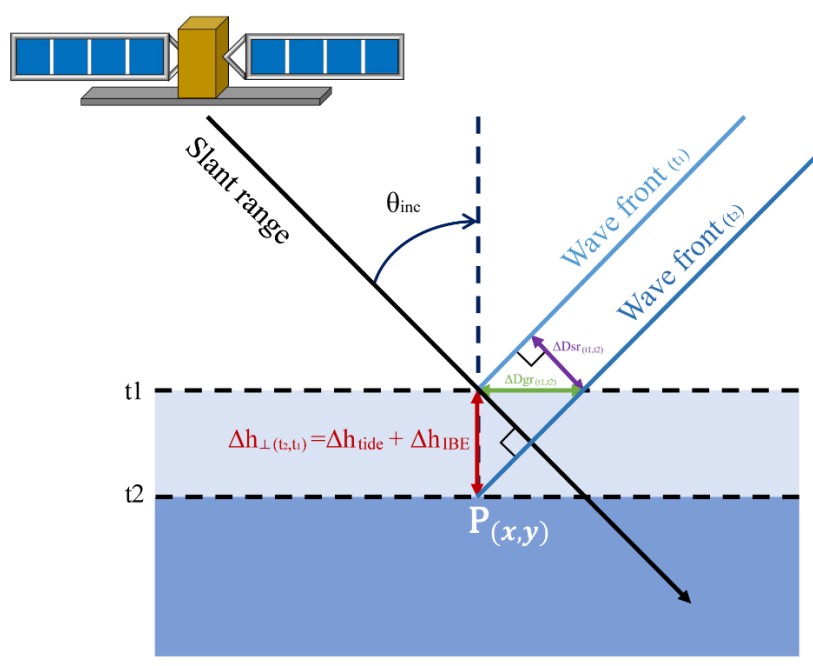

**Figure 3.** Side-looking spaceborne SAR imaging geometry. A vertical displacement of a Point $P_{(x,y)}$ from $t_1$ to $t_2$ is imaged at different slant range positions ($\Delta D_{sr(t_2,t_1)}$) depending on its elevation.

Assuming the horizontal displacement between two SAR image pairs that are closely spaced in time is very small, we can cancel the horizontal displacement and obtain the differential vertical bias in the ground range ($\Delta DD_{gr}$) by differencing two ground range displacement fields (Rignot, 1998; Joughin et al., 2010; Marsh et al., 2013; Christianson et al., 2016; Joughin et al., 2016; Friedl et al., 2020):

$$\Delta DD_{gr} = \Delta D_{gr(t3,t2)} - \Delta D_{gr(t2,t1)} \tag{3}$$

where $\Delta D_{gr(t_2,t_1)}$ and $\Delta D_{gr(t_3,t_2)}$ are the vertical displacement differences in the ground range direction from the displacement map generated from the acquisition dates $t_1$ and $t_2$, and the acquisition dates $t_2$ and $t_3$, respectively. Therefore, the double differential vertical displacement ($\Delta DD_{h_\perp}$) can be calculated as the double differential vertical bias in the ground range ($\Delta DD_{gr}$) from both image pairs multiplied by $\tan \theta_{inc}$:

$$(\Delta DD_{h_\perp} = \Delta DD_{gr} * \tan \theta_{inc} \tag{4}$$

The REMA DEM was used consistently for both $\theta_{inc}$ (in radians) estimation and as the external DEM for co-registration in the offset tracking process, ensuring uniform referencing across displacement fields. The $\theta_{inc}$ was calculated for the first acquisition of each image pair. The local surface normal was derived from the REMA 200 m mosaic DEM. The vertical displacement caused by tidal forcing has minimal impact on the ice shelf's overall surface slope. While slope-induced errors are most significant in areas with localized topographic variability, ephemeral grounding events produce range-direction displacement anomalies that exceed those caused by background slope variations, making these events clearly distinguishable. Consequently, we are confident that using a time-invariant DEM does not compromise the accuracy of our results, as the impact of slope variability on $\theta_{inc}$ and the resulting displacement estimates remains minimal.

Double-differential vertical displacement maps of PIIS were produced using differential range offset tracking, applied to slant range displacement fields. Ephemeral grounding events, indicated by near-zero displacement in the maps (Figure 4a-c) and flattened interferometric fringes in DInSAR (Figure 4d), resulted in visible 'spots'. We analysed each displacement map, noting dates with clear 'spots' at central PIIS, where the area around ice rumple L exhibited near-zero displacement (Figure 4a and 4c). Red block D in Figure 1 denote the region for extracting mean double-differential vertical displacement time series. The double-differential vertical displacement time series was compared with double-differential tidal height time series, derived from the CATS2008_v2023 ocean tide model (Howard et al., 2024) using Tide Model Driver 3.0 (Greene et al., 2023) at (-75.186576°S, -100.617021°W).

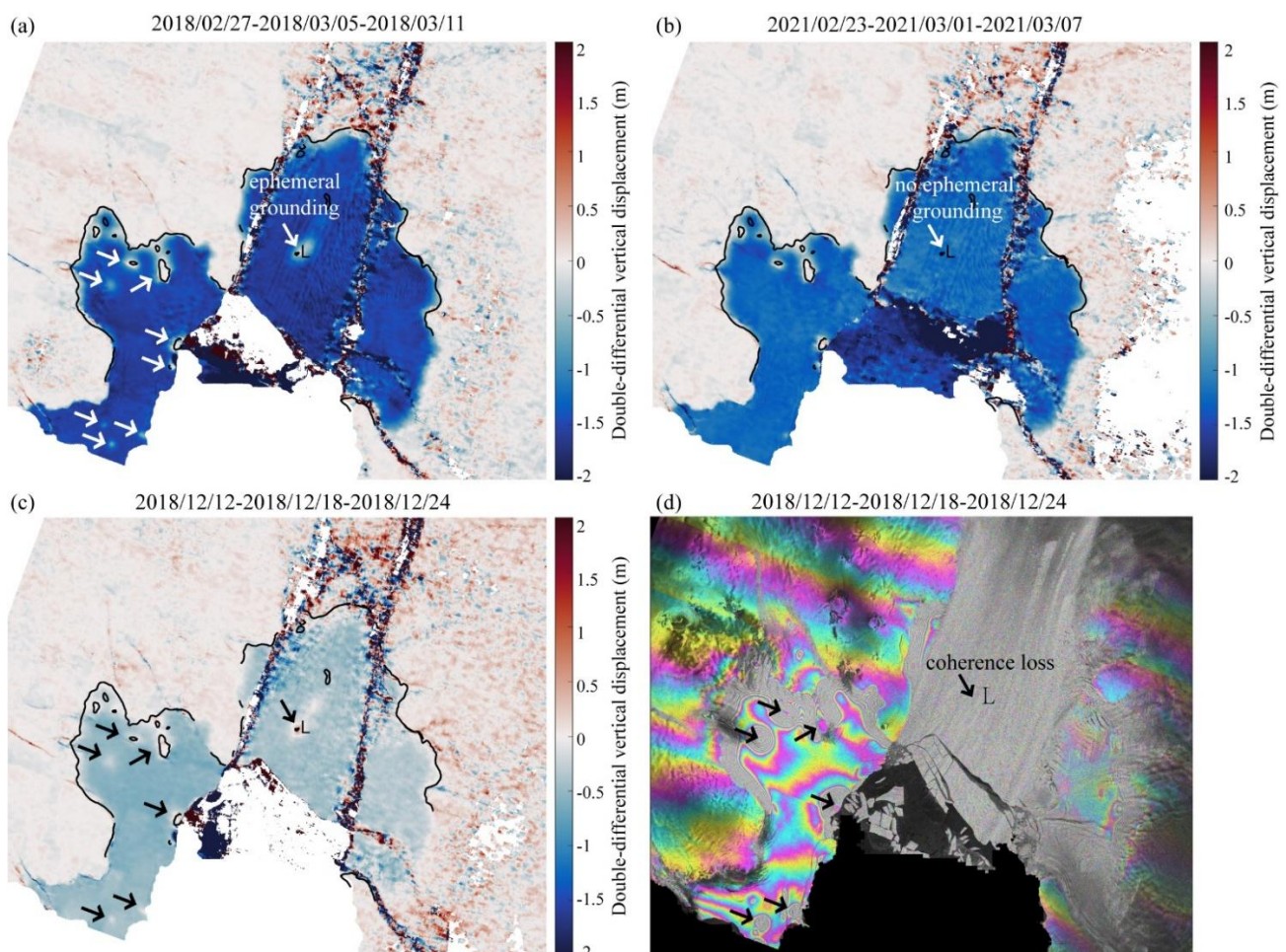

**Figure 4.** Double-differential vertical displacement compared with DInSAR interferogram, showing ephemeral grounding. (a) Double-differential displacement between 2018/02/27-2018/03/05 and 2018/03/05-2018/03/11. (b) Displacement between 2021/02/23-2021/03/01 and 2021/03/01-2021/03/07. (c) Displacement between 2018/12/12-2018/12/18 and 2018/12/18-2018/12/24. White arrows in (a) and (b), and black arrows in (c), indicate the location of ephemeral grounding, marked by near-zero displacement. (d) DInSAR interferogram for 2018/12/12-2018/12/18 and 2018/12/18-2018/12/24. Black arrows highlight ephemeral grounding sites at the northern PIIS. The DInSAR interferogram fails to capture this signal at ice rumple L due to coherence loss.

We extracted grounding line positions using Otsu's method (Otsu, 1979), which determines an optimal global threshold to convert each grayscale image into binary format. Following thresholding, morphological operations were applied to fill holes and close gaps. Grounding line positions were then extracted from the processed binary images.

### 2.2. REMA strips data correction

Elevation data from the CryoSat-2 Baseline-D Level 2 SARIn product (Meloni et al., 2019), spanning from July 2010 to June 2022, were used to correct and co-register the REMA 2 m spatial resolution time-stamped DEM stripes version 4.1 product,

acquired between October 2010 and December 2022 (Howat et al., 2022b). These REMA strips are referenced to the WGS84 ellipsoid but are not co-registered to satellite altimetry by default. The correction and co-registration procedures were implemented using the "Basal melt rates Using REMA and Google Earth Engine (BURGEE)" processing framework developed by Zinck et al. (2023a, 2023b). Processing steps are outlined in Figure 5.

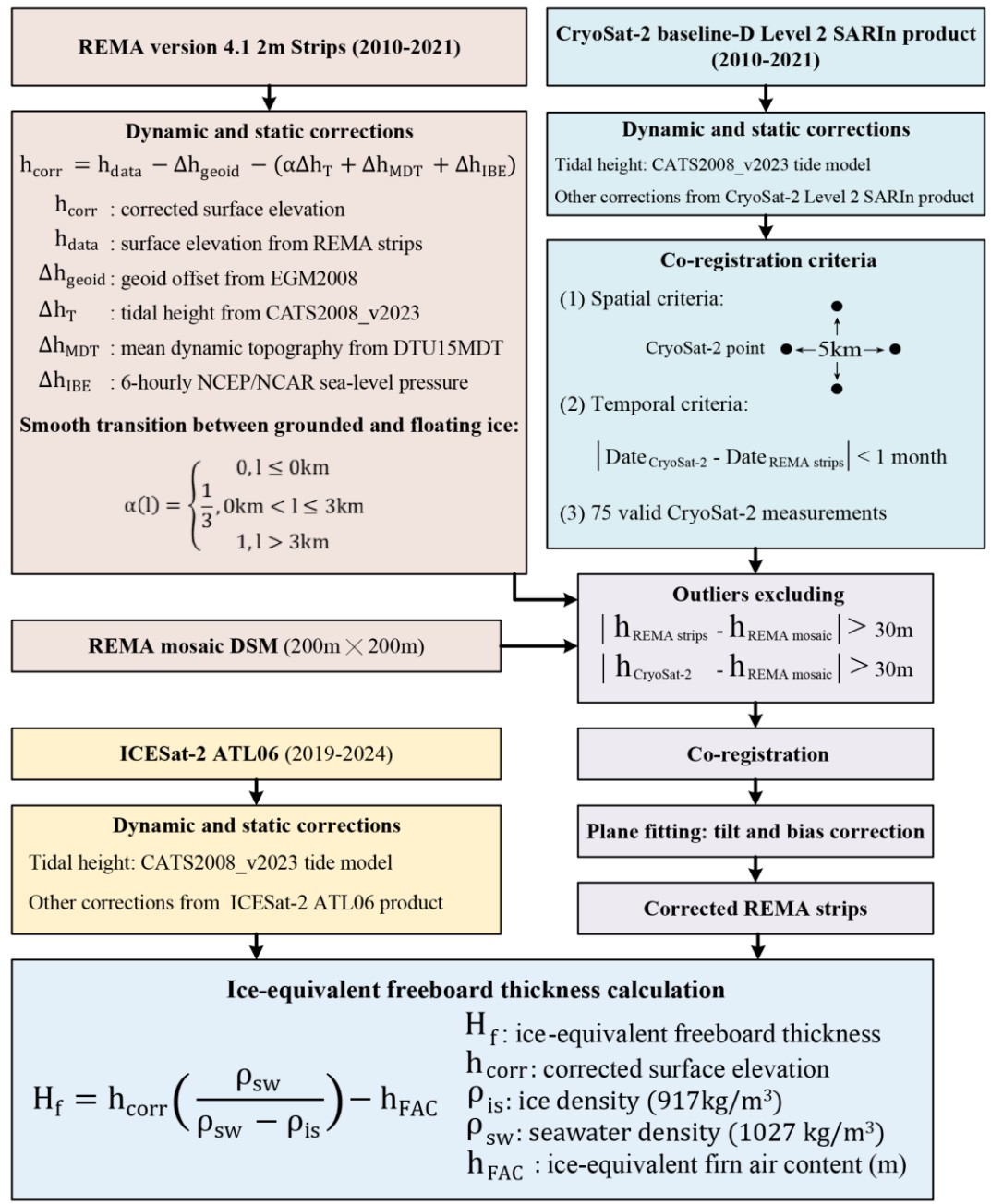

**Figure 5.** Processing steps for correction of REMA DEM, CryoSat-2, and ICESat-2 data.

Dynamic and static corrections were applied to both the REMA strips and the CryoSat-2 dataset to bring all elevations into a consistent reference frame, following the methodology described by Zinck et al. (2023a). For REMA, the corrected surface elevation ($h_{corr}$) was calculated as:

$$h_{corr} = h_{data} - \Delta h_{geoid} - \alpha(\Delta h_T + \Delta h_{MDT} + \Delta h_{IBE}) \tag{5}$$

where $h_{data}$ is the uncorrected surface elevation, $\Delta h_{Geoid}$ is the geoid offset from EGM2008 (Pavlis et al., 2012), $\Delta h_T$ is the tidal height from the CATS2008_v2023 ocean tide model (6-hour intervals, ~3 km resolution), $\Delta h_{MDT}$ is the mean dynamic topography from the DTU15MDT dataset (Andersen et al., 2015), and $\Delta h_{IBE}$ is the inverse barometer effect based on 6-hourly NCEP/NCAR sea-level pressure residuals (Kalnay et al., 1996), referenced to a mean sea level pressure of 1013 hPa. Tidal

and barometric corrections were applied based on the acquisition time of the first stereo image in each DEM strip. The stereo image pairs used to generate the DEMs are typically acquired within a short time interval—usually within minutes to a few hours. Therefore, applying tidal and inverse barometric effect (IBE) corrections based on the acquisition time of the first image introduces only minimal temporal bias. The coefficient $\alpha$ ensures a smooth transition between grounded and floating ice, varying from 0 to 1 with distance from the floating ice edge to the grounding line (Shean et al., 2019), as defined by the ASAID

product (Bindschadler et al., 2011a):

$$\alpha(l) = \begin{cases} 0, & l \leq 0\text{km} \\ \dfrac{1}{3}, & 0\text{km} < l \leq 3\text{km} \\ 1, & l > 3\text{km} \end{cases} \tag{6}$$

The ASAID grounding line product serves as an input to the BURGEE framework and is the same dataset used in Zink et al. (2023a).

CryoSat-2 data were similarly corrected using the same tide model and additional fields from the Level 2 SARIn product

(Howard et al., 2019; Zhang et al., 2020). Erroneous elevation measurements resulting from failed interferometric cross-track positioning were excluded based on quality flags provided by European Space Agency.

To identify and remove elevation outliers, we used the REMA 200 m mosaic DEM (Howat et al., 2019; 2022b) as a reference surface for both the REMA 2 m strips and the CryoSat-2 data. In regions of the PIIS where uncorrected REMA strips exhibited unrealistic elevation changes exceeding 30 m, we applied a more conservative threshold of 100 m elevation difference to

170 exclude outliers.

Co-registration of REMA strips to CryoSat-2 followed a modified procedure from Zinck et al. (2023a), with the following criteria: 1) The longitudinal and latitudinal spacing between CryoSat-2 footprints must be at least 5 km to ensure uniform distribution within the REMA 2m strip data coverage; 2) The acquisition time interval between CryoSat-2 data and REMA strip data must not exceed one month to minimize elevation change impacts over time; and 3) A minimum of 75 valid CryoSat-

2 data points must be distributed within the REMA strip coverage area to enable sufficient data for plane fitting and co-registration, thereby eliminating tilt and vertical bias in the REMA 2m strip DEM.

Residuals between each REMA strip and the CryoSat-2 data were used to apply tilt and vertical shift corrections through plane fitting. The final REMA strips are referenced to the EGM2008 geoid, ensuring both high internal consistency and improved absolute accuracy.

To assess the accuracy of the corrected REMA strips, we compared three strips from 2019–2021 with nearly contemporaneous ICESat-2 ATL06 data (Smith et al., 2019; Smith et al., 2023). The ICESat-2 elevations were converted to heights relative to the instantaneous sea surface by referencing them to the EGM2008 geoid and applying corrections for ocean tides and the inverse barometer effect, following Wang et al. (2021). Processing steps see Figure 5. At overlapping locations between the datasets, we calculated the mean elevation difference (REMA minus ICESat-2) and the standard deviation of this bias. As

shown in Table 2, the corrected REMA strips exhibited lower standard deviations compared to the uncorrected data, indicating reduced uncertainty. However, a consistent negative mean bias remained, with the corrected REMA elevations appearing systematically lower than those from ICESat-2.

**Table 2** The means and standard deviations of uncorrected and corrected REMA strip elevations minus the ICESat-2 elevation.

| Date | Days Gap (day) | Data | Counts | Mean (m) | Standard deviation (m) |
|---|---|---|---|---|---|
| 2019/12/23 2019/12/28 | 5 | Uncorrected REMA strip | 2335 | -5.16 | 9.34 |
| | | Corrected REMA strip | 7285 | -1.14 | 2.85 |
| 2020/01/11 2020/01/09 | 2 | Uncorrected REMA strip | 6551 | 0.23 | 10.11 |
| | | Corrected REMA strip | 7837 | -2.64 | 1.81 |
| 2021/11/30 2021/11/24 | 6 | Uncorrected REMA strip | 827 | 0.76 | 5.99 |
| | | Corrected REMA strip | 802 | -3.77 | 2.56 |
| Total | | Uncorrected REMA strip | 9713 | -1.14 | 10.03 |
| | | Corrected REMA strip | 15924 | -1.93 | 2.54 |

This bias likely results from the differing measurement principles of the two satellite systems: CryoSat-2 (used for REMA

correction) operates in the Ku-band and can penetrate the upper snowpack, whereas ICESat-2 uses green laser altimetry, which reflects off the snow surface. As a result, CryoSat-2—and by extension, the corrected REMA strips—tend to report slightly lower surface elevations than ICESat-2, especially over snow-covered areas. Additional factors such as residual temporal offsets, snow accumulation variability, and surface roughness may also contribute. Based on this comparison, we estimate the uncertainty of the corrected REMA strips as $-1.93 \pm 2.54$ m, equivalent to $15.44 \pm 20.32$ m in floating ice thickness.

Surface elevation changes over the PIIS were derived from the corrected REMA strips. Additionally, MODIS optical imagery from the Images of the Antarctic Ice Shelves Version 2 dataset (Scambos et al., 2022), with a spatial resolution of 250 m and spanning from 1 January 2001 to 23 October 2023, was used to identify changes in surface ridges.

### 2.3 Ice-equivalent freeboard thickness calculation

To estimate changes in ice-equivalent freeboard thickness near ice rumple L, we used both the corrected REMA strips and ICESat-2 data. Specifically, ICESat-2 tracks 965 and 1094, which pass through ice rumple L, were analysed. Ice-equivalent freeboard thickness ($H_f$) was calculated using Equation (7), following the methods of Griggs and Bamber (2011) and Shean et al. (2019):

$$H_f = h_{corr}\left(\frac{\rho_{sw}}{\rho_{sw} - \rho_{is}}\right) - h_{FAC} \qquad (7)$$

where $h_{corr}$ is the corrected surface elevation, $\rho_{is}$ is the ice density (917 kg/m$^3$), $\rho_{sw}$ is the seawater density (1027 kg/ m$^3$), $h_{FAC}$ is the firn air content of ice equivalent (in meters) derived from the NASA GSFC-FDM v1.2.1 dataset (Medley et al., 2022a; 2022b), with a 5-day temporal resolution spanning from 1 January 1980 to 30 June 2022.

### 2.4 Rift propagation observation

Previous studies have suggested that such grounding may be linked to the formation of transverse rifts south of ice rumple L (Joughin et al., 2021), potentially contributing to calving events between 2015 and 2020. However, Joughin et al. (2021) also point out that due to the limitations in the clarity of Sentinel-1 IW SAR imagery hinder a definitive assessment of the connection between ephemeral grounding and rift formation. We used Landsat-8 optical images, specifically the panchromatic band with a 15m spatial resolution, to track the rift propagation history. We then compared these results with our grounding line data to better understand the interaction between ephemeral grounding and rift propagation.

## 3 Results

### 3.1 Changes in the double-differential vertical displacement

Ephemeral grounding region, characterized by double-differential vertical displacements close to zero, shows significant correlation with oceanic tidal variations (Figures 6-7 and Movie S1). The tidal height difference was calculated from data extracted at a point near the ice rumple L (longitude 100.6149°W, latitude 75.1867°S), corresponding to the exact acquisition times of each Sentinel-1 image, which were at 4:35 AM on each date (Supplementary Table S1). One or two near-zero vertical displacement signals were detected at ice rumple L from at least November 2016 through April 2020, followed by a reappearance in December 2020. These signals are highlighted by yellow arrows in Figure 6a and marked by red vertical lines in Figure 6b. The reduced number of signals before August 2016 and after December 2021 likely reflects data limitations during periods when Sentinel-1B was not operational. Near-zero vertical displacement signals also occurred in 2016, 2017,

and after the 2018 calving event. In December 2020, a similar signal appeared upstream of ice rumple L and progressively

migrated toward the rumple, indicating that ephemeral grounding occurred as a thicker section of the ice shelf moved across

the southern side of the sea ridge.

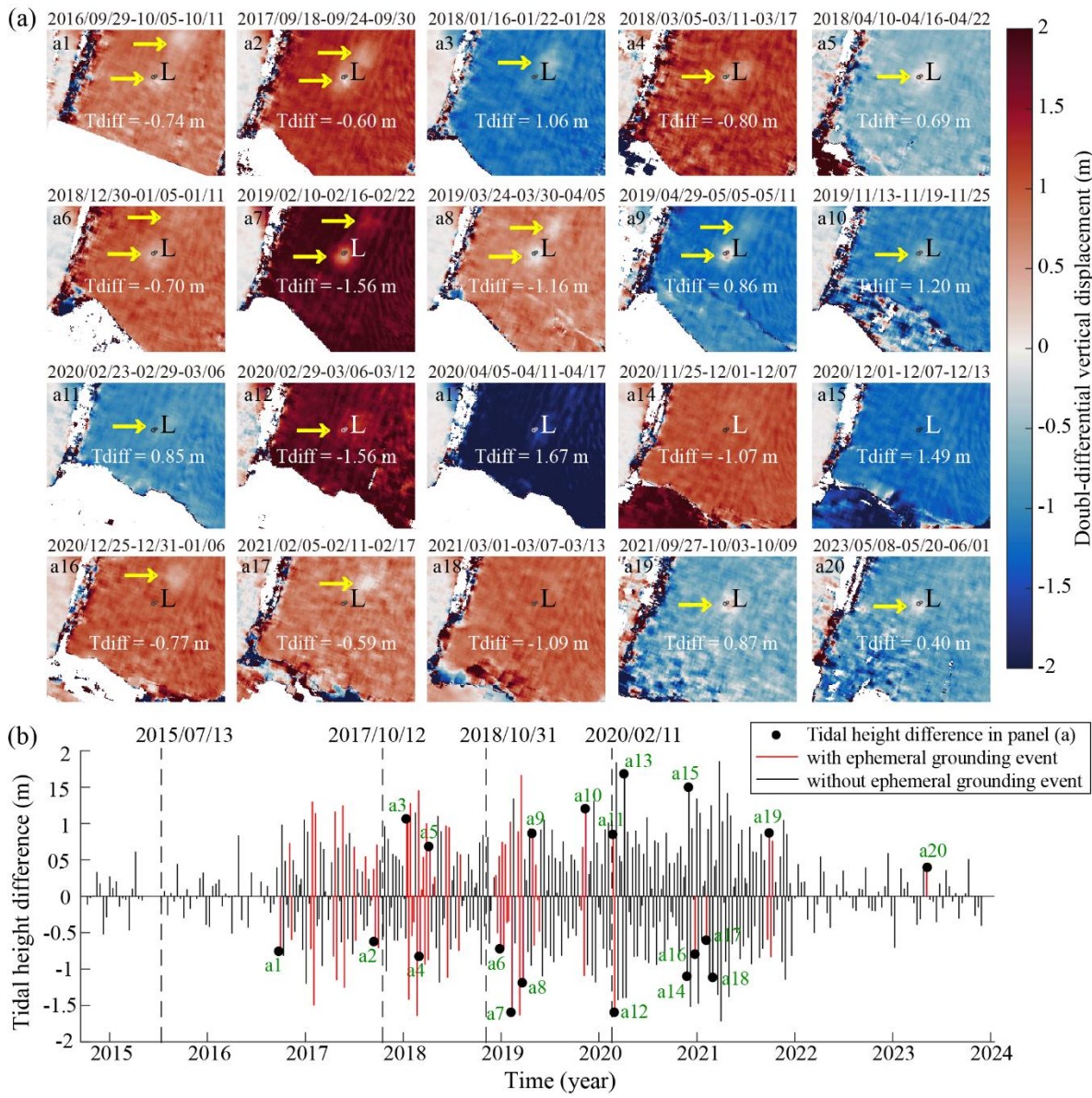

**Figure 6.** Two-dimensional double-differential vertical displacement changes and time series of double-differential tidal height differences. (a) Spatial distribution of double-differential vertical displacement changes between November 2016 and May

2023. Yellow arrows highlight inferred ephemeral grounding signals in each displacement map. The tidal height difference (Tdiff) is labelled in each frame. (b) Time series of double-differential tidal height differences (black vertical lines) and inferred ephemeral grounding events (red vertical lines). Dashed lines indicate the timing of four major calving events: 13 July 2015, 12 October 2017, 31 October 2018, and 11 February 2020.

Figures 6a and 7a demonstrate that positive displacement anomalies generally correspond to negative tidal height differences (and vice versa), indicating an inverse linear relationship between these variables. However, Figure 7b reveals no clear correlation between tidal height and grounding region area, suggesting ephemeral grounding is not solely controlled by tidal forcing. Figure 7c shows 64 ephemeral grounding events from November 2016 to March 2021, with 35 occurring during neap tides and 29 during spring tides. From Figures 7c and 7d, it can be observed that larger grounded areas are evident during spring tides, when tidal amplitudes reach their maximum, while smaller grounded areas are observed during neap tides, when tidal heights are at their lowest. These patterns suggest that the variability in grounded area is reflecting the periodicity of tides. Together with Figure 6a, which shows the changes of the two near-zero vertical displacement signals, it suggests that thick ice advection from upstream may contribute to the grounding events. Therefore, ice dynamics likely play a significant role in the grounding process as well.

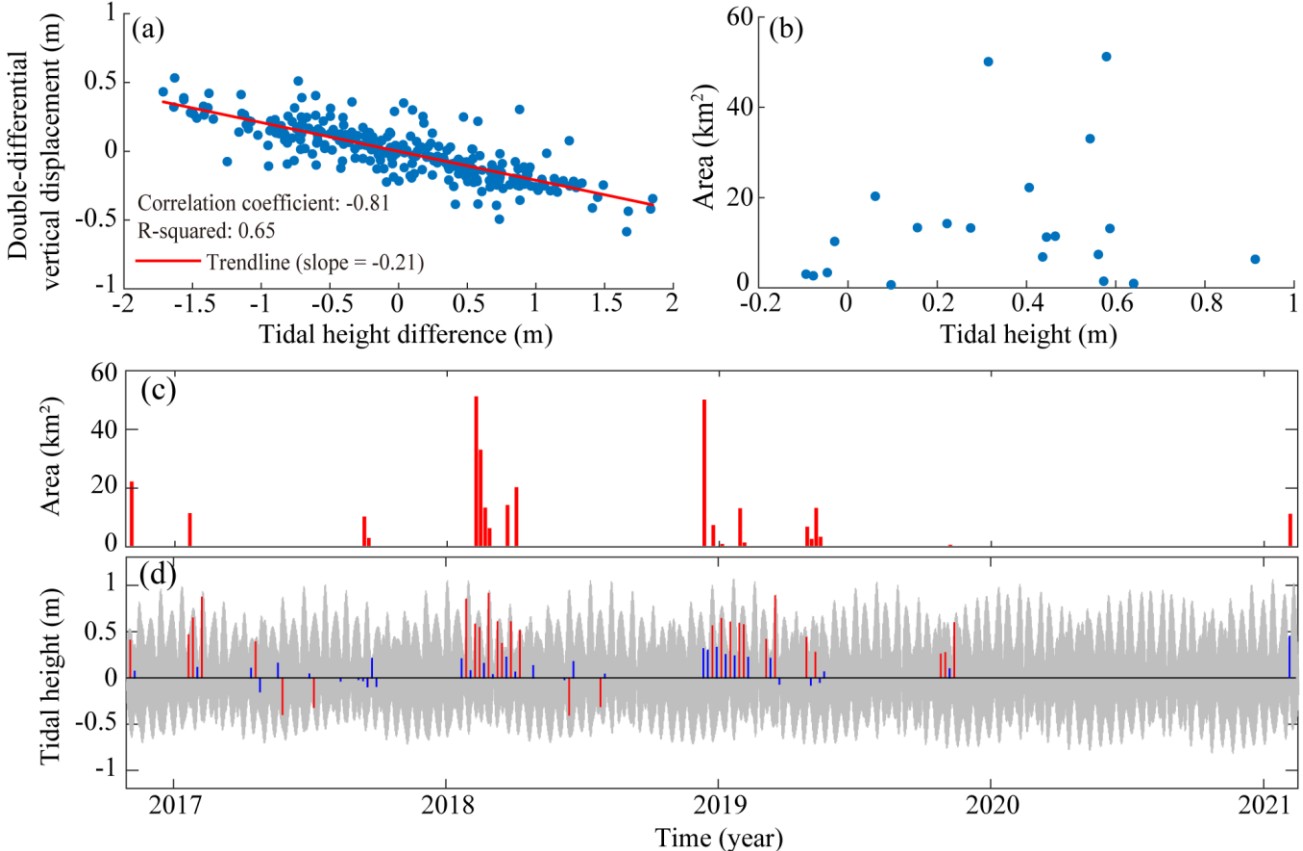

**Figure 7.** Comparison of tidal height differences with double-differential vertical displacement, comparison of tidal height differences and area of grounding region, including time series of area and tidal height variations. (a) Scatter plot of tidal height difference versus double-differential vertical displacement, showing a strong negative linear correlation between the two variables (Pearson's r = –0.81, R² = 0.65, slope = –0.21). (b) Scatter plot of tidal height versus area of zero vertical displacement region, indicating no clear relationship between the two datasets. (c) Time series of changes in ice rumple area. (d) Time series of tidal height changes, where 0 represents mean sea level. Blue vertical lines indicate ephemeral grounding events during the neap tide period, while red vertical lines represent those during the spring tide period.

## 3.2 Changes in surface features and ice thickness

Figure 8 shows the evolution of the surface ridges' elevation and grounding areas using double-differential vertical displacement calculation (Section 2.1) from December 2010 to January 2021. Some ridges higher than 75 m were advected from upstream and passed through the area near the ice rumple L (Figure 8b-k). Near ice rumple L (red point in Figure 8), surface elevations remained around ~65 m between 2012–2017 and again during 2019–2020 (Figures 8d–h and 8j–k). The highest elevation (~85 m) was recorded in 2018, while the lowest (~54 m) occurred in 2021. Between 2020 and 2021, surface elevation declined by ~10 m, equivalent to ~70 m of ice-equivalent freeboard thickness. The grounding line–enclosed area—corresponding to the region of zero vertical displacement—was largest in 2018 (Figure 8i).

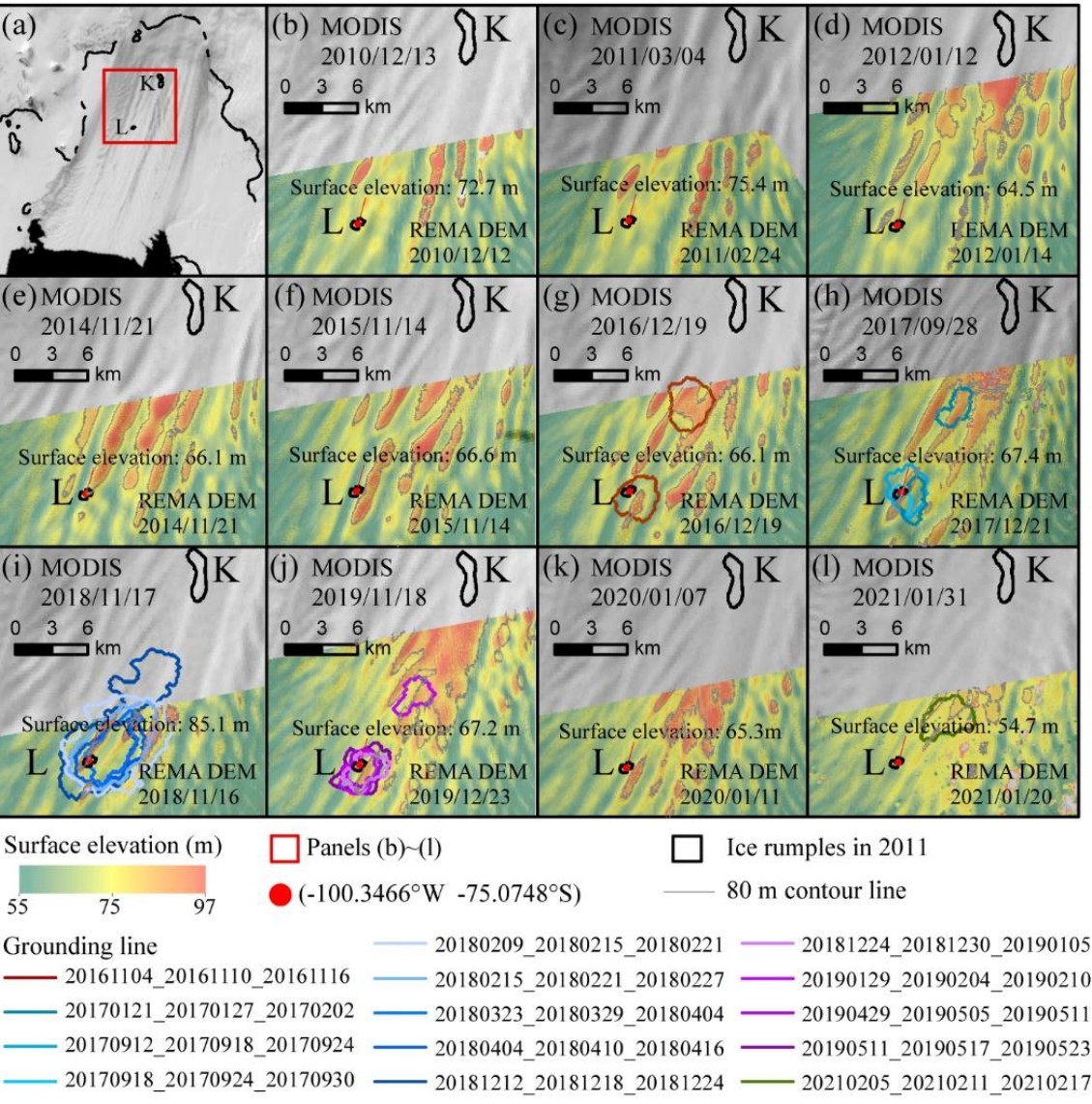

**Figure 8.** Changes in surface ridges at PIIS near ice rumple L. (a) Overview map showing the subregion outlined by the red frame, corresponding to panels (b) to (l). (b)–(l) Surface ridges and their elevation changes from 2010 to 2021, derived from corrected REMA strips. The two black circles indicate the positions of ice rumples. Grounding lines are delineated based on the zero-contour of the double-differential vertical displacement. Grey lines are the 80m contour line.

Profiles of ice-equivalent freeboard thickness derived from ICESat-2 (Figure 9) link surface elevation and grounding area

changes. ICESat-2 has three pairs of beams, each consisting of a strong and a weak beam. For our analysis of ice thickness

changes, we selected two pairs of beams (gt2l, gt2r, gt3l, and gt3r) that pass through the ephemeral grounding region. The

strong beams transmit with higher energy than the weak beams, and the weaker beams are positioned to the left of their paired

strong beams. Thus, in our ICESat-2 data, 'gt2l' and 'gt3l' correspond to the weak beam positions, while 'gt2r' and 'gt3r'

correspond to the strong beam positions (Figure 9a). Figure 9b shows mean thickness trends around the rumple along ICESat-

2 tracks 965 and 1094 between 75.15°S and 75.05°S (Figure 9a). Track 965 reveals increasing ice thickness from 2015 to 2021,

while track 1094 shows a decrease from 2015 to 2017, a rebound in 2018, and a decline after 2020. Bottom elevation profiles

derived from ICESat-2 (Figures 9c-f) further reveal changes in grounding status. The ice shelf was ungrounded on 27 August

2020, 5 March 2021, and 25 May 2022, but showed weak grounding on 6 June 2020. Figures 9e and 9f suggest that the bed

elevation beneath the rumple is likely too high in the BedMachine v3 dataset (red dashed line). Therefore, our results could

help correct this potential error in the BedMachine v3 dataset. By integrating double-differential vertical displacement data

with bottom elevation profiles, we find that ephemeral grounding signatures disappeared after March 2020 and reappeared in

November 2020.

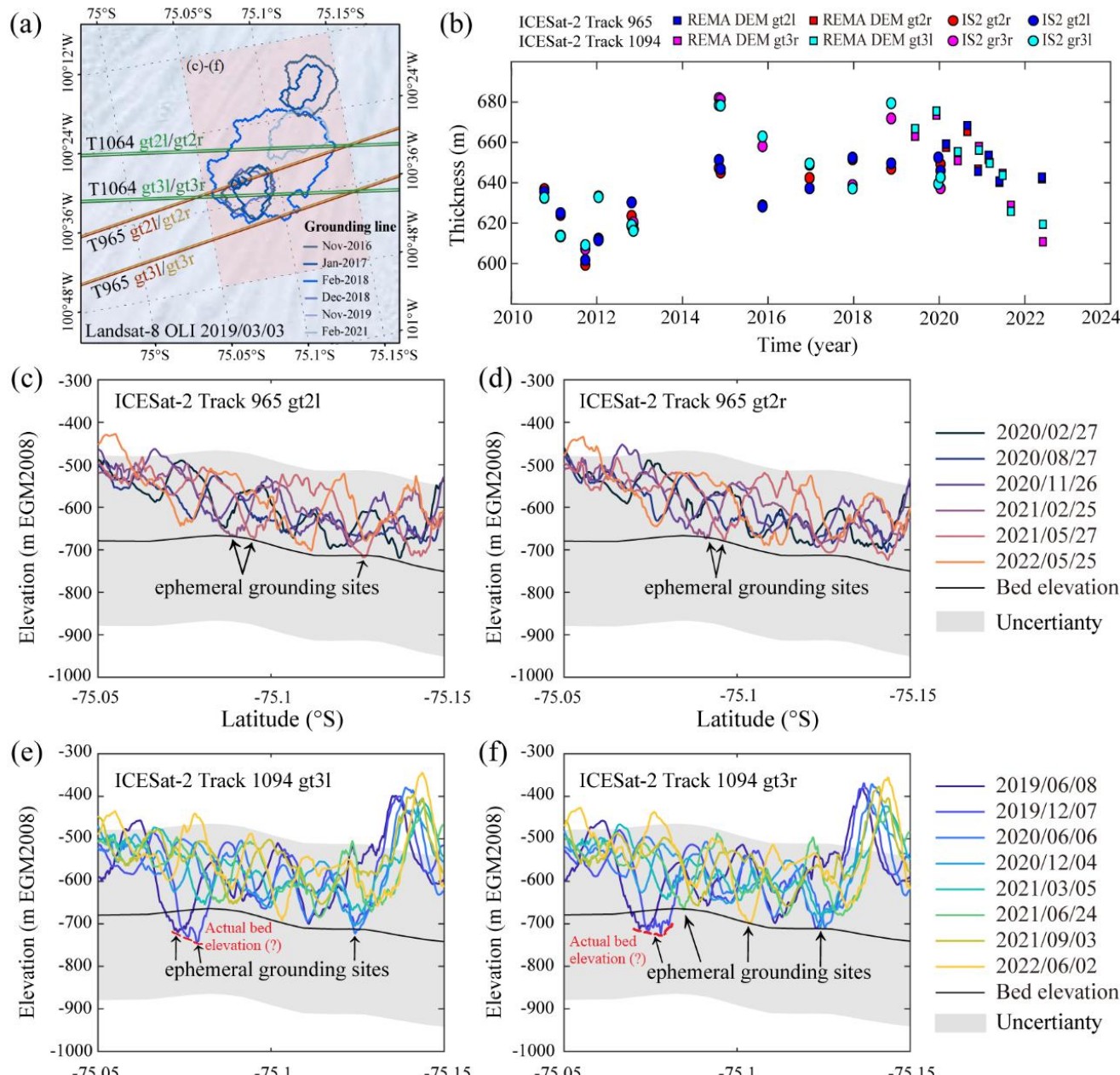

**Figure 9.** Time series of mean ice-equivalent freeboard thickness and ice shelf bottom elevation profiles along ICESat-2 tracks 965 and 1094. (a) ICESat-2 track 1094 and track 965 that used for ice-equivalent freeboard thickness change analysis and grounding lines near the ice rumple L from April 2011 to February 2021. Background is from Landdsat-8 OLI optic image on 3 March 2019. (b) Time series of mean ice-equivalent freeboard thickness (2010–2022). (c)–(d) Ice shelf bottom elevation profiles along ICESat-2 tracks 965 (gt2l and gt2r) between February 2020 and May 2022. (e)–(f) Ice shelf bottom elevation profiles along ICESat-2 tracks 1094 (gt3l and gt3r) between June 2019 and June 2022. Bed elevations are from the BedMachine v3 dataset (Morlighem et al., 2020; Morlighem, 2022), converted from EIGEN-6C4 to the EGM2008 geoid to match the

vertical datum of REMA strips. The estimated vertical uncertainty is ±200 m (shown as a grey transparent box). The potential actual bed elevation is marked by a red dashed line.

## 3.3 Rift propagation history from 2013 to 2019

Using Landsat images, we tracked the propagation history of the rifts from 2013 to 2019 (Figure 10). Rift R1 first appeared in the image from 15 December 2017 (Figure 10e), after the region passed through the ephemeral grounding zone, as seen in Figure 10c. Similarly, Rift R2 appeared in the 11 December 2018 image (Figure 10g), following its passage through the same grounding region. These two rifts ultimately led to the 2020 calving event. Therefore, our results suggest that ephemeral grounding events are linked to rift propagation, indirectly influencing the ice shelf calving process.

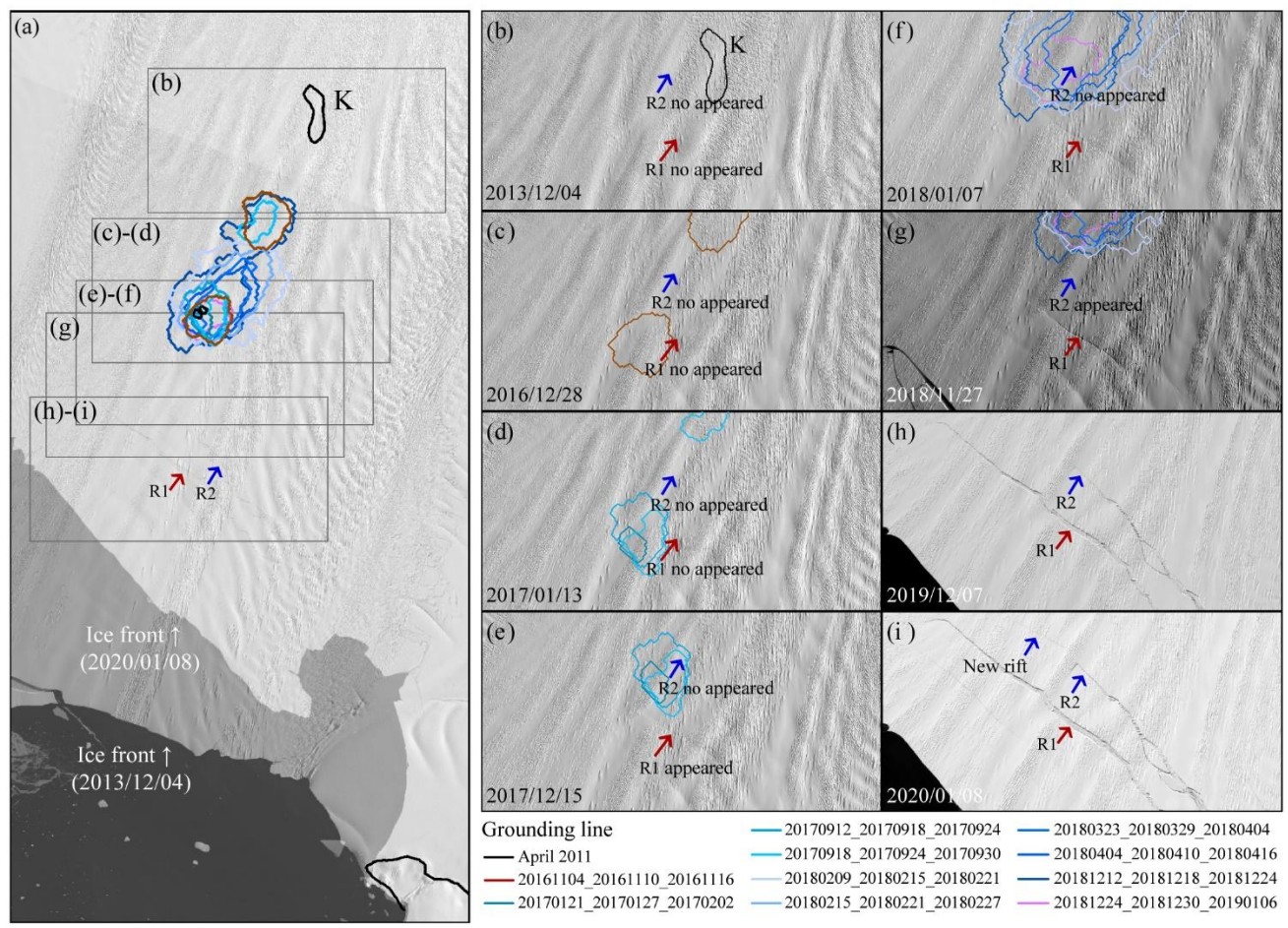

**Figure 10.** Rift propagation history from 2013 to 2019. (a) Overview map showing the positions of panels (b) to (i). The background image is a Landsat-8 panchromatic image from 8 January 2020, overlaid on another Landsat-8 panchromatic image from 4 December 2013, of Pine Island Glacier. (b)-(i) show the propagation history of the rifts R1 (red arrow) and R2 (blue arrow), which led to the 2020 calving event. The black circles indicate the positions of ice rumple K. Grounding lines

are delineated based on the near-zero value of the double-differential vertical displacement.

## 4 Discussion

By integrating vertical displacement patterns, tidal height differences, and ICESat-2-derived ice thickness profiles, we captured ephemeral grounding of PIIS between 2014 and 2023. These findings emphasize the importance of combined geodetic and altimetric observations in resolving ephemeral grounding behaviour.

Our results reveal recurring of ephemeral grounding at ice rumple L from at least November 2016 through April 2020, followed by a reappearance in December 2020. This signal is modulated by tidal dynamics and variations in ice shelf thickness. Near-zero vertical displacement signals were observed during multiple years and were most prominent during spring tide periods when tidal amplitudes were highest. This finding supports the idea that tidal variations can modulate the vertical position of the ice shelf base, causing it to intermittently contact the seafloor and resulting in ephemeral grounding (Minchew et al., 2017).

The dual-satellite configuration significantly enhanced detection capabilities for ephemeral grounding events. When both Sentinel-1A and Sentinel-1B were operational, their combined 6-day repeat cycle increased the probability of capturing imagery during periods of large tidal variation, when ephemeral grounding is most readily observable. However, during single-satellite periods—before Sentinel-1B's launch in April 2016 and after its failure in January 2022—the extended 12-day repeat cycle of Sentinel-1A alone substantially reduced opportunities to coincide with optimal tidal conditions, hampering detection

of these transient phenomena. This temporal sampling limitation underscores how the deployment of higher-resolution SAR satellites with improved revisit frequencies will enhance our ability to observe ephemeral grounding events, ultimately enabling the construction of denser, more temporally continuous records of grounding line dynamics. Our grounding line results also highlight that the DROT method can derive more detailed information than DInSAR at the fast-moving ice shelf, providing a valuable dataset for modelling input.

Thickness of ice advected from upstream has also observed to modulate the grounding of the rumple. Notably, the surface elevation peaked in 2018 and declined significantly between 2020 and 2021, coinciding with changes in grounding behaviour. Near-zero vertical displacement signals, indicative of ephemeral grounding, were detected at ice rumple L from November 2016 through April 2020. These signals disappeared during the 2020–2021 thinning period but reappeared in December 2020. In that instance, a similar signal emerged upstream of the rumple and gradually migrated toward it, suggesting that a thicker

section of the ice shelf had moved over the sea ridge, re-establishing ephemeral contact with the bed. With time series of ephemeral grounding activities, the accurately derived ice draft elevations could be used to correct the bed elevation under the ephemeral grounding area, which could be important to ice dynamics modelling study of PIG.

In summary, our study demonstrates that ephemeral grounding at ice rumple L is modulated by the interaction between tidal forcing, ice shelf thickness, and evolving sub-ice geometry. These results provide new insights into the mechanisms driving

ephemeral grounding behaviour. Notably, we find the rift that caused the 2020 calving event appeared after pass through the ephemeral grounding region. Arndt et al. (2018) emphasized the importance of final pinning points in controlling calving line

orientation, raising the possibility that ice rumple L may have acted as a final pinning point after the 2015 calving event, thereby influencing rift propagation and subsequent calving. Previous studies (Sun and Gudmundsson, 2023; Joughin et al., 2021) have suggested that calving is the key process causing the speedup of PIG after 2017. These findings underscore the need for high-resolution ice shelf modelling to evaluate how ephemeral grounding affects stress redistribution and overall ice shelf stability.

De Rydt et al. (2014) demonstrated that both the height of the ridge and the gap between the ridge and the ice shelf strongly influence the inflow of warm bottom waters into the cavity, and consequently, the melt rate. The melt rate may influence the ice thickness near to the grounding line upstream than the ice rumples K and L. This process may have contributed to the ice thickness changes upstream and indirectly influenced the disappearance of ephemeral grounding signals following the 2020 calving event. We have added further analysis on the basal melt rate and ocean temperature in the Appendix A. Although smaller-scale basal channels and keel geometries are primarily shaped by melt-driven processes (Bindschadler et al., 2011b; Dutrieux et al., 2013; Stanton et al., 2013; Dutrieux et al., 2014b; Joughin et al., 2016), the lack of direct, high-temporal-resolution basal melt rate measurements after 2020 limits our ability to capture short-lived grounding events and confirm the role of ocean-driven melting. Future work should prioritize the integration of dense time series from new SAR missions and in situ oceanic data to better resolve ephemeral grounding behaviour and its implications for ice shelf evolution and calving dynamics in a warming climate.

**5 Conclusion**

This study presents the time series of ephemeral grounding events between 2014 and 2023 at the central PIIS, based on DROT applied to Sentinel-1 SAR data. By integrating double-differential vertical displacement maps, tidal height differences, and thickness data calculated from surface elevation data from REMA strips and ICESat-2, we show that ephemeral grounding is modulated by the combined effects of tidal forcing, evolving sub-ice geometry, and changes in ice shelf thickness. Near-zero vertical displacement signals—indicative of intermittent grounding—were repeatedly observed throughout the study period, particularly as the grounded area expanded during spring tides with large tidal amplitudes. Changes in ice thickness also play an important role in driving ephemeral grounding at the PIIS.

We show that ice shelf thickening preceded grounding events, while thinning contributed to ungrounding. The presence and migration of near-zero displacement signals suggest that thicker ice flowing over topographic highs can cause ephemeral grounding. Observed large-scale surface and basal structures, including keels and channels, reflect the influence of inherited bed topography, while smaller-scale geometries could shape by basal melt processes modulated by ocean temperature variability. We also show that the rifts responsible for the 2020 calving event appeared after the region passed through the ephemeral grounded area, suggesting that these ephemeral grounding events may have changed the stress distribution of the ice front and contributed to the formation of the rifts.

Our findings demonstrate the highly accurate remote-sensing techniques for monitoring grounding processes. The grounding lines derived from our DROT results can be scaled up to regional applications and provide critical boundary conditions for ice flow modelling efforts. We also reveal that ephemeral grounding influences stress redistribution, calving dynamics, and the long-term stability of vulnerable ice shelves of PIG. These observations could be used to validate the relevant processes in numerical modelling, which is currently poorly represented. In the future, improved satellite coverage, denser SAR time series, and in situ ocean measurements will provide comprehensive database to apply our method in deriving grounding line behaviours of much larger scale.

## Appendix A. Oceanic condition changes and analysis

To address the oceanic condition changes, we extracted time series data on mean basal melt rates from 2010 to 2017 using the MEaSUREs ITS_LIVE Antarctic Quarterly 1920 m Ice Shelf Height Change and Basal Melt Rates v1 dataset (Paolo et al., 2023; 2024). This dataset offers quarterly basal melt rate estimates, with uncertainties, from 17 March 1992 to 16 December 2017, at a 1920 m spatial resolution. However, these estimates are based on surface elevation changes from radar altimetry and ice fluxes from the Glacier Energy and Mass Balance model, not direct observations. Additionally, it does not cover our primary observation period from 2020 to 2023.

To further investigate oceanic influences, we examined ocean temperature time series from the PIG-N (longitude 102.0987°W, latitude 74.8644°S) and PIG-S (longitude 102.1588°W, latitude 75.0546°S) mooring locations using mooring data (Zhou et al., 2024; 2025). These records span from 2016 to 2024 and capture temperature variations at depths of 300–700 meters below mean sea level. This pan-Antarctic mooring compilation contains data on temperature, salinity, and current velocity in the Southern Ocean (90°S–60°S) since 1975, with contributions from data centres, research institutes, and individual data owners (Zhou et al., 2024). However, the moorings located in Pine Island Bay and not directly beneath the ice shelf, which limits their applicability to sub-shelf melting processes.

Profiles of ice-equivalent freeboard thickness derived from ICESat-2 link surface elevation and grounding changes (Figure A). Figure Aa shows mean thickness trends around the rumple along ICESat-2 tracks 965 and 1094 between 75.15°S and 75.05°S (Figure 9b). Track 965 reveals increasing ice thickness from 2015 to 2021, while track 1094 shows a decrease from 2015 to 2017, a rebound in 2018, and a decline after 2020.

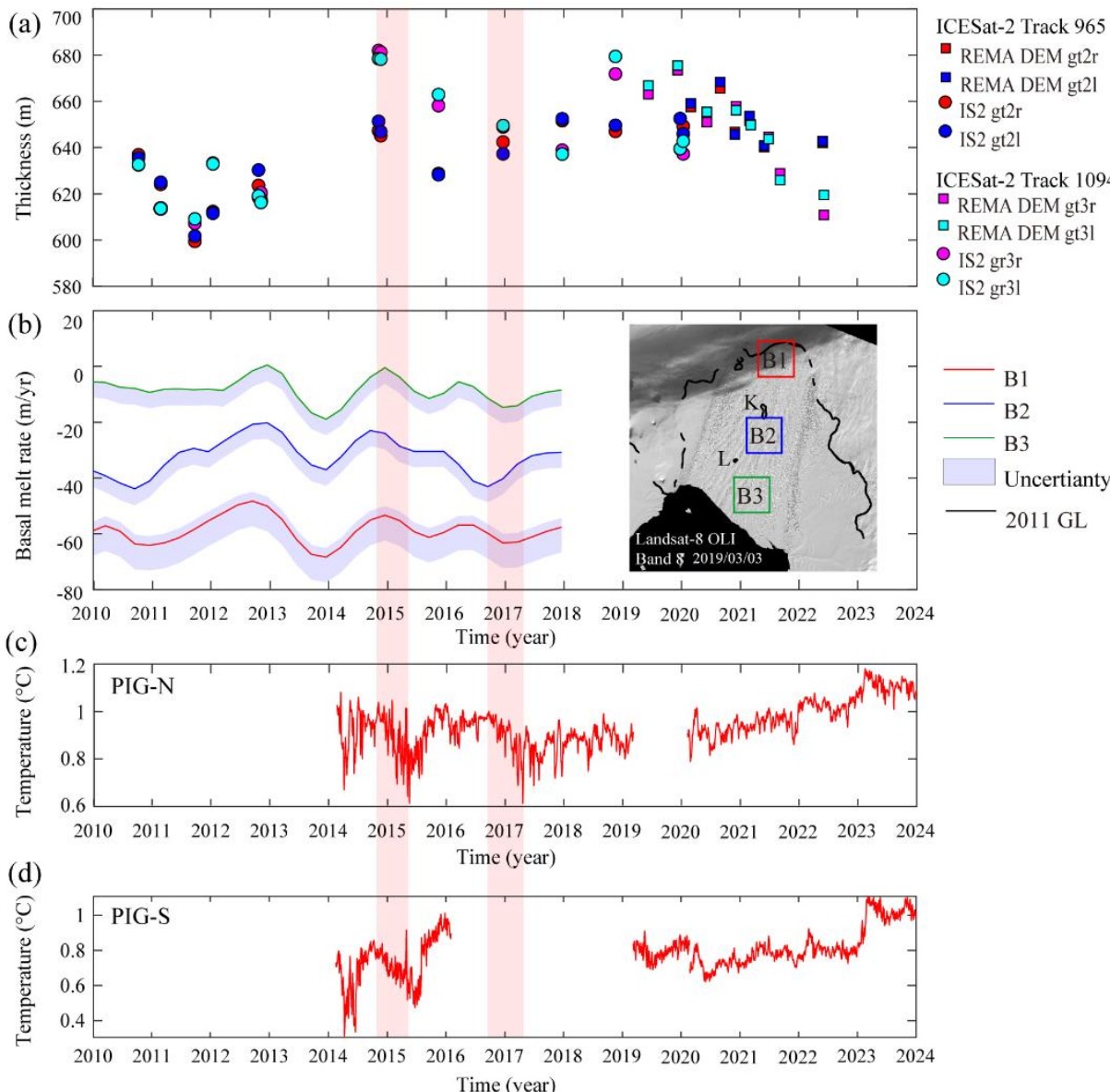

**Figure A.** Time series of mean ice-equivalent freeboard thickness, basal melt rate, and ocean temperature. (a) Time series of mean ice-equivalent freeboard thickness (2010–2022). (b) Time series of mean basal melt rate (2010–2017), averaged across blocks B1, B2, and B3, extracted from the MEaSUREs ITS_LIVE Antarctic Quarterly 1920 m Ice Shelf Height Change and Basal Melt Rates v1 dataset (Paolo et al., 2024). (c)-(d) reveal time series of ocean temperature at the PIG-N and PIG-S mooring stations from 2014 to 2024.

The basal melt rate time series show a decrease in melting around 2015, coinciding with a peak in ice-equivalent freeboard thickness at all three locations (B1–B3; Figure Ab). During the same period, ocean temperatures near 600 meters depth decreased at both the PIG-N and PIG-S mooring stations (Figures Ac and Ad). At B2, located between ice rumples L and K, the basal melt rate increased after 2015 but declined again after 2017 (Figures Aa and Ab). This decline corresponds with a

drop in ocean temperature recorded at PIG-S (Figure Ad). However, from 2020 to 2023, ocean temperatures near 600 m depth at PIB showed a continuous increase, which could have contributed to enhanced basal melting of the ice shelf during that time (Figures Ac and Ad).

Smaller-scale basal channel and keel geometries are primarily shaped by melt-driven processes (Bindschadler et al., 2011b; Dutrieux et al., 2013; Stanton et al., 2013; Dutrieux et al., 2014b; Joughin et al., 2016). Mooring observations from 2014 to 2024 reveal two distinct periods of ocean temperature decline around 2015 and 2017 (Figure Ac, d), during which basal melting near ice rumple L also decreased (Figure Ab). Following 2020, however, ocean temperatures began to rise again.

Correspondingly, ice thickness time series (Figures Aa) show a substantial thinning of approximately 70 m. Although direct basal melt rate measurements are unavailable for this period, the observed warming at 600 m depth near PIB suggests that the ice shelf base may have reached this depth, potentially enhancing basal melting. This increased melting would have further thinned the ice shelf, thereby widening the thickness gap between the ice base and the submarine ridge.

In summary, these data show periods of temperature decline around 2015 and 2017, which were accompanied by reduced ice

thickness near ice rumple L, followed by warming after 2020 and a corresponding ice shelf thinning of approximately 70 meters. However, direct basal melt rate measurements are unavailable for the post-2020 period (Figure A). While the observed warming at 600 m depth near PIB suggests increased basal melting that likely contributed to the thinning, variations in ocean temperature and basal melt rates alone cannot fully explain the observed changes in ice shelf thickness or influence the small-scale keels.

***Code and sample availability***: All codes and processed time series data used for analysis and plotting in this study are available from Chien et al. (2025a), including ice front positions delineated from Landsat panchromatic imagery and Sentinel-1 SAR imagery based on Google Earth Engine, double-differential vertical displacement, corrected REMA strips,  and MODIs images for Figure 8. The zenodo link provided in Chien et al. (2025a) will be made public after acceptance of the paper. The grounding lines extracted from the double-differential vertical displacement map are available in the supplementary material

of this study. The Sentinel-1 image IDs and ephemeral area can be accessed in Supplementary Tables S1 and S2. Reviewers can access the code and datasets through the link below:

https://zenodo.org/records/15844913?token=eyJhbGciOiJIUzUxMiJ9.eyJpZCI6ImY5MTY3ZGI4LTZmYWQtNDcwOS05ZmFiLTQyMTU4YzVhMjRhZSIsImRhdGEiOnt9LCJyYW5kb20iOiI4YmQ2MTY2Nzg4ZTUyYzNhYWZiYmMyZDQ4MDg1NmM1ZSJ9.VNGqgibhOoN5KN39EhcRyTK3Ras3T79O83lszsJ0ag05foJxtk3BK63HlGGoKT6-

425 XCSHBwmcMB046GqCYPS4NQ

***Data availability:*** All software (except GAMMA, which is commercial software and was used to generate displacement in slant-range direction), codes, and satellite and climate datasets used in this study are publicly available and can be obtained from the following sources: The MATLAB plotting codes on which this article is based are available in Greene et al (2017) and Greene et al. (2021). The BURGEE codes for corrected REMA strips are available in Zinck et al. (2023b). The tidal model

driver based on MATLAB code is available in Greene et al. (2023). Sentinel-1 images are available for free download from the Alaska Satellite Facility website at https://asf.alaska.edu/. Processed MODIS images are available in Scambos et al. (2022). BedMachine version 3 dataset is from Morlighem (2022). REMA 200 m DEM mosaic and REMA 2 m DEM strips are available

from Howat et al. (2022a) and Howat et al. (2022b), respectively. CryoSat Baseline-D SARIn Level 2 data are available on the ESA CryoSat Science Server at https://science-pds.cryosat.esa.int/#Cry0Sat2_data%2FIce_Baseline_ D%2FSIR_SIN_L2. ICEsat-2 Level 2 ATL06 product is available from Smith et al. (2023). ASAID grounding line products are available from Bindschadler et al. (2011a), Rignot et al. (2016), and this study (Chien et al., 2025a). Firn air content is available from Medley et al. (2022b). 6-hourly NCEP/NCAR sea-level pressure is tagged in Google Earth Engine (NCEP_RE_sea_level_pressure). Basal melt rate product can be accessed from Paolo et al., (2024). The ocean temperature time series at the PIG-N and PIG-S mooring locations is available and regularly updated in NetCDF format via the SEANOE database at https://doi.org/10.17882/99922 (Zhou et al., 2024).

***Video supplement:*** Movie S1 "Double-differential vertical displacement changes from November 2014 to November 2023 at the PIIS" can be accessed at the zenodo link provided by Chien et al. (2025b).

***Author contribution:*** YC, CZ, and SS designed the experiments and YC carried them out. YC developed the MATLAB code and performed all the experiments. BZ provided the corrected CryoSat-2 dataset. YC prepared the manuscript with contributions from all co-authors.

***Competing interests:*** The contact author has declared that none of the authors has any competing interests.

***Acknowledgements:*** This research is funded by the National Natural Science Foundation of China (41941010 and 42171133) and the Fundamental Research Funds for the Central Universities (2042024kf0016). We thank all the organizations or projects listed in Open Research. We thank the anonymous reviewers and editors for their insightful comments to improve the manuscript. We sincerely thank Anne Solgaard and Anders Kusk for providing helpful suggestions related to SAR image post-processing. We sincerely thank Jan Wuite for providing helpful suggestions related to tidal correction. We sincerely thank Hilmar Gudmundsson for providing helpful suggestions related to the ephemeral grounding change analysis.

***Financial support:*** This research is funded by the National Natural Science Foundation of China (42171133 and 41941010) and the Fundamental Research Funds for the Central Universities (2042024kf0016).

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
