# Peer review of "Ephemeral grounding on the Pine Island Ice Shelf, West Antarctica, from 2014 to 2023"

_EGUsphere, 2025_

## Referee Comment (RC1)

**Summary:**

This study uses differential range offset tracking of Sentinel-1 SAR data to investigate ephemeral grounding of a pinning point in the centre of Pine Island Glacier Ice Shelf between 2014 and 2023. The study attributes this ephemeral grounding primarily to basal melting, and suggests that ice shelf calving and atmospheric forcing may also play a secondary role.

**General Comments**

In this manuscript, the authors present a thorough and original analysis of the ephemeral grounding dynamics at Pine Island ice shelf. These observations make an important contribution by identifying the temporal evolution of this pinning point and exploring its drivers, in a region that plays an important role in the stability of the West Antarctic Ice Sheet. I commend the authors for this effort. However, in its current form, I suggest the manuscript requires a few substantive revisions before publication. These revisions mainly concern: (1) the strength of evidence supporting the key interpretations; (2) the clarity and accuracy of the text; (3) the presentation of figures and results. I present some general comments below, followed by more specific comments on the figures and text.

First, many of the interpretations and conclusions – particularly regarding the role of atmospheric forcing and the potential long-term evolution of the pinning point – are insufficiently supported by the presented data. In particular, the claim that the ephemeral pinning point may evolve into a final pinning point is presented as a definitive outcome without sufficient justification. If the authors wish to include this hypothesis, it should be clearly framed as speculative and appropriately qualified to reflect the current lack of direct evidence. Similarly, the evidence for atmospheric drivers (especially the links to the AAO and ONI variability) is not clearly demonstrated, and no direct basal melt observations are used to support the argument that the ephemeral grounding dynamics are driven by basal melt patterns.

Second, there are some factual inaccuracies and misleading statements that should be addressed. For example, the first sentence of the abstract is unclear. Pinning points usually occur over topographic highs, where the ice base remains locally grounded. An ephemeral pinning point refers to a temporary pinning point. Depending on the prior state of the ice shelf, pinning points can either form due to local thinning or thickening of the ice shelf. Prior to 2016, when the authors observe ephemeral grounding to start, there is no grounding in this area. Therefore in this case study, the ice shelf would have

had to experience local thickening for the ephemeral grounding to begin. This sentence should be changed to improve the clarity and convey the nuance within ephemeral grounding formation.

Third, the manuscript contains a number of grammatical errors and typos that must be corrected (e.g., full stop before the references in line 20 and the numbers 10 and 15 on lines 13 and 17 in the abstract). I encourage the authors to carefully proofread the manuscript to ensure all these errors are corrected. Some (but not all) are noted in the specific comments below.

I also recommend that the abstract should begin with a sentence emphasizing the broader significance of the study. For instance, why is it important to detect and monitor regions of ephemeral grounding? Or a more general motivation relating to ice shelf dynamics and their role in buttressing grounded ice could help frame the study more effectively.

Finally, throughout this paper, the authors haven't discussed the pinning point closer to the grounding line that ungrounded in 2011. Following that ungrounding, a thick column of ice was advected downstream through the ice shelf and is likely related to the ephemeral grounding of ice rumple L between 2015 and 2019. For further details of this process, see Joughin et al., 2016: https://doi.org/10.1002/2016GL070259 and Lowery et al., 2025: doi.org/10.5194/egusphere-2025-267.

**Methods**

Section 2.1:
- To reduce the noise of the Sentinel-1 data, the authors take a number of steps, including removing pixels depending on the azimuth displacement and range displacement. However, it is unclear how the authors selected these boundaries of acceptable data, and how much data this excluded. Providing more description of this would be useful for readers.

- $\theta_{inc}$ highly depends on the ice shelf surface slope. To calculate this, the authors have used the REMA 200m mosaic. However, the ice shelf slope on PIG is highly variable through time. The authors should comment on how calculating $\theta_{inc}$ against a time constant geometry impacts the results.

Section 2.2:

- Section 2.2, which describes the correction and co-registration of the REMA strips, is difficult to follow due to the disjointed structure of the text. The narrative jumps between different stages of the processing workflow, which limits the reader's ability to fully assess the methods used. To improve clarity, the authors should consider restructuring this section using an ordered, numbered list or a flow chart that clearly outlines each step in the process. Additionally, the authors should more clearly distinguish the correction vs co-registration steps, and clarify whether the final 'corrected' DSM is both corrected and co-registered. Despite the lack of clarity in this description, it is evident that the authors have carefully considered the dataset processing, and the results suggest that the method has been applied appropriately.

- Within this section, the authors rely heavily on Zinck et al. (2023), to the extent that readers would need to consult that paper in order to fully understand the methods applied here. While it is reasonable to reference prior work, key methodological details - particularly those directly affecting data processing - should be clearly stated within the manuscript. For example, lines 159–161 refer to 'criteria mentioned in Zinck et al. (2023a)' for coregistering REMA DSM strips with CryoSat-2 data, but these criteria are not explicitly outlined. Other examples of this are in lines 145, 157 and 179. Including a clear summary of the key elements would enhance the transparency and accessibility of the methodology.

- This section ends with Table 2 which shows the differences between the corrected and uncorrected REMA Strips and ICESat-2 elevation data. While the standard deviation has been significantly reduced as a result of the correction process, the mean difference has increased. It is also interesting to note that the resulting mean differences for each of the subdivisions are negative. Please could the authors comment on these differences including why there might be a bias in the correction processes they have used and the resulting impact on their results. For instance, could the negative bias stem from the differences in penetration depths between CryoSat-2 and ICESat-2?

**Results**

Section 3.1:

- This section would benefit from a more detailed comparison between the observed double-differential vertical displacement and the modelled tidal signal.

Specifically, it would be interesting to examine how the timing of the observed ephemeral grounding events relates to the *absolute* tide heights sampled at each Sentinel-1 acquisition. I would expect the ephemeral grounding to occur most frequently at low tide, so it would be interesting to see if this is the case. Note that this cannot be determined from the double-difference tide heights alone, as even a relatively large differential tide value could have missed sampling low absolute tides, and could potentially therefore miss observing an ephemeral grounding event. This effect may explain why ephemeral grounding events are not limited to the largest differential tide values shown in Fig. 4b. Including a plot or analysis that explicitly compares the timing of grounding events to absolute tide levels would substantially strengthen this section. It would also help validate the interpretation that tidal forcing is a driver of ephemeral grounding patterns.

- Line 205 states *'These coupled atmospheric and oceanic forcings modulate oceanic conditions (Huguenin et al., 2024) and, consequently, the ephemeral grounding behaviour'*. However, there is no evidence within this paper to justify the latter half of this statement and no reference has been provided. I am struggling to see any relationship between the ONI or AAO time series presented in Figure 4 and the timing of the ephemeral grounding events. If the authors wish to make this claim, additional analysis is needed to demonstrate a statistical correlation. Otherwise, I recommend removing or significantly softening this statement.

Section 3.2:

- The term *'changes in surface features'* should be clarified. Is this referring to the advection of features with ice flow, change of shape, or something else?

- This discussion would also benefit from the inclusion of elevation *change* measurements. For example, in lines 219-221, the authors claim the surface ridges are higher in 2015 than in 2019. This could readily be demonstrated (and made much more convincing)  by including a plot of change in elevation.

- In line 234 the authors state that thinner ice crossed over the seabed ridge. It would be useful to provide a quantitative estimate of this thinning magnitude (e.g. showing if it is beyond the uncertainty range of your measurements).

**Discussion**

Throughout the paper there appears to be some confusion or miscommunication regarding basal keels. On a large scale, the ice shelf has a central keel. However, this is overlaid with smaller scale basal channel-keel geometry. The large-scale shape of the ice shelf is largely determined by the shape of the bed upstream of the grounding line, as discussed by the authors in lines 262-264. However, there has been little discussion of the smaller scale channel-keel geometry. In line 259 the authors claim these 'basal channels and keels form at a similar location (Figure 4)', yet Figure 4 does not seem to provide evidence for this. However, some of the basal channel/keel structure does form as a result of the bed topography upstream (Lowery et al., 2025), but others form purely from melt-driven processes (Dutrieux et al., 2013). The discussion would benefit from a clearer distinction between these different scales of basal morphology and their respective formation mechanisms.

In line 282, the authors claim to be able to infer that basal melting was weak during the cold period from 2015 to 2020. However, this can not be inferred from the data presented in this manuscript. If the authors wish to make this claim, it should be verified using observational melt rate datasets, such as those from Paolo et al (2022), Davison et al (2023), or Adusumilli et al., (2020). Without such evidence, I recommend that this statement should be revised or removed.

Between line 285 and 291, the authors discuss the feedback between calving events and basal melt rates. However, the modelled results in Bradley et al., (2022) showed that the 2020 calving had no significant impact on basal melt rates. Within the model, the calving front needed to retreat inland of the seabed ridge before a significant (10%) increase in melt rates was seen in the inner cavity. I'm therefore not convinced this argument can be used to justify the ice shelf thinning between 2020 and 2021.

Additionally, I would suggest citing De Rydt et al. (2014) when discussing the impact of the height of the water column above the ridge and melt rates. (https://doi.org/10.1002/2013JC009513)

The authors claim that the changes in atmospheric forcing reduced melt rates which therefore caused the ice shelf to thicken between 2010 and 2015. In line 233, the authors say the ice elevation is 35m higher in 2015 than in 2010. This corresponds to a ~350m increase in ice thickness over 5 years, or a thickening rate of ~70m/year during this period. Is it reasonable to assume that this thickness increase has come from a ~70m/yr decrease in basal melt rates, equivalent to a >50% decrease in melt rates? As mentioned above, the authors should verify whether such a dramatic change is

supported by observational datasets. If not, the argument should be reconsidered or more cautiously framed.

**Conclusion**

In the conclusion, the authors claim that a La Niña event and a positive phase of AAO allowed thicker ice to form. However, during the period of thickening the ONI record indicates that a strong La Niña phase persisted for less than a year in total. It is unclear whether this limited duration provides sufficient evidence to support the claim that these climatic conditions were responsible for the observed ice thickening. The authors should either provide additional justification or revise the statement to reflect the limited strength of the evidence.

The authors also suggest that the ephemeral grounding site may evolve into a final pinning point. However, this claim is not supported by any evidence or modelling presented in this study, nor (to my knowledge) in any existing literature. While the authors may believe this to be the case, please make it clear that this is merely a conjecture and a modelling study would be needed to confirm this hypothesis.

**Figure 1**

- The current base map doesn't make the marine ridge clear, as suggested in the figure caption. I would suggest changing this.
- There is currently an overlap in colours in the calving front and grounding line colour maps which is confusing. I suggest using dashed lines for either the GL or calving front, or use two different colour maps that do not overlap.

**Figure 3**

- There is a missing label for feature 'L' on panel (d)
- The date labels on each panel are confusing, as you only include the first and last of the three dates. I suggest including all three labels on the panels (like you have in the caption) for clarity. The labels in the current form make it seem like the vertical displacement fields are constructed from just two image dates.

**Figure 4**

- In Panel (a) is there a reason why only periods with negative double-difference vertical displacement across the shelf are shown? Panel (b) suggests there are

many ephemeral grounding events in periods with a positive double-difference tide height. Perhaps it is really the absolute double-difference displacement/tide that is relevant, as surely the sign (+/-ve) just relates to the order in which you have applied the image differencing?
- The text labels in Panel (a) are too small.
- It would be useful to label the relative tide heights and the timing of the calving events on the maps in panel (a).
- Panel (b) is confusing; the caption says this includes '*Examples of 2D double-differential vertical displacement changes from November 2016 to May 2023*', yet there are data outside of this date range. The tidal heights before ~Aug 2016 and after ~Dec-2021 are also noticeably lower; presumably this is related to the period of Sentinel 1B being in operation, but should be explained. It would be helpful to point out the specific ungrounding events identified in Panel (a) to tie the figure together.
- The caption is missing dataset citations, including the CATS2008 tide model, ONI and AAO indices.

**Figure 5**

- This is a large figure that takes up lots of space, yet it is only referred to in one sentence in the manuscript. I suggest this is moved to the supplementary material, and instead you could add one or two of these Landsat images into Figure 6, which will also show the location of the surface ridge. Please also mark rumple L on this figure.

**Figure 6**
- I would also encourage the authors to include fewer panels in this figure; perhaps just those discussed in the text to show the main points (the reader doesn't gain much from the extra panels, and it's quite overwhelming).
- This figure also lacks a location map; perhaps using some of the space to have a Landsat image (such as currently in Fig 5) showing the location of this region would be useful.
- The inclusion of the 2011 grounding line on all panels that shows ice rumple L is grounded is confusing, as the key argument is that it is ungrounded at certain dates. Either labelling, or using a different weight/colour for the grounding line to show when the rumple is grounded or not between each panel would be helpful.
- Is elevation the most important thing to plot here? Maybe the authors should consider plotting thickness or ice draft instead

- As the authors discuss elevation change in the text, adding a panel to this figure with a change map would be helpful
- In this section, the authors also discuss how these ridges are higher or lower than 80m. It would be helpful if the 80m contour line was added to Figure 6.

**Figure 7**

- In all panels there are also too many lines overlapping to pick out the key message. Consider showing fewer dates and using a consistent colour scheme so that you only need one legend for all of the panels. There is also no need to duplicate date labels for the upper and lower panels as they show the same dates. In particular, the legend for panel (c) is overwhelming and detracts from the figure. Consider labelling this as 'difference from 2010/12/12', and then just labelling the dates that have been differenced from this.
- The BedMachine / BedMachine error lines are confusing. A solid black line and shaded area could be used to show the error (ensuring the shading is underneath the main data lines).
- Panel (c) should show thickness change not elevation change. This would then better tie in with the text in e.g. lines 252-5.

**Specific Comments**

Line 22 - specify whether by 'retreat' you mean grounding line or calving front retreat

Line 32 - typo in the citation, should be 'Miles' not 'Milles'

Line 34 - how confident are the authors that 'the direct buttressing effect of ephemeral grounding sites is minimal' ? Has this been modelled?

Line 47 - the phrase 'tidal fluctuations associated with ephemeral grounding' seems the wrong way round, as ephemeral grounding is caused by tidal fluctuations (not the other way around). Could be reworded to something like - *'particularly tidal fluctuations that drive ephemeral grounding patterns'*?

Line 47 - suggest adding 'can be observed *at grounding zones* using several satellite …'

Line 52 - The grounding line (and its importance for identifying pinning points/ephemeral grounding) should be introduced much earlier in the paper. Important to make this clear, otherwise this detailed paragraph on different GL measurement techniques feels out of place.

Line 59-60 - This sentence is currently unclear and requires grammatical edits. It is also somewhat misleading as it implies that Friedl et al. (2020) found that DROT GLs are *in general* 2 km seaward of DInSAR, whereas they only tested this at a single location (Petermann Glacier) and provide considerable discussion on the larger associated errors of DROT. DInSAR GL also in theory gives the tidal flexure limit. Consider re-writing this sentence to clarify these points.

Line 60 - This knowledge gap should be explained more clearly earlier - perhaps at the end of the paragraph at line 46.

Line 62 - Again, the information about the velocity increases should come earlier. This is the first time the reader is hearing about this yet it is a key motivation of this study

Line 63 - It would be useful to state here that it is because of these reasons stated above that you are therefore using DROT instead of DInSAR in your analysis. It would be useful to then quantify the impact this has on the accuracy of your results compared to using DInSAR.

Line 88  - the authors have used the acronym NCC without defining it.

Line 124 - be consistent in use of capitalisation of 'ice rumple L' / 'Ice rumple L' throughout. I would suggest no capitalisation is needed.

Line 134-5 - suggest using quotation marks for this acronym.

Line 141 - please clarify what is meant by '*we tested elevation differences in the DEM mosaic*'.

Line 143 - CATS2008 tide model is missing a citation here; should use Howard et al. (2019). Also please clarify the version (the most up-to-date is CATS2008_v2023).

Line 145 - it is unclear in this paragraph the source of the corrections and the dataset they are being applied to. Does *'data provided by the ESA'* mean that these are the corrections provided in the CryoSat-2 dataset, which have then been applied to the REMA DSM strips? This should be clarified, with appropriate citation given to the

original dataset. Note that the acronym ESA should be defined, and no 'the' is needed preceding 'ESA'.

Line 146-7 - This sentence is unclear. Is this referring to the CryoSat-2 data used for the co-registration? Please clarify.

Line 155 - please define ASAID, provide the appropriate citation (Bindschadler & Choi, 2011), and explain how it was used to define α (i.e. was the distance between the grounding line and hydrostatic point used to determine the width to apply a smooth transition from grounded to floating ice?). Also note that the ASAID grounding line marks the break-in-surface slope rather than the tidal flexure limit (as it was derived using optical imagery + altimetry), so this choice should be justified (as opposed to, say, the MEaSUREs DInSAR-derived grounding line product that marks the tidal flexure limit).

Line 156 - Please comment on how the choice to apply the tidal and IBE corrections based on the acquisition time of the first stereo image may or may not affect the results here.

Line 162 - What does 'CryoSat-2 distribution' mean here? Please clarify this sentence.

Line 167 - Please provide some more detail on the validation using ICESat-2.

Line 179 - More detail should be provided about the methods applied here, so that the reader does not have to rely on reading these two referenced papers. Specifying the equation(s) used here would be useful.

Line 182 - Define acronym

Line 184 - Please specify which Landsat satellite(s) were used, and provide appropriate citation.

Line 185 - Presumably the Landsat data is at a higher resolution (30m)? As currently worded it seems like both Landsat and MODIS are 250 m.

Line 187 - This sentence should include citations for the ONI and AAO data.

Line 190 - Please ensure that the precise acquisition time and coincident modelled tide heights for each Sentinel-1 acquisition are provided in the supplementary material.

Line 192 - Typo, should be 'matches'

Line 192-3 - The way this sentence is written makes it seem that Figure 4(b) shows that the displacement matches the double differential tidal height, which it does not. Perhaps somehow the double difference tide height values on the maps in panel (a) could be labelled to show this.

Line 195 - Typo, missing verb

Line 197 - Presuming that the '2011 grounding line' refers to the ASAID grounding line, Bindschadler & Choi (2011) should be cited here. This GL was not derived using the DInSAR method; it was derived from a combination of optical imagery and ICESat laser altimetry, marking the break-in-surface-slope. Additionally, looking at Figure 4 it is not clear that the GL is even marked on the maps in panel (a), so please address this.

Line 197-8 - Note that the Friedl et al. (2020) results are from one single study at Petermann Glacier. The way this is currently written implies that a more general/widespread relationship between DROT and DInSAR GLs has been shown in this review paper, which it does not. I suggest making this clearer in the text. Also (as commented above) both DROT and DInSAR GLs in theory indicate the landward limit of tidal flexure.

Line 198 - You should use 'grounding line' here (not 'grounding zone'). Be careful with use of line vs zone throughout.

Line 256 - Typo, should be 're-grounding'

Line 262 - I suggest re-wording to avoid the use of opposites in the sentence (e.g. 'low/high' and 'troughs/ridges').

Line 278 - Please define SAM acronym and provide a citation

Line 285 - This sentence suggests that the increase in PIG melt rates during this time were observed, when in fact this is results from a modelling study. This should be made clear in this sentence.

Line 325 - Typo, should be 'Grounding line products'. Also missing a reference for the ASAID grounding line product - Bindschadler & Choi (2011).

---

## Referee Comment (RC2)

Qian et al provide a detailed records of the ephemeral grounding events of a pining point on Pine Island Ice Shelf using Sentinel-1 SAR images and differential range offset tracking (DROT) method between 2014 and 2023. They claimed that ephemeral grounding is caused by basal melting, which is influenced by ice shelf calving and atmospheric forcing such as El Niño and La Niña. Although this is an interesting record, there is a lack of evidence to support the claims made by the authors, especially the results regarding basal melting are largely unconvincing and lacks care in overall logic and explanation. The claim in the abstract that the ephemeral grounding site may evolve into a final pining point and may influence future ice shelf is rather speculative and has not been discussed anywhere in the paper. Overall I think this study requires substantial revision and its current form is not suitable for publication in the Cryosphere.

**Major Comments**

The current results especially Figure 4 fail to establish the link between different tidal phases and intermittent grounding events of the Pine Island pinning point, because the double differential vertical displacement from DROT method and the double differential tidal heights do not provide information on tidal phases or tidal heights at specific timestamps. Therefore, it is not possible to identity whether a grounding event is associated with low tide or vice versa just based on these the current results.

Line 57-60: 'DROT-derived grounding line position were 2 km seaward of DInSAR and 2 km landward of H positions', is this 2 km bias applies to everywhere around Antarctica or just for specific locations? This statement also implies that DROT can only locate the middle location of the true grounding zone as a proxy grounding line, so if there is a 2 km bias, can we still trust DROT in identify the grounding line for Pine Island ice rumple? What if its maximum length is less than 4 km? There isn't any scale bar to measure this directly in the figures.

Section 3.1:

- Line 197-199 'Our results are consistent with Friedl et al (2020)….', first there is no scale bar in the Figure 4 making it impossible to measure the distance between DROT-derived GL and the 2011 DInSAR GL; second, the thinning and thickening of the ice shelf can cause the shrinking and growing of the pining point, this will change grounding line extent of the ice rumple, how to rule out this possibility?
- The authors claimed that pining point reappeared on 21 October 2021 in Line 199, but why this pining point locates upstream of the ice rumple L in Figure 4?
- The authors concluded that there are several possible factors causing this ephemeral grounding including atmospheric forcings such as La Nina and AO by including the analysis of ONI and AAO index. This conclusion is merely based on qualitative analysis just by roughly aligning the timeline of the empheral grounding with ONI and AAO phases, instead of providing a quantitative analysis or direct

observation or modelling results of basal melting in Pine Island Ice Shelf. This makes the conclusion rather speculative and not convincing.

Section 3.2

- The selected ice ridge is difficult to identify in Figure 5 given the small figure size, it is actually more visible from the elevation map in Figure 6, I suggest putting Figure 6 before Figure 5 to explain the choice of this ice ridge.
- Without labeling ice rumple L in Figure 5 and ice ridge in Figure 6, it is difficult to link the movement of ice ridge passing through the pining point, making it hard to follow the logic.

Discussion

- Line 249, the authors claimed that deep keels no long contacted the submarine ridge in 2020, this does not seem to be the case from Figure 7b where the orange line (2020) is in contact with the bedmachine bed topography.
- Line 259: can you label the basal channels in Figure 4?
- Line 263-264: yes the W shaped troughs can allow thicker ice to be advected downstream and form surface ridge, but this depends on the bed topography, and again does this matter to ice rumple L discussed in this paper?
- The discussions on possible factors influencing basal melt rates are mainly build on qualitative analysis of El Niño and La Niña events and iceberg calving events, without providing detailed records on ocean temperature and time evolving basal melt rates, making the arguments largely unconvincing.

**Specific Comments**

Abstract:

1) The entire abstract needs to be rewritten. The first sentence 'Ephemeral grounding sites form when ice shelves thin or relative sea level rises...' is difficult to understand, need rephrase. I suggest first explaining on what condition ice rumple can be formed – for example the bottom of the ice shelf gets grounded on a bathymetric high, then mentioning that the thinning of ice shelf or vertical movements of ice shelf caused by tides can cause 'ephemeral grounding'.
2) Please remove line numbers from the abstract

Line 31: 'Over time, some pinning points have disappeared entirely, particularly since 1973' this statement is unexpected here and doesn't seem to be linked to the following statement on ephemeral grounding, I suggest rephrasing or deleting it.

Line 32: should be (Miles and Bingham, 2024), not 'Milles'. Please change this reference throughout the manuscript.

Line 38: what do you mean by 'it is believed'? How reliable was this ungrounding event in 1973-1989?

Line 43: rephrase 'is now understood to result'

Lie 44-46: 'After four calving events....remain unclear' I am not sure why mentioning calving events here, why does calving have anything to do with ephemeral grounding and the changes of ice rumple L?

Line 88: what is NCC?

Line 168-175: this part reads repetitive and needs rewriting, no need to repeat everything listed in the table

Line 174: how is this 3 m uncertainty derived?

Figure 5:

1) Please label the location of ice rumple L in all subplots
2) Please provide a large zoomed-in map of the ice ridge, the current figure size makes it very difficult to distinguish this surface feature from neighboring surface undulations

---

## Author Comment (AC1)

**Response to the review of "Ephemeral grounding on the Pine Island Ice Shelf, West Antarctica, from 2014 to 2023," submitted to The Cryosphere.**

**Reviewer #2:**

We sincerely thank reviewer 2 for the thoughtful review and constructive feedback, which has led to a significant improvement of the manuscript. We fully agree with the reviewer on the major concerns. Accordingly, we made the following major changes to the manuscript:

1. We critically re-examined our interpretations considering the results, making significant changes to the presentation of figures and our analytical conclusions.
2. We extensively rewrote the manuscript to enhance clarity and readability throughout.
3. We improved the visualization of all figures, added, and removed figures to improve the presentation.

In this document, we provide a detailed, point-by-point response to all comments. For clarity, the reviewer's comments are shown in black, and our responses are shown in blue, with proposed new text in **bold.** Some sentences were underlined to emphasize key content closely related to the comments.

Qian et al provide a detailed records of the ephemeral grounding events of a pining point on Pine Island Ice Shelf using Sentinel-1 SAR images and differential range offset tracking (DROT) method between 2014 and 2023. They claimed that ephemeral grounding is caused by basal melting, which is influenced by ice shelf calving and atmospheric forcing such as El Niño and La Niña. Although this is an interesting record, there is a lack of evidence to support the claims made by the authors, especially the results regarding basal melting are largely unconvincing and lacks care in overall logic and explanation.

The claim in the abstract that the ephemeral grounding site may evolve into a final pining point and may influence future ice shelf is rather speculative and has not been discussed anywhere in the paper. Overall I think this study requires substantial revision and its current form is not suitable for publication in the Cryosphere.

**Response 1:**

We agree that the original discussion lacks evidence supporting the claims about atmospheric forcing and basal melting in the long-term evolution of the ephemeral pinning point. We've removed speculative statements and the hypothesis of atmospheric drivers, while rephrasing our assessment of ocean activities' impact in the abstract, discussion, and conclusion.

**Major Comments**

The current results especially Figure 4 fail to establish the link between different tidal phases and intermittent grounding events of the Pine Island pinning point, because the double differential vertical displacement from DROT method and the double differential tidal heights do not provide information on tidal phases or tidal heights at specific timestamps. Therefore, it is not possible to identity whether a grounding event is associated with low tide or vice versa just based on these the current results.

**Response 2:** We agree with the reviewer. We have revised Figure 4 and the corresponding description in Section 3 to include a tidal height time series from the CATS2008_v2023 tidal model, overlaid with the acquisition times of the SAR image pairs used in the DROT analysis. This revision enables a direct comparison between tidal height and the observed differential vertical displacements. We have also updated the figure caption and the main text in Section 3.1 to more clearly explain how grounding status is inferred based on local tidal heights at specific timestamps.

- **The revised Section 3.1 is as follows:**

"Ephemeral grounding regions, characterized by double-differential vertical displacements close to zero, are influenced by oceanic tidal variations (Figures 6-7 and Movie S1). The tidal height difference was calculated from data extracted at a point near the ice rumple L (longitude 100.6149°W, latitude 75.1867°S), corresponding to the exact acquisition times of each Sentinel-1 image, which were at 4:35 AM on each date (Supplement Table S1). One or two near-zero vertical displacement signals were detected at ice rumple L from at least November 2016 through April 2020, followed by a reappearance in December 2020. These signals are highlighted by yellow arrows in Figure 6a and marked by red vertical lines in Figure 6b. The reduced number of signals before ~August 2016 and after ~December 2021 likely reflects data limitations during periods when Sentinel-1B was not operational. Near-zero vertical displacement signals also occurred in 2016, 2017, and after the 2018 calving event. In December 2020, a similar signal appeared upstream of ice rumple L and progressively migrated toward the rumple, indicating that ephemeral grounding occurred as a thicker section of the ice shelf moved across the southern side of the sea ridge."

[Figure]

**Figure 6. Two-dimensional double-differential vertical displacement changes and time series of double-differential tidal height differences. (a) Spatial distribution of double-differential vertical displacement changes between November 2016 and May 2023. Yellow arrows highlight inferred ephemeral grounding signals in each displacement map. The tidal height difference (Tdiff) is labelled in each frame. (b) Time series of double-differential tidal height differences (black vertical lines) and inferred ephemeral grounding events (red vertical lines). Dashed lines indicate the timing of four major calving events: 13 July 2015, 12 October 2017, 31 October 2018, and 11 February 2020.**

"Figures 6a and 7a show that positive displacement anomalies are generally associated with negative tidal height differences, and vice versa—indicating a negative linear correlation between the two

variables. However, Figure 7b shows no clear relationship between tidal height and the area of the grounding region, suggesting that tidal forcing alone does not control the ephemeral grounding. In contrast, Figure 7c indicates that 64 ephemeral grounding events occurred between November 2016 and March 2021—35 during the neap tide period and 29 during the spring tide period. Notably, Figures 7c and 7d show that larger grounded areas are observed during spring tides, when tidal amplitudes are at their peak, while smaller grounded areas occur during neap tides, when tidal heights are at their lowest. These patterns suggest that the variation in grounded area is more closely linked to tidal period rather than tidal height alone. Together with Figure 6a, which shows the changes of the two near-zero vertical displacement signals, it suggests that thick ice advection from upstream may contribute to the grounding events. Consequently, ice dynamics likely play a significant role in the grounding process as well."

[Figure]

Figure 7. Comparison of tidal height differences with double-differential vertical displacement, comparison of tidal height differences and area of grounding region, including time series of area and tidal height variations. (a) Scatter plot of tidal height difference versus double-differential vertical displacement, showing a strong negative linear correlation between the two variables (Pearson's r = – 0.81, R² = 0.65, slope = –0.21). (b) Scatter plot of tidal height versus area of zero vertical displacement region, indicating no clear relationship between the two datasets. (c) Time series of changes in ice rumple area. (d) Time series of tidal height changes, where 0 represents mean sea level. Blue vertical lines indicate ephemeral grounding events during the neap tide period, while red vertical lines represent those during the spring tide period.

Line 57-60: 'DROT-derived grounding line position were 2 km seaward of DInSAR and 2 km landward of H positions', is this 2 km bias applies to everywhere around Antarctica or just for specific locations? This statement also implies that DROT can only locate the middle location of the true grounding zone as a proxy grounding line, so if there is a 2 km bias, can we still trust DROT in identify the grounding line for Pine Island ice rumple? What if its maximum length is less than 4 km? There isn't any scale bar to measure this directly in the figures.

**Response 3:** We thank the reviewer for this valuable suggestion. The ~2 km offset between DROT-derived grounding lines and those obtained from DInSAR and hydrostatic methods, as noted in Friedl et al. (2020), specifically pertains to Petermann Glacier. This offset should not be interpreted as a systematic bias applicable across Antarctica, as the width of the grounding zone and local ice shelf geometry vary substantially between regions.

In response to this concern, we have revised the relevant discussion of the DROT method in the Introduction as follows:

**"…Both DROT and InSAR methods in theory indicate the landward limit of tidal flexure. While InSAR is widely used to map grounding line migration, its effectiveness is limited in fast-flowing areas due to phase aliasing unless very short repeat intervals are available. For instance, Milillo et al. (2017) used 1-day repeat COSMO-SkyMed data to track grounding line changes at PIIS.**

**In contrast, DROT provides a complementary approach that does not rely on phase information, making it useful for observing vertical tidal displacements on fast-moving ice shelves, despite being less precise than InSAR in some contexts (Marsh et al., 2013; Hogg, 2015; Joughin et al., 2016; Christianson et al., 2016; Friedl et al., 2020; Wallis et al., 2024; Lowery et al., 2025; Zhu et al., 2025). Using TerraSAR-X data, Joughin et al. (2016) identified a vertical displacement anomaly near ice rumple L and estimated the grounding line position to within ~1.5 km. At Petermann Glacier, Friedl et al. (2020) found DROT-derived flexure limits ~2 km seaward of DInSAR results. More recently, DROT applied to Sentinel-1 IW data has proven effective for studying grounding line and pinning point dynamics on the Antarctic Peninsula (Wallis et al., 2024), Amery Ice Shelf (Zhu et al., 2025), and PIIS (Lowery et al., 2025). However, Lowery et al. (2025) focused only on the year 2017, leaving later changes unresolved. Thus, the evolution of grounding behaviour at ice rumple L following four subsequent calving events—in 2015, 2017, 2018, and 2020—remains poorly constrained."**

To evaluate the accuracy of our results, we compared the grounding lines derived from DROT with those obtained using the DDInSAR method (Rignot et al., 2014; Mohajerani et al., 2021). In the central part of the ice shelf, both methods identify similar ephemeral grounding regions, although the area delineated by DDInSAR is smaller than that outlined by DROT (Figure R1). Despite this difference, both approaches consistently detect the presence of ephemeral grounding signals. In other regions, while the DROT-derived grounding line is generally positioned slightly farther seaward compared to the DDInSAR result, the majority of the DROT grounding area still lies within the grounding zone identified by DDInSAR. Based on this consistency, we consider our results to be reliable.

[Figure]

Figure R1. Comparison of grounding line positions derived from Sentinel-1 imagery using the DROT and DDInSAR methods in 2018.

Rignot, E., Mouginot J., Morlighem M., Seroussi H., and Scheuchl B. (2014), Widespread, rapid grounding line retreat of Pine Island, Thwaites, Smith, and Kohler glaciers, West Antarctica, from 1992 to 2011, Geophys. Res. Lett., 41, 3502–3509, doi:10.1002/2014GL060140.

Mohajerani, Y., Jeong, S., Scheuchl, B. et al. (2021), Automatic delineation of glacier grounding lines in differential interferometric synthetic-aperture radar data using deep learning. Sci Rep 11, 4992, https://doi.org/10.1038/s41598-021-84309-3

**Section 3.1:**

• **Line 197-199** 'Our results are consistent with Friedl et al (2020)....', first there is no scale bar in the Figure 4 making it impossible to measure the distance between DROT-derived GL and the 2011 DInSAR GL; second, the thinning and thickening of the ice shelf can cause the shrinking and growing of the pining point, this will change grounding line extent of the ice rumple, how to rule out this possibility?

**Response 4:** We appreciate this observation and have revised Figure 4 and the associated text accordingly (see Response 2). Yellow arrows have been added to highlight the ephemeral grounding signals in Figure 4. We have also revised Figures 5 and 6 from the original submission and combined them into Figure 8 in the revised manuscript. A scale bar has been added to the updated Figure 8 to enable measurement of the distance between the DROT-derived grounding line and the 2011 DInSAR grounding line.

We agree with the reviewer that changes in ice shelf thickness can affect grounding at pinning points and may help explain the observed variability. Previous studies have suggested that ephemeral grounding may reflect temporal variations in ice shelf thickness (Rignot, 2002; Schmeltz et al., 2001). In our study, we not only examined the timing of grounding events but also analyzed concurrent changes in ice thickness. To achieve this, we used ICESat-2 data, which offer higher temporal resolution than REMA DEM strips, to investigate whether ephemeral grounding occurred between 2019 and 2022.

• **The revised discussion about the ice thickness changes in Section 3.2 is as follow:**

**3.2 Changes in surface ridges and ice thickness**

**Figure 8 shows the evolution of surface ridges and their elevations from December 2010 to January 2021. Near ice rumple L (red point in Figure 8), surface elevations remained around ~65 m between 2012–2017 and again during 2019–2020 (Figures 8d–h and 8j–k). The highest elevation (~85 m) was recorded in 2018, while the lowest (~54 m) occurred in 2021. Between 2020 and 2021, surface elevation declined by ~10 m, equivalent to ~70 m of ice-equivalent freeboard thickness. The area enclosed by the grounding line, corresponding to the region of zero vertical displacement, was the largest in 2018 (Figure 8i).**

[Figure]

**Figure 8. Changes in surface ridges at PIIS near ice rumple L. (a) Overview map showing the subregion outlined by the red frame, corresponding to panels (b)–(l). (b)–(l) Surface ridges and their elevation changes from 2010 to 2021, derived from corrected REMA DEMs. The two black circles indicate the positions of ice rumples. Grounding lines are delineated based on the zero-contour of the double-differential vertical displacement. Grey lines are the 80m contour line.**

**Profiles of ice-equivalent freeboard thickness derived from ICESat-2 (Figure 9) link surface elevation and grounding changes. Figure 7a shows mean thickness trends around the rumple along ICESat-2 tracks 965 and 1094 between 75.15°S and 75.05°S (Fig. 1b). Track 965 reveals increasing ice thickness from 2015 to 2021, while track 1094 shows a decrease from 2015 to 2017, a rebound in 2018, and a decline after 2020. Bottom elevation profiles derived from ICESat-2 (Figures 9b-e) further reveal changes in grounding status. The ice shelf was ungrounded on 27 August 2020, 5 March 2021, and 25 May 2022, but showed weak grounding on 6 June 2020. By integrating double-differential vertical displacement data with bottom elevation profiles, we find that ephemeral grounding signatures disappeared after March 2020 and reappeared in November 2020.**

[Figure]

**Figure 9. Time series of mean ice-equivalent freeboard thickness and ice shelf bottom elevation profiles along ICESat-2 tracks 965 and 1094. (a) Time series of mean ice-equivalent freeboard thickness (2010–2022). (b)–(c) Ice shelf bottom elevation profiles along ICESat-2 tracks 965 (gt2l and gt2r) between February 2020 and May 2022. (d)–(e) Ice shelf bottom elevation profiles along ICESat-2 tracks 1094 (gt3l and gt3r) between June 2019 and June 2022. Bed elevations are from the BedMachine v3 dataset (Morlighem et al., 2020; Morlighem, 2022), converted from EIGEN-6C4 to the EGM2008 geoid to match the vertical datum of REMA DSM strips. The estimated vertical uncertainty is ±200 m (shown as a grey transparent box).**

Rignot, E.: Ice-shelf changes in pine island bay, Antarctica, 1947–2000, J. Glaciol., 48, 247–256, https://doi.org/10.3189/172756502781831386, 2002.

Schmeltz, M., Rignot, E., and MacAyeal, D. R.: Ephemeral grounding as a signal of ice-shelf change, J. Glaciol., 47, 71–77, https://doi.org/10.3189/172756501781832502, 2001.

• The authors claimed that pining point reappeared on 21 October 2021 in Line 199, but why this pining point locates upstream of the ice rumple L in Figure 4?

**Response 5:** We apologize for the lack of clarity in the originally submitted manuscript. The feature identified as a "reappeared" pinning point on 21 October 2021 is located slightly upstream of the previously grounded area, within a region where grounding was also observed in 2016 (Figure 4 a1), 2017 (Figure 4 a2), and multiple times in 2018 (Figure 4 a6–a9, a17).

- **We have revised the Section 3.1 to:**

"…One or two near-zero vertical displacement signals were detected at ice rumple L from at least November 2016 through April 2020, followed by a reappearance in December 2020. These signals are highlighted by yellow arrows in Figure 6a and marked by red vertical lines in Figure 6b. The reduced number of signals before ~August 2016 and after ~December 2021 likely reflects data limitations during periods when Sentinel-1B was not operational. Near-zero vertical displacement signals also occurred in 2016, 2017, and after the 2018 calving event. In December 2020, a similar signal appeared upstream of ice rumple L and progressively migrated toward the rumple, indicating that ephemeral grounding occurred as a thicker section of the ice shelf moved across the southern side of the sea ridge."

The authors concluded that there are several possible factors causing this ephemeral grounding including atmospheric forcings such as La Nina and AO by including the analysis of ONI and AAO index. This conclusion is merely based on qualitative analysis just by roughly aligning the timeline of the ephemeral grounding with ONI and AAO phases, instead of providing a quantitative analysis or direct observation or modelling results of basal melting in Pine Island Ice Shelf. This makes the conclusion rather speculative and not convincing.

**Response 6:** We agree that our original conclusion overstated the influence of this climatic factor. We have removed the sections related to atmospheric forcing and discuss the part related to basal melting carefully in Section 4 and Appendix A.

- **The revised content in Section 4 is as follows:**

**Section 4 (last paragraph):**

"…**De Rydt et al. (2014) demonstrated that both the height of the ridge and the gap between the ridge and the ice shelf strongly influence the inflow of warm bottom waters into the cavity, and consequently, the melt rate. The melt rate may influence the ice thickness near to the grounding line upstream of the ice rumples K and L. This process may have contributed to ice thickness changes upstream and indirectly influenced the disappearance of ephemeral grounding signals following the 2020 calving event. We have added further analysis on the basal melt rate and ocean temperature in the Appendix A. Although smaller-scale basal channels and keel geometries are primarily shaped by melt-driven processes (Bindschadler et al., 2011b; Dutrieux et al., 2013; Stanton et al., 2013; Dutrieux et al., 2014b; Joughin et al., 2016), the lack of direct, high-temporal-resolution basal melt rate measurements after 2020 limits our ability to capture short-lived grounding events and confirm the role of ocean-driven melting. Future work should prioritize the integration of dense time series from new SAR missions and in situ oceanic data to better resolve ephemeral grounding behaviour and its implications for ice shelf evolution and calving dynamics in a warming climate.**"

**Appendix A. Oceanic condition changes and analysis**

[revised manuscript text omitted]

**Section 3.2**

• The selected ice ridge is difficult to identify in Figure 5 given the small figure size, it is actually more visible from the elevation map in Figure 6, I suggest putting Figure 6 before Figure 5 to explain the choice of this ice ridge.

**Response 7:** We agree that the ice ridge is more clearly visible in the elevation map shown in Figure 6. We have reorganized Figure 5 and Figure 6 by combining the MODIS imagery with the surface elevation data (see revised Figure 8 in response 2). The ice rumples detected each year are also plotted to highlight changes in the ephemeral grounding locations. Additionally, we have added the 80m contour line and labeled the elevation values near ice rumple L to facilitate identification of elevation changes.

• Without labeling ice rumple L in Figure 5 and ice ridge in Figure 6, it is difficult to link the movement of ice ridge passing through the pining point, making it hard to follow the logic.

**Response 8:** We appreciate this suggestion and have figures to include clear labels marking the location of ice rumple L in each panel (see revised Figure 6 in response 2 and revised Figure 8 in response 2 and 7).

**Discussion**

• **Line 249,** the authors claimed that deep keels no long contacted the submarine ridge in 2020, this does not seem to be the case from Figure 7b where the orange line (2020) is in contact with the bedmachine bed topography.

**Response 9:** We thank the reviewer for this careful observation. Upon closer examination of the double-differential vertical displacement and ICESat-2 ice shelf bottom elevation profiles, we found ephemeral grounding signals both before April 2020 and after December 2020. We have corrected it in the manuscript as follows:

**Discussion (paragraph 3)**

**"Changes in surface elevation and ice-equivalent freeboard thickness indicate that the ice shelf underwent thickening prior to grounding events and thinning prior to ungrounding. Notably, the surface elevation peaked in 2018 and declined significantly between 2020 and 2021, coinciding with changes in grounding behavior. Near-zero vertical displacement signals—indicative of ephemeral grounding—were detected at ice rumple L from at least November 2016 through April 2020, then disappeared during the 2020–2021 thinning period before reappearing in December 2020. In that instance, a similar signal emerged upstream of the rumple and gradually migrated toward it, suggesting that a thicker section of the ice shelf had moved over the sea ridge, re-establishing ephemeral contact with the bed."**

- **Line 259:** can you label the basal channels in Figure 4?

**Response 10:** Thank you for the suggestion. We have reviewed Figure 4 and considered labeling the basal channels. However, to maintain clarity and avoid overcrowding, we did not add explicit labels for basal channels in revised Figure 8 (see Response 2 and 7). Instead, we enhanced the figure with combined surface elevation data and MODIS imagery in revised Figure 8, where the surface ridges and troughs are more clearly visible.

- **Line 263-264:** yes, the W shaped troughs can allow thicker ice to be advected downstream and form surface ridge, but this depends on the bed topography, and again does this matter to ice rumple L discussed in this paper?

**Response 11:** The W-shaped troughs do play a role in the formation and maintenance of ice rumple L by allowing thicker ice to be advected downstream, contributing to the development of surface ridges in that area. We agree with Reviewer 1 that distinguishing between the formation processes of large-scale and small-scale keels helps clarify the discussion. Accordingly, we have revised the text in Section 4 to explicitly link the general mechanism of W-shaped troughs to the specific case of ice rumple L, making this connection clearer for readers.

- **The revised Section 4 in the manuscript is as follows:**

**"Changes in surface elevation and ice-equivalent freeboard thickness indicate that the ice shelf underwent thickening prior to grounding events and thinning prior to ungrounding. Notably, the surface elevation peaked in 2018 and declined significantly between 2020 and 2021, coinciding with changes in grounding behavior. Near-zero vertical displacement signals—indicative of ephemeral grounding—were detected at ice rumple L from at least November 2016 through April 2020, then disappeared during the 2020–2021 thinning period before reappearing in December 2020. In that instance, a similar signal emerged upstream of the rumple and gradually migrated toward it, suggesting that a thicker section of the ice shelf had moved over the sea ridge, re-establishing ephemeral contact with the bed.**

**Evidence of corrugations with periodic spacing on the submarine ridge supports the idea that this seafloor landscape was formed by sub-ice-shelf keels modulated by tidal motion (Graham et al., 2013; Davies et al., 2017). Surface elevation data (Figure 8(b)-(l)) show that the center of the ice shelf is higher**

**than the flanks, suggesting the presence of a central basal keel beneath the shelf. This large-scale geometry is primarily shaped by bed topography upstream of the grounding line. Specifically, the bed exhibits a W-shaped profile, with two troughs flanking a central topographic high (Lowery et al., 2025). These troughs may channel thicker ice toward the downstream ridge, promoting the formation of surface ridges aligned with basal keels. As ice flows downstream from the grounding line, local surface elevation adjusts toward hydrostatic equilibrium (Shean, 2016). The consistent positioning of basal channels and keels reflects this inherited bed structure. Therefore, the large-scale development of basal keels and channels is strongly controlled by upstream bed topography (Lowery et al., 2025)."**

• The discussions on possible factors influencing basal melt rates are mainly build on qualitative analysis of El Niño and La Niña events and iceberg calving events, without providing detailed records on ocean temperature and time evolving basal melt rates, making the arguments largely unconvincing.

**Response 12:** We thank the reviewer for this insightful comment. We agree that our initial discussion relied too heavily on qualitative correlations with ENSO and calving events, which limited the strength of our arguments. We have removed those interpretations.

**Specific Comments**

**Abstract:**

1) The entire abstract needs to be rewritten. The first sentence 'Ephemeral grounding sites form when ice shelves thin or relative sea level rises...' is difficult to understand, need rephrase. I suggest first explaining on what condition ice rumple can be formed – for example the bottom of the ice shelf gets grounded on a bathymetric high, then mentioning that the thinning of ice shelf or vertical movements of ice shelf caused by tides can cause 'ephemeral grounding'.

**Response 13:**

We agree with the reviewer. We have re-written the abstract and carefully revised relevant statements throughout the manuscript.

• **The revised abstract was as follows:**

"…**Recurring ice keels beneath the ice shelf cause ephemeral grounding events that remain poorly understood but may significantly influence ice shelf stress fields and flow dynamics.**"

2) Please remove line numbers from the abstract

**Response 14:** Done.

**Line 31:** 'Over time, some pinning points have disappeared entirely, particularly since 1973' this statement is unexpected here and doesn't seem to be linked to the following statement on ephemeral grounding, I suggest rephrasing or deleting it.

**Response 15:** Deleted.

**Line 32:** should be (Miles and Bingham, 2024), not 'Milles'. Please change this reference throughout the manuscript.

**Response 16:** We corrected this error throughout the manuscript.

**Line 38:** what do you mean by 'it is believed'? How reliable was this ungrounding event in 1973-1989?

**Response 17:** Thank you for requesting clarification. The ungrounding event during 1973–1989, reported by Miles and Bingham (2024), is supported by indirect evidence such as changes in ice flow speed and satellite imagery. However, due to limited data availability, the exact timing and location remain uncertain. The phrase "it is believed" reflects this uncertainty and the reliance on the best available, albeit indirect, observations. We have revised the sentence to:

**"This oceanic forcing initially caused transient grounding of the central ice shelf on a submarine ridge from the 1940s through the 1970s, followed by complete ungrounding between 1973 and 1989 (Jenkins et al., 2010; Smith et al., 2017; Miles and Bingham, 2024)."**

**Line 43:** rephrase 'is now understood to result'

**Response 18:** The revised sentence is as follows:
**"Despite the grounding line retreat, the Pine Island Ice Shelf (PIIS) was observed to maintain intermittent contact with the bathymetric high when thick ice column being advected from the upstream deep trough (Joughin et al., 2016; Lowery et al., 2025). This ephemeral grounding is now attributed to interactions between sub-ice keels and a submarine ridge (Graham et al., 2013; Joughin et al., 2016; Shean, 2016; Davies et al., 2017)."**

**Lie 44-46:** 'After four calving events...remain unclear' I am not sure why mentioning calving events here, why does calving have anything to do with ephemeral grounding and the changes of ice rumple L?

**Response 19:** We thank the reviewer for this insightful comment. We believe the ephemeral grounding could have facilitated rift propagation, and the development of calving events in the past. As such, we have revised the abstract and the introduction to emphasize this point and have added further analysis on the rift propagation process in Sections 2.4, 3.3, and 4.

- **The revised sections are as follow:**

**Abstract**

**"The evolution of ephemeral grounding sites offers valuable insights into changes in ice shelf thickness, which can affect buttressing, alter ice flow dynamics, and influence ice shelf stability. Long-term observations of these sites are crucial for understanding how thickness, basal conditions, and tidal interactions evolve over time."**

**"Landsat-8 images reveal that the rifts that cause the 2020 calving event formed after the region passed through the ephemeral grounded area suggesting that ephemeral grounding events may contribute to the formation of rifts. These findings provide new insights into the mechanisms driving ephemeral grounding behaviour and highlight its potential role in modulating ice shelf stability."**

**1. Introduction (paragraphs 4)**

**"… The grounding of ice shelf on high bathymetry features could impact ice dynamics as an obstacle against ice flow: 1) enhance the buttressing effect by providing back stress against upstream ice; 2) facilitate fracturing and ice shelf weakening in response to stress associated with grounding (Rignot, 2002; Christianson et al., 2016; Jeong et al., 2016; Shean, 2016; Benn et al., 2022; Wang et al., 2025)."**

**1. Introduction (paragraphs 6)**

**"…More recently, DROT applied to Sentinel-1 IW data has proven effective for studying grounding line and pinning point dynamics on the Antarctic Peninsula (Wallis et al., 2024), Amery Ice Shelf (Zhu**

et al., 2025), and PIIS (Lowery et al., 2025). However, Lowery et al. (2025) focused only on the year 2017, leaving later changes unresolved. Thus, the evolution of grounding behaviour at ice rumple L following four subsequent calving events—in 2015, 2017, 2018, and 2020—remains poorly constrained."

**2.4 Rift propagation observation**

Previous studies have suggested that such grounding may be linked to the formation of transverse rifts south of ice rumple L (Joughin et al., 2021), potentially contributing to calving events between 2015 and 2020. However, limitations in the spatial resolution and clarity of SAR imagery hinder a definitive assessment of the connection between ephemeral grounding and rift formation. We used Landsat-8 optical images, specifically the panchromatic band with a 15m spatial resolution, to track the rift propagation history. We then compared these results with our grounding line data to better understand the interaction between ephemeral grounding and rift propagation.

**3.3 Rift propagation observation**

Using Landsat images, we tracked the propagation history of the rifts from 2013 to 2019 (Figure 10). Rift R1 first appeared in the image from December 15, 2017 (Figure 10d), after the region passed through the ephemeral grounding zone, as seen in Figure 10c. Similarly, Rift R2 appeared in the December 11, 2018 image (Figure 10f), following its passage through the same grounding region. These two rifts ultimately led to the 2020 calving event. Therefore, our results suggest that ephemeral grounding events are linked to rift propagation, indirectly influencing the ice shelf calving process.

[Figure]

Grounding line
— 20161104_20161110_20161116
— 20170121_20170127_20170202      20180209_20180215_20180221      — 20180404_20180410_20180416
— 20170912_20170918_20170924      — 20180215_20180221_20180227    — 20181212_20181218_20181224
— 20170918_20170924_20170930      — 20180323_20180329_20180404    — 20181224_20181230_20190106

**Figure 10. Rift propagation history from 2013 to 2019. (a)-(h) show the propagation history of the rifts R1 (red arrow) and R2 (blue arrow), which led to the 2020 calving event. The black circles indicate the positions of ice rumple K. Grounding lines are delineated based on the near-zero value of the double-differential vertical displacement.**

**4 Discussion (paragraph 5)**

**"In summary, our study demonstrates that ephemeral grounding at ice rumple L is modulated by the interaction between tidal forcing, ice shelf thickness, and evolving sub-ice geometry. These results provide new insights into the mechanisms driving ephemeral grounding behaviour. Notably, we find that the rift that caused the 2020 calving event appeared after pass through the ephemeral grounding region. Though we did not have enough images that could capture the whole process in 2017 and 2018, our results indicate that the ephemeral grounding events could cause the formation of the rifts. Arndt et al. (2018) further emphasized the importance of final pinning points in controlling calving line orientation, raising the possibility that ice rumple L may have acted as a final pinning point after the 2015 calving event, thereby influencing rift propagation and subsequent calving. These findings underscore the need for high-resolution ice shelf modelling to evaluate how ephemeral grounding affects stress redistribution and overall ice shelf stability."**

**5 Conclusion (paragraph 2 and 3)**

**"… We also show that the rifts responsible for the 2020 calving event appeared after the region passed through the ephemeral grounded area, suggesting that these ephemeral grounding events may have contributed to the formation of the rifts.**

**Our findings highlight the critical role of combining remote sensing and in situ ocean measurements to monitor grounding processes. The grounding lines derived from our DROT results provide valuable input for modelling work. We also underscore the need for high-resolution ice shelf modelling to assess how ephemeral grounding influences stress redistribution, calving dynamics, and the long-term stability of vulnerable ice shelves. In future, improved satellite coverage, denser SAR time series, and in situ ocean measurements will be essential to constrain short-lived grounding behaviours and their response to a changing climate."**

**Line 88:** what is NCC?

**Response 20:** NCC is "normal cross-correlation coefficient". Th revised sentence is as follows:

**"First, we retained only pixels with a normalized cross-correlation value greater than 0.05, which also used by Solgaard et al. (2021) to ensure reliable displacement measurements."**

**Line 168-175:** this part reads repetitive and needs rewriting, no need to repeat everything listed in the table

**Response 21:** Thank you for this helpful suggestion. We have revised the section to provide a more concise summary of the key observations, eliminating repetitive details already covered in the table. The updated content is as follows:

**"As shown in Table 2, the corrected REMA strips exhibited lower standard deviations compared to the uncorrected data, indicating reduced uncertainty. However, a consistent negative mean bias remained, with the corrected REMA elevations appearing systematically lower than those from ICESat-2.**

**This bias likely results from the differing measurement principles of the two satellite systems: CryoSat-2 (used for REMA correction) operates in the Ku-band and can penetrate the upper snowpack, whereas**

**ICESat-2 uses green laser altimetry, which reflects off the snow surface. As a result, CryoSat-2—and by extension, the corrected REMA strips—tend to report slightly lower surface elevations than ICESat-2, especially over snow-covered areas. Additional factors such as residual temporal offsets, snow accumulation variability, and surface roughness may also contribute. Based on this comparison, we estimate the uncertainty of the corrected REMA strips as −1.93 ± 2.54 m, equivalent to 15.44 ± 20.32 m in floating ice thickness."**

**Line 174:** how is this 3 m uncertainty derived?

**Response 22:** The ±3 m vertical displacement uncertainty was initially estimated from the mean difference between ICESat-2 elevations and the corrected REMA DSM. However, we have now revised this estimate to −1.93 ± 2.54 m, based on the total difference between the two datasets (see Response 21).

**Figure 5:**

1) Please label the location of ice rumple L in all subplots

**Response 23:** Revised (see response 2, 4, and 7).

2) Please provide a large zoomed-in map of the ice ridge, the current figure size makes it very difficult to distinguish this surface feature from neighboring surface undulations

**Response 24:** The zoom-in map is increased in revised Figure 8 (see Response 4 and 7).

---

## Author Comment (AC2)

**Response to the review of "Ephemeral grounding on the Pine Island Ice Shelf, West Antarctica, from 2014 to 2023," submitted to The Cryosphere.**

**Reviewer #1:**

We sincerely thank reviewer 1 for the thoughtful review and constructive feedback, which has led to a significant improvement of the manuscript. We fully agree with the reviewer on the major concerns. Accordingly, we made the following major changes to the manuscript:

1. We critically re-examined our interpretations considering the results, making significant changes to the presentation of figures and our analytical conclusions.
2. We extensively rewrote the manuscript to enhance clarity and readability throughout.
3. We improved the visualization of all figures, added, and removed figures to improve the presentation.
4. For clarity, we quoted the original sentence in each specific comment in grey to facilitate easier comparison.

In this document, we provide a detailed, point-by-point response to all comments. The reviewer's comments are shown in black, and our responses are shown in blue, with proposed new text in **bold.** Some sentences were underlined to emphasize key content closely related to the comments.

**Summary:**

This study uses differential range offset tracking of Sentinel-1 SAR data to investigate ephemeral grounding of a pinning point in the centre of Pine Island Glacier Ice Shelf between 2014 and 2023. The study attributes this ephemeral grounding primarily to basal melting, and suggests that ice shelf calving and atmospheric forcing may also play a secondary role.

**General Comments**

In this manuscript, the authors present a thorough and original analysis of the ephemeral grounding dynamics at Pine Island ice shelf. These observations make an important contribution by identifying the temporal evolution of this pinning point and exploring its drivers, in a region that plays an important role in the stability of the West Antarctic Ice Sheet. I commend the authors for this effort. However, in its current form, I suggest the manuscript requires a few substantive revisions before publication. These revisions mainly concern: (1) the strength of evidence supporting the key interpretations; (2) the clarity and accuracy of the text; (3) the presentation of figures and results. I present some general comments below, followed by more specific comments on the figures and text.

We greatly appreciate your recognition of the scientific significance of our work.

First, many of the interpretations and conclusions – particularly regarding the role of atmospheric forcing and the potential long-term evolution of the pinning point – are insufficiently supported by the presented data. In particular, the claim that the ephemeral pinning point may evolve into a final pinning point is presented as a definitive outcome without sufficient justification. If the authors wish to include this hypothesis, it should be clearly framed as speculative and appropriately qualified to reflect the current lack of direct evidence. Similarly, the evidence for atmospheric drivers (especially the links to the AAO and ONI variability) is not clearly demonstrated, and no direct basal melt observations are used to support the argument that the ephemeral grounding dynamics are driven by basal melt patterns.

**Response 1:** We agree that the original discussion overstated the certainty regarding the role of atmospheric forcing and the potential long-term evolution of the ephemeral pinning point. In response, we have removed

speculative statements and the hypothesis of atmospheric drivers, while rephrasing our assessment of ocean activities' impact. We have revised the corresponding content in the abstract, discussion, and conclusion accordingly.

Second, there are some factual inaccuracies and misleading statements that should be addressed. For example, the first sentence of the abstract is unclear. Pinning points usually occur over topographic highs, where the ice base remains locally grounded. An ephemeral pinning point refers to a temporary pinning point. Depending on the prior state of the ice shelf, pinning points can either form due to local thinning or thickening of the ice shelf. Prior to 2016, when the authors observe ephemeral grounding to start, there is no grounding in this area. Therefore, in this case study, the ice shelf would have had to experience local thickening for the ephemeral grounding to begin. This sentence should be changed to improve the clarity and convey the nuance within ephemeral grounding formation.

**Response 2:** We agree with the reviewer. We have re-written the abstract and carefully revised relevant statements throughout the manuscript.

- **The revised abstract is as follows:**

"**The evolution of ephemeral grounding sites offers valuable insights into changes in ice shelf thickness, which can affect buttressing, alter ice flow dynamics, and influence ice shelf stability. Long-term observations of these sites are crucial for understanding how thickness, basal conditions, and tidal interactions evolve over time.** Vertical displacement data derived from Sentinel-1A/B imagery reveals the history of ephemeral grounding events at Pine Island Ice Shelf (PIIS) from 2014 to 2023. The grounding line positions derived from vertical displacement data provide critical constraints for ice sheet modelling applications. Our results suggest that ephemeral grounding at ice rumple L is modulated by the interaction between tidal forcing, ice shelf thickness, and evolving sub-ice-shelf geometry. A prominent central keel, shaped by inherited bed topography, promotes repeated contact with a submarine ridge. **Although smaller-scale keels may be governed by basal melting processes influenced by ocean temperature variability, the lack of direct, high-temporal-resolution basal melt rate measurements after 2020 limits our ability to capture short-lived grounding events and confirm the role of ocean-driven melting. Landsat-8 images reveal that the rifts that cause the 2020 calving event formed after the region passed through the ice rumple L suggesting that ephemeral grounding events may contribute to the formation of rifts.** These findings provide new insights into the mechanisms driving ephemeral grounding behaviour and highlight its potential role in modulating ice shelf stability.**"

Third, the manuscript contains a number of grammatical errors and typos that must be corrected (e.g., full stop before the references in line 20 and the numbers 10 and 15 on lines 13 and 17 in the abstract). I encourage the authors to carefully proofread the manuscript to ensure all these errors are corrected. Some (but not all) are noted in the specific comments below.

**Response 3:** We thank the reviewer for the detailed and careful review. We have carefully proofread the entire manuscript and corrected the errors.

I also recommend that the abstract should begin with a sentence emphasizing the broader significance of the study. For instance, why is it important to detect and monitor regions of ephemeral grounding? Or a more general motivation relating to ice shelf dynamics and their role in buttressing grounded ice could help frame the study more effectively.

**Response 4:** We agree. The abstract was re-written to highlight the significance of this study.

Finally, throughout this paper, the authors haven't discussed the pinning point closer to the grounding line that ungrounded in 2011. Following that ungrounding, a thick column of ice was advected downstream through the ice shelf and is likely related to the ephemeral grounding of ice rumple L between 2015 and 2019. For further details of this process, see Joughin et al., 2016: https://doi.org/10.1002/2016GL070259 and Lowery et al., 2025: doi.org/10.5194/egusphere-2025-267.

**Response 5:** We appreciate the reviewer's insightful observation and relevant literature. This perspective provides valuable context for interpreting the observed changes in ice shelf thickness and grounding behavior.

In response, we have added a description of the upstream ungrounding event in paragraph 4 of Section 1 in the revised manuscript. This includes a discussion of its potential role in advecting a thicker ice column downstream, which may have contributed to the transient grounding observed at rumple L. Recommended previous works of Joughin et al. (2016) and Lowery et al. (2025) are cited and discussed in the introduction and discussion sections.

- **The revised paragraph in the introduction Section is as follows:**

"……Continued ice shelf thinning subsequently drove an ~8.7 km retreat of the grounding line along the main trunk between 1992 and 2009, resulting in further ungrounding from an ice plain (Joughin et al., 2010; Dutrieux et al., 2014a; Rignot et al., 2014). **Despite the grounding line retreat, the Pine Island Ice Shelf (PIIS) was observed to maintain intermittent contact with the bathymetric high when thick ice column being advected from the upstream deep trough (Joughin et al., 2016; Lowery et al., 2025). This ephemeral grounding is now attributed to interactions between sub-ice keels and a submarine ridge (Graham et al., 2013; Joughin et al., 2016; Shean, 2016; Davies et al., 2017)."**

**Methods**

**Section 2.1:**

   - To reduce the noise of the Sentinel-1 data, the authors take a number of steps, including removing pixels depending on the azimuth displacement and range displacement. However, it is unclear how the authors selected these boundaries of acceptable data, and how much data this excluded. Providing more description of this would be useful for readers.

**Response 6:** A more detailed explanation of the selection of boundaries for acceptable azimuth and range displacements, as well as the amount of data excluded by these thresholds are provided in the revised manuscript.

- **The following paragraph and a flow chart are added to the Method section:**

"**To reduce noise and remove outliers in the Sentinel-1 offset tracking data, we employed a multi-step filtering and calibration approach. First, we retained pixels only with** **a normalized cross-correlation value greater than 0.05, which also used by Solgaard et al. (2021) to ensure reliable displacement measurements.** **Second, we extracted the residual displacement at 458 widely distributed, non-moving points over the exposed bedrock (Figure 1(a)).** **Outliers beyond one standard deviation were removed, following the same criteria that used in Chen et al. (2020), and the mean residual displacement for each time interval was calculated and used to calibrate the displacement maps by subtracting this mean value. To further remove noise and erroneous measurements, we examined the distribution of azimuth and range displacements across the entire time series (Figure S1) and established empirical thresholds based on reasonable minimum and maximum velocities of ice movement at PIIS.** **We invalidated pixels with slant range displacements less than -60 m or greater than 75 m for 6-day gaps, and less than -120 m or**

**greater than 150 m for 12-day gaps, which can exclude a small portion of pixels and improved the consistency and quality of the final displacement fields."**

[Figure]

**Figure 2.** Processing steps of range displacement generation and DROT.

- θ inc highly depends on the ice shelf surface slope. To calculate this, the authors have used the REMA 200m mosaic. However, the ice shelf slope on PIG is highly variable through time. The authors should comment on how calculating θ inc against a time constant geometry impacts the results.

**Response 7:** We agree that the surface slope of the Pine Island Ice Shelf varies over time, as may impact our surface elevation results. However, we are limited by the temporal resolution of available DEM products, which prevents the use of surface elevation data contemporaneous with the Sentinel-1 acquisitions.

We believe the impact of slope variability on our results is minimal for several reasons. First, we used the REMA 200 m mosaic DEM for both $\theta_{inc}$ estimation and as the external DEM for co-registration during offset tracking in GAMMA, ensuring a consistent reference across displacement fields. Second, while tidal forcing causes vertical motion, it has limited impact on the overall surface slope of the ice shelf. The influence of slope variability is primarily associated with localized topographic undulations. During ephemeral grounding events, the resulting range-direction displacement changes are significantly larger than those caused by a constant background slope, allowing such events to be reliably identified. Therefore, we are confident that the use of a time-invariant DEM does not compromise the integrity of our results.

- **We also recognize that our processing description was unclear and have revised the relevant text in the manuscript as follows:**

**"The REMA DEM was used consistently for both $\theta_{inc}$ (in radians) estimation and as the external DEM for co-registration in the offset tracking process, ensuring uniform referencing across displacement fields. The $\theta_{inc}$ was calculated for the first acquisition of each image pair. The local surface normal was derived from the REMA 200 m mosaic DEM. The vertical displacement caused by tidal forcing has minimal impact on the ice shelf's overall surface slope. While slope-induced errors are most significant in areas with localized topographic variability, ephemeral grounding events produce range-direction displacement anomalies that exceed those caused by background slope variations, making these events clearly distinguishable. Consequently, we are confident that using a time-invariant DEM does not compromise the accuracy of our results, as the impact of slope variability on $\theta_{inc}$ and the resulting displacement estimates remains minimal."**

**Section 2.2:**

- Section 2.2, which describes the correction and co-registration of the REMA strips, is difficult to follow due to the disjointed structure of the text. The narrative jumps between different stages of the processing workflow, which limits the reader's ability to fully assess the methods used. To improve clarity, the authors should consider restructuring this section using an ordered, numbered list or a flow chart that clearly outlines each step in the process. Additionally, the authors should more clearly distinguish the correction vs co-registration steps, and clarify whether the final 'corrected' DSM is both corrected and co-registered. Despite the lack of clarity in this description, it is evident that the authors have carefully considered the dataset processing, and the results suggest that the method has been applied appropriately.

**Response 8:** We thank the reviewer for this helpful comment. We have substantially revised Section 2.2 to present the processing workflow in a more logical and sequential format. Specifically, we now outline the procedures in a clearly ordered list, distinguishing each step in the correction and co-registration process. We also clarify that the final "corrected" DSM refers to a product that has undergone both vertical correction and horizontal co-registration. We also add a flow chart that clearly outlines each step in the process.

- **The revised flow chart and the part related to correction and co-registration of the REMA strips are as follows:**

**"Elevation data from the CryoSat-2 Baseline-D Level 2 SARIn product (Meloni et al., 2019), spanning from July 2010 to June 2022, were used to correct and co-register the Reference Elevation Model of Antarctica (REMA) 2 m spatial resolution time-stamped Digital Surface Model (DSM) Version 4.1 product, acquired between October 2010 and December 2022 (Howat et al., 2022b). These REMA strips are referenced to the WGS84 ellipsoid but are not co-registered to satellite altimetry by default. The correction and co-registration procedures were implemented using the "Basal melt rates Using REMA and Google Earth Engine (BURGEE)" processing framework developed by Zinck et al. (2023a, 2023b). Processing steps are outlined in Figure 5."**

[Figure]

**Figure 5. Processing steps for corrections of REMA, CryoSat-2, and ICESat-2 data.**

- Within this section, the authors rely heavily on Zinck et al. (2023), to the extent that readers would need to consult that paper in order to fully understand the methods applied here. While it is reasonable to reference prior work, key methodological details - particularly those directly affecting data processing -should be clearly stated within the manuscript. For example, lines 159–161 refer to 'criteria mentioned in Zinck et al. (2023a)' for coregistering REMA DSM strips with CryoSat-2 data, but these criteria are not explicitly outlined. Other examples of this are in lines 145, 157 and 179. Including a clear summary of the key elements would enhance the transparency and accessibility of the methodology.

**Response 9:** We have revised Section 2.2 to include a more complete summary of the core methodological steps from Zinck et al. (2023a) making the manuscript self-contained for readers. Specifically, we now explicitly describe the criteria used for co-registering REMA strips to CryoSat-2 data, including thresholds for spatial overlap, filtering of outliers, and acceptance criteria for vertical shifts and standard deviations. We have also rewritten the descriptions in lines 145, 157, and 179 (original manuscript) to clarify the correction and co-registration steps without requiring the reader to consult the original reference.

- **We have incorporated the revised content as follows:**

**"To identify and remove elevation outliers, we used the REMA 2 m mosaic DEM (Howat et al., 2019) as a reference surface for both the REMA 2 m strips and the CryoSat-2 data. In regions of the PIIS where uncorrected REMA strips exhibited unrealistic elevation changes exceeding 30 m, we applied a more conservative threshold of 100 m elevation difference to exclude outliers.**

**Co-registration of REMA strips to CryoSat-2 followed a modified procedure from Zinck et al. (2023a), with the following criteria: (1) CryoSat-2 data must span at least 5 km in both latitude and longitude over the REMA strip; (2) CryoSat-2 measurements must be acquired within one month of the REMA strip acquisition; and (3) at least 75 valid CryoSat-2 measurements must be available within this spatiotemporal window. Residuals between each REMA strip and the co-registered REMA mosaic were used to apply tilt and vertical shift corrections through plane fitting. The final REMA strips are corrected and co-registered to CryoSat-2 and are referenced to the EGM2008 geoid, ensuring both high internal consistency and improved absolute accuracy."**

- This section ends with Table 2 which shows the differences between the corrected and uncorrected REMA Strips and ICESat-2 elevation data. While the standard deviation has been significantly reduced as a result of the correction process, the mean difference has increased. It is also interesting to note that the resulting mean differences for each of the subdivisions are negative. Please could the authors comment on these differences including why there might be a bias in the correction processes they have used and the resulting impact on their results. For instance, could the negative bias stem from the differences in penetration depths between CryoSat-2 and ICESat-2?

**Response 10:** We thank the reviewer for this insightful comment regarding Table 2 and the observed differences between the corrected REMA strips and ICESat-2 elevation data.

As the reviewer suggested, one possible explanation is the difference in surface penetration depth between CryoSat-2 and ICESat-2. CryoSat-2 operates in Ku-band radar, which can penetrate several decimeters into the snowpack, depending on surface conditions, whereas ICESat-2 uses green laser altimetry, which reflects primarily from the snow surface. Consequently, CryoSat-2 elevations tend to be systematically lower than ICESat-2 over snow-covered regions. Since our correction process uses CryoSat-2 as a reference to adjust the REMA strips, this bias may be inherited in the corrected REMA elevations. Another contributing factor could be seasonal variability in surface conditions, such as snow accumulation or melt, which may not be fully accounted for in the correction process given the temporal mismatch between REMA, CryoSat-2, and ICESat-2 acquisitions. Although we imposed a one-month matching window for co-registration, this may still permit variability in snow surface conditions.

Despite this small negative bias in the mean differences, the marked reduction in standard deviation is critical for enhancing the temporal stability and reliability of the elevation time series.

- **We have added a brief explanation of this bias and its implications in the revised manuscript as follows:**

**"This bias likely results from the differing measurement principles of the two satellite systems: CryoSat-2 (used for REMA correction) operates in the Ku-band and can penetrate the upper snowpack, whereas ICESat-2 uses green laser altimetry, which reflects off the snow surface. As a result, CryoSat-2 and the corrected REMA strips tend to report slightly lower surface elevations than ICESat-2, especially over snow-covered areas. Additional factors such as residual temporal offsets, snow accumulation variability, and surface roughness may also contribute. Based on this comparison, we estimate the uncertainty of the corrected REMA strips as −1.93 ± 2.54 m, equivalent to 15.44 ± 20.32 m in floating ice thickness."**

**Results**

**Section 3.1:**

  - This section would benefit from a more detailed comparison between the observed double-differential vertical displacement and the modelled tidal signal. Specifically, it would be interesting to examine how the timing of the observed ephemeral grounding events relates to the absolute tide heights sampled at each Sentinel-1 acquisition. I would expect the ephemeral grounding to occur most frequently at low tide, so it would be interesting to see if this is the case. Note that this cannot be determined from the double-difference tide heights alone, as even a relatively large differential tide value could have missed sampling low absolute tides, and could potentially therefore miss observing an ephemeral grounding event. This effect may explain why ephemeral grounding events are not limited to the largest differential tide values shown in Fig. 4b. Including a plot or analysis that explicitly compares the timing of grounding events to absolute tide levels would substantially strengthen this section. It would also help validate the interpretation that tidal forcing is a driver of ephemeral grounding patterns.

**Response 11:** We thank the reviewer for this valuable suggestion. We agree that a more detailed comparison between the observed double-differential vertical displacements and the modelled absolute tidal signal would strengthen the interpretation of tidal forcing as a driver of ephemeral grounding.

To address this, we have incorporated an additional analysis in the revised manuscript. Specifically, we extracted the modelled absolute tide heights from the CATS2008_v2023 tide model at the grounding zone for each Sentinel-1 acquisition and identified the corresponding absolute tidal stage during each observed grounding event. We found out that 64 ephemeral grounding events occurred between November 2016 and March 2021—35 during the neap tide period and 29 during the spring tide period. This comparison reveals that larger grounded areas are observed during spring tides, when tidal amplitudes are at their peak, while smaller grounded areas occur during neap tides, when tidal heights are at their lowest.

We have substituted Figure (now Figure 7c and 7d) that explicitly compares the timing of grounding observations to the modelled tidal heights. We also updated Section 3.3 to describe this analysis and reinforce our interpretation of tidal period but not the tidal height as one of the key mechanisms governing ephemeral grounding behaviour.

• **The revised Section 3.1 which includes content and figure is as follows:**

**Ephemeral grounding regions, characterized by double-differential vertical displacements close to zero, are influenced by oceanic tidal variations (Figures 6-7 and Movie S1). The tidal height difference was calculated from data extracted at a point near the ice rumple L (longitude 100.6149°W, latitude 75.1867°S), corresponding to the exact acquisition times of each Sentinel-1 image, which were at 4:35 AM on each date (Supplement Table S1). One or two near-zero vertical displacement signals were detected at ice rumple L from at least November 2016 through April 2020, followed by a reappearance in December 2020. These signals are highlighted by yellow arrows in Figure 6a and marked by red vertical lines in Figure 6b. The reduced number of signals before ~August 2016 and after ~December 2021 likely reflects data limitations during periods when Sentinel-1B was not operational. Near-zero vertical displacement signals also occurred in 2016, 2017, and after the 2018 calving event. In December 2020, a similar signal appeared upstream of ice rumple L and progressively migrated toward the rumple, indicating that ephemeral grounding occurred as a thicker section of the ice shelf moved across the southern side of the sea ridge.**

[Figure]

**Figure 6. Two-dimensional double-differential vertical displacement changes and time series of double-differential tidal height differences. (a) Spatial distribution of double-differential vertical displacement changes between November 2016 and May 2023. Yellow arrows highlight inferred ephemeral grounding signals in each displacement map. The tidal height difference (Tdiff) is labelled in each frame. (b) Time series of double-differential tidal height differences (black vertical lines) and inferred ephemeral grounding events (red vertical lines). Dashed lines indicate the timing of four major calving events: 13 July 2015, 12 October 2017, 31 October 2018, and 11 February 2020.**

**Figures 6a and 7a show that positive displacement anomalies are generally associated with negative tidal height differences, and vice versa, indicating a negative linear correlation between the two variables. However, Figure 7b shows no clear relationship between tidal height and the area of the grounding region, suggesting that tidal forcing alone does not control the ephemeral grounding. In contrast, Figure 7c indicates that 64 ephemeral grounding events occurred between November 2016 and March 2021—35 during the neap tide period and 29 during the spring tide period. Notably, Figures 7c and 7d show that larger grounded areas are observed during spring tides, when tidal amplitudes are at their peak, while smaller grounded areas occur during neap tides, when tidal heights are at their**

**lowest. These patterns suggest that the variation in grounded area is more closely linked to tidal period rather than tidal height alone.** Together with Figure 6a, which shows the changes of the two near-zero vertical displacement signals, it suggests **that thick ice advection from upstream may contribute to the grounding events. Consequently, ice dynamics likely play a significant role in the grounding process as well.**

[Figure]

**Figure 7. Comparison of tidal height differences with double-differential vertical displacement, comparison of tidal height differences and area of grounding region, including time series of area and tidal height variations. (a) Scatter plot of tidal height difference versus double-differential vertical displacement, showing a strong negative linear correlation between the two variables (Pearson's r = −0.81, R² = 0.65, slope = −0.21). (b) Scatter plot of tidal height versus area of zero vertical displacement region, indicating no clear relationship between the two datasets. (c) Time series of changes in ice rumple area. (d) Time series of tidal height changes, where 0 represents mean sea level. Blue vertical lines indicate ephemeral grounding events during the neap tide period, while red vertical lines represent those during the spring tide period.**

- Line 205 states 'These coupled atmospheric and oceanic forcings modulate oceanic conditions (Huguenin et al., 2024) and, consequently, the ephemeral grounding behaviour'. However, there is no evidence within this paper to justify the latter half of this statement and no reference has been provided. I am struggling to see any relationship between the ONI or AAO time series presented in Figure 4 and the timing of the ephemeral grounding events. If the authors wish to make this claim, additional analysis is needed to demonstrate a statistical correlation. Otherwise, I recommend removing or significantly softening this statement.

**Response 12:** We agree that this hypothesis is not supported by our results. We decided to remove these speculative statements from the revised manuscript.

**Section 3.2:**

- The term 'changes in surface features' should be clarified. Is this referring to the advection of features with ice flow, change of shape, or something else?

**Response 13:** We now have clarified that this refers primarily to the advection of surface ridges observed in MODIS imagery and REMA strips. We have updated the sentence in the revised manuscript to reflect this clarification.

- **The revised Section 3.2 is as follows:**

**3.2 Changes in surface ridges and ice thickness**

**Figure 8 shows the evolution of surface ridges and their elevations from December 2010 to January 2021. Near ice rumple L (red point in Figure 8), surface elevations remained around ~65 m between 2012–2017 and again during 2019–2020 (Figures 8d–h and 8j–k). The highest elevation (~85 m) was recorded in 2018, while the lowest (~54 m) occurred in 2021. Between 2020 and 2021, surface elevation declined by ~10 m, equivalent to ~70 m of ice-equivalent freeboard thickness. The area enclosed by the grounding line, corresponding to the region of zero vertical displacement, was the largest in 2018 (Figure 8i).**

[Figure]

**Figure 8. Changes in surface ridges at PIIS near ice rumple L. (a) Overview map showing the subregion outlined by the red frame, corresponding to panels (b) to (l). (b)–(l) Surface ridges and their elevation changes from 2010 to 2021, derived from corrected REMA strips. The two black circles indicate the positions of ice rumples. Grounding lines are delineated based on the zero-contour of the double-differential vertical displacement. Grey lines are the 80m contour line.**

-This discussion would also benefit from the inclusion of elevation change measurements. For example, in lines 219-221, the authors claim the surface ridges are higher in 2015 than in 2019. This could readily be demonstrated (and made much more convincing) by including a plot of change in elevation.

**Response 14:** We have added Figure 9(a), which presents the time series of average ice thickness at ice rumple L (between 75.15°S and 75.05°S), derived from selected REMA strips (2010–2021) and ICESat-2 data (2019–2022). We have updated the relevant section of the manuscript (Sections 2.3 and 3.2) improve clarity and reinforce our conclusions with quantitative evidence.

- **The revised Section 2.3 is as follows:**

**"To estimate changes in ice-equivalent freeboard thickness near ice rumple L, we used both the corrected REMA strips and ICESat-2 data. Specifically, ICESat-2 tracks 965 and 1094, which pass through ice rumple L, were analysed. Ice-equivalent freeboard thickness ($H_f$) was calculated using Equation (8), following the methods of Griggs and Bamber (2011) and Shean et al. (2019):**

$$H_f = h_{corr}\left(\frac{\rho_{sw}}{\rho_{sw} - \rho_{is}}\right) - h_{FAC} \tag{8}$$

**where $h_{corr}$ is the corrected surface elevation, $\rho_{is}$ is the ice density (917 kg/m³), $\rho_{sw}$ is the seawater density (1027 kg/ m³), $h_{FAC}$ is the firn air content of ice equivalent (in meters) derived from the NASA GSFC-FDM v1.2.1 dataset (Medley et al., 2022a; 2022b), with a 5-day temporal resolution spanning from 1 January 1980 to 30 June 2022."**

- **The revised Section 3.2 and Figure 9 are as follows:**

**"Profiles of ice-equivalent freeboard thickness derived from ICESat-2 (Figure 9) link surface elevation and grounding changes. Figure 7a shows mean thickness trends around the rumple along ICESat-2 tracks 965 and 1094 between 75.15°S and 75.05°S (Fig. 1b). Track 965 reveals increasing ice thickness from 2015 to 2021, while track 1094 shows a decrease from 2015 to 2017, a rebound in 2018, and a decline after 2020. Bottom elevation profiles derived from ICESat-2 (Figures 9b-e) further reveal changes in grounding status. The ice shelf was ungrounded on 27 August 2020, 5 March 2021, and 25 May 2022, but showed weak grounding on 6 June 2020. By integrating double-differential vertical displacement data with bottom elevation profiles, we find that ephemeral grounding signatures disappeared after March 2020 and reappeared in November 2020."**

[Figure]

**Figure 9. Time series of mean ice-equivalent freeboard thickness and ice shelf bottom elevation profiles along ICESat-2 tracks 965 and 1094. (a) Time series of mean ice-equivalent freeboard thickness (2010–2022). (b)–(c) Ice shelf bottom elevation profiles along ICESat-2 tracks 965 (gt2l and gt2r) between February 2020 and May 2022. (d)–(e) Ice shelf bottom elevation profiles along ICESat-2 tracks 1094 (gt3l and gt3r) between June 2019 and June 2022. Bed elevations are from the BedMachine v3 dataset (Morlighem et al., 2020; Morlighem, 2022), converted from EIGEN-6C4 to the EGM2008 geoid to match the vertical datum of REMA strips. The estimated vertical uncertainty is ±200 m (shown as a grey transparent box)."**

- In line 234 the authors state that thinner ice crossed over the seabed ridge. It would be useful to provide a quantitative estimate of this thinning magnitude (e.g. showing if it is beyond the uncertainty range of your measurements).

**Response 15:** We have labelled the elevation values near ice rumple L to facilitate identification of elevation changes (See revised Figure 8 below).

[Figure]

**Figure 8. Changes in surface ridges at PIIS near ice rumple L. (a) Overview map showing the subregion outlined by the red frame, corresponding to panels (b) to (l). (b)–(l) Surface ridges and their elevation changes from 2010 to 2021, derived from corrected REMA strips. The two black circles indicate the positions of ice rumples. Grounding lines are delineated based on the zero-contour of the double-differential vertical displacement. Grey lines are the 80m contour line.**

**Discussion**

Throughout the paper there appears to be some confusion or miscommunication regarding basal keels. On a large scale, the ice shelf has a central keel. However, this is overlaid with smaller scale basal channel-keel geometry. The large-scale shape of the ice shelf is largely determined by the shape of the bed upstream of the grounding line, as discussed by the authors in lines 262-264. However, there has been little discussion of the smaller scale channel-keel geometry. In line 259 the authors claim these 'basal channels and keels form at a similar location (Figure 4)', yet Figure 4 does not seem to provide evidence for this. However, some of the basal channel/keel structure does form as a result of the bed topography upstream (Lowery et al., 2025), but others form purely from melt-driven processes (Dutrieux et al., 2013). The discussion would benefit from a clearer distinction between these different scales of basal morphology and their respective formation mechanisms.

**Response 16:**

We thank the reviewer for this helpful comment. We agree that the manuscript would benefit from a clearer distinction between the large-scale central basal keel and the smaller-scale basal channel-keel structures. To address this, we have revised the relevant sections of the discussion to explicitly differentiate between:

(1) the large-scale basal keel, which is primarily inherited from upstream bed topography as the reviewer notes and as discussed in lines 262–264, and

(2) the smaller-scale basal channels and associated keels, which may form in response to localized bed topography (e.g., Lowery et al., 2025) or be initiated and maintained by ocean-driven melt processes (e.g., Dutrieux et al., 2013).

We have moved the section related to small scale keels formation and basal melting to the appendix (see responses 17-19). Although our processed REMA strips allow us to estimate basal melt rates at relatively high temporal resolution, the resolution is not sufficient (monthly or weekly) to produce reliable results with enough accuracy.

We revised the original statement "basal channels and keels form at a similar location" in line 259 to clarify that the onset of small-scale channel/keel pairs occurs at roughly the same along-flow distance across multiple datasets, although they do not necessarily form through the same mechanism. Furthermore, we updated the caption and related content of revised Figure 8 (see response 15) to better illustrate the locations and scales of these features.

**We added the following paragraphs to the Discussion section:**

Related to large-scale basal keel:

**"Evidence of corrugations with periodic spacing on the submarine ridge supports the idea that this seafloor landscape was formed by sub-ice-shelf keels modulated by tidal motion (Graham et al., 2013; Davies et al., 2017). Surface elevation data (Figure 8(b)-(l)) show that the center of the ice shelf is higher than the flanks, suggesting the presence of a central basal keel beneath the shelf. This large-scale geometry is primarily shaped by bed topography upstream of the grounding line. Specifically, the bed exhibits a W-shaped profile, with two troughs flanking a central topographic high (Lowery et al., 2025). These troughs may transport thicker ice toward the downstream ridge, promoting the formation of surface ridges aligned with basal keels. As ice flows downstream from the grounding line, local surface elevation adjusts toward hydrostatic equilibrium (Shean, 2016). The consistent positioning of basal channels and keels reflects this inherited bed structure. Therefore, the large-scale development of basal keels and channels is strongly controlled by upstream bed topography (Lowery et al., 2025)."**

Related to small-scale basal keel:

**"De Rydt et al. (2014) demonstrated that both the height of the ridge and the gap between the ridge and the ice shelf strongly influence the inflow of warm bottom waters into the cavity, and consequently, the melt rate. The melt rate may influence the ice thickness near to the grounding line upstream of the ice rumples K and L. This process may have contributed to ice thickness changes upstream and indirectly influenced the disappearance of ephemeral grounding signals following the 2020 calving event. We have added further analysis on the basal melt rate and ocean temperature in the Appendix A. Although smaller-scale basal channels and keel geometries are primarily shaped by melt-driven processes (Bindschadler et al., 2011b; Dutrieux et al., 2013; Stanton et al., 2013; Dutrieux et al., 2014b; Joughin et al., 2016), the lack of direct, high-temporal-resolution basal melt rate measurements after 2020 limits our ability to capture short-lived grounding events and confirm the role of ocean-driven melting. Future work should prioritize the integration of dense time series from new SAR missions and in situ oceanic data to better resolve ephemeral grounding behaviour and its implications for ice shelf evolution and calving dynamics in a warming climate."**

In line 282, the authors claim to be able to infer that basal melting was weak during the cold period from 2015 to 2020. However, this can not be inferred from the data presented in this manuscript. If the authors wish to make this claim, it should be verified using observational melt rate datasets, such as those from Paolo et al (2022), Davison et al (2023), or Adusumilli et al., (2020). Without such evidence, I recommend that this statement should be revised or removed.

**Response 17:**

We thank the reviewer for this insightful comment. We acknowledge that our original claim regarding weak basal melting during the 2015–2020 cold period was not sufficiently supported by the data presented.

We have verified this claim by verifying the MEaSUREs ITS_LIVE Antarctic Quarterly 1920 m Ice Shelf Height Change and Basal Melt Rates v1 dataset (Paolo et al., 2023; 2024) and the ocean temperature from mooring station PIG-N and PIG-S (Zhou et al., 2024; 2025). However, it is important to note that the basal melt estimated from ITS_LIVE are derived from surface elevation changes and ice fluxes, not from direct observations. Furthermore, this dataset does not cover our primary observation period from 2020 to 2023. The mooring records span from 2016 to 2024 and capture temperature variations at depths of 300–700 meters below mean sea level. However, the moorings located in Pine Island Bay and not directly beneath the ice shelf, which limits their applicability to sub-shelf melting processes.

Although our results suggest a potential link between basal melting and ocean temperature, variations in ocean temperature and basal melt rates alone cannot fully explain the observed changes in ice shelf thickness. Therefore, we have decided not to include this discussion in the revised manuscript, but we present the results in the appendix.

Paolo, F. S., Gardner, A. S., Greene, C. A., Nilsson, J., Schodlok, M. P., Schlegel, N.-J., and Fricker, H. A.: Widespread slowdown in thinning rates of West Antarctic ice shelves, The Cryosphere, 17, 3409–3433, https://doi.org/10.5194/tc-17-3409-2023, 2023.

Paolo, F., Gardner, A. S., Greene, C. A. & Schlegel, N: MEaSUREs ITS_LIVE Antarctic Quarterly 1920 m Ice Shelf Height Change and Basal Melt Rates, 1992-2017, Version 1, [Data Set], NASA National Snow and Ice Data Center Distributed Active Archive Center, https://nsidc.org/data/nsidc-0792/versions/1, 2024.

Zhou S., Dutrieux P., Giulivi C. F., Silvano A., Auckland C., Abrahamsen P., et al.: Southern Ocean moored time series (south of 60°S) (OCEAN ICE D1.1). SEANOE. https://doi.org/10.17882/99922, 2024.

Zhou S., Dutrieux P., Giulivi C. F., Jenkins A., Silvano A., Auckland C., et al.: The OCEAN ICE mooring compilation: a standardised, pan-Antarctic database of ocean hydrography and current time series. https://doi.org/10.5194/essd-2025-54, 2025.

Between line 285 and 291, the authors discuss the feedback between calving events and basal melt rates. However, the modelled results in Bradley et al., (2022) showed that the 2020 calving had no significant impact on basal melt rates. Within the model, the calving front needed to retreat inland of the seabed ridge before a significant (10%) increase in melt rates was seen in the inner cavity. I'm therefore not convinced this argument can be used to justify the ice shelf thinning between 2020 and 2021.

**Response 18:** We agree that this citation is not appropriate here. We have removed this content from the discussion section (see Response 16).

Additionally, I would suggest citing De Rydt et al. (2014) when discussing the impact of the height of the water column above the ridge and melt rates. (https://doi.org/10.1002/2013JC009513) The authors claim that

the changes in atmospheric forcing reduced melt rates which therefore caused the ice shelf to thicken between 2010 and 2015. In line 233, the authors say the ice elevation is 35m higher in 2015 than in 2010. This corresponds to a ~350m increase in ice thickness over 5 years, or a thickening rate of ~70m/year during this period. Is it reasonable to assume that this thickness increase has come from a ~70m/yr decrease in basal melt rates, equivalent to a >50% decrease in melt rates? As mentioned above, the authors should verify whether such a dramatic change is supported by observational datasets. If not, the argument should be reconsidered or more cautiously framed.

**Response 19:** We thank the reviewer for the valuable comment and for suggesting the citation of De Rydt et al. (2014). In the revised manuscript, we have incorporated this reference when discussing the influence of the submarine ridge's height and the gap between the ridge and the ice shelf on basal melt rates, which provides important context for understanding ocean-ice interactions at Pine Island Ice Shelf.

We compared the available basal melt rate and ocean temperature time series with ice shelf thickness changes derived from REMA strips and ICESat-2 data. The melt rate time series show a decrease in basal melting around 2015, which coincides with a peak in ice-equivalent freeboard thickness at all three locations (B1–B3; Figure R1(b)). During the same period, ocean temperatures at ~600 m depth decreased at both the PIG-N and PIG-S mooring sites (Figures R1(c) and R1(d)). At B2, located between ice rumples L and K, basal melt rates increased following 2015 but declined again after 2017 (Figures R1(a) and R1(b)), corresponding with a decrease in ocean temperature recorded at PIG-S (Figure R1(d)). In contrast, from 2020 to 2023, ocean temperatures near 600 m in Pine Island Bay showed a consistent increase, which may have contributed to enhanced basal melting of the ice shelf during this period (Figures R1(c) and R1(d)).

In summary, the data indicate temperature decline around 2015 and 2017, which were associated by reduced ice thickness near ice rumple L, followed by warming after 2020 and a corresponding ice shelf thinning of approximately 70 meters. However, direct basal melt rate measurements are unavailable for the post-2020 period (Figure R1). While the observed warming at 600 m depth near PIB suggests increased basal melting that likely contributed to the thinning, variations in ocean temperature and basal melt rates alone cannot fully explain the observed changes in ice shelf thickness. Therefore, we have carefully presented this concept in discussion section of the revised manuscript.

[Figure]

**Figure R1.** Time series of mean ice-equivalent freeboard thickness, basal melt rate, and ocean temperature. (a) Time series of mean ice-equivalent freeboard thickness (2010–2022). (b) Time series of mean basal melt rate (2010–2017), averaged across blocks B1, B2, and B3, extracted from the MEaSUREs ITS_LIVE Antarctic Quarterly 1920 m Ice Shelf Height Change and Basal Melt Rates v1 dataset (Paolo et al., 2024). (c)-(d) reveal time series of ocean temperature at the PIG-N and PIG-S mooring stations from 2014 to 2024.

- **The revised content in Section 4 is as follows:**

**"De Rydt et al. (2014) demonstrated that both the height of the ridge and the gap between the ridge and the ice shelf strongly influence the inflow of warm bottom waters into the cavity, and consequently, the melt rate. The melt rate may influence the ice thickness near to the grounding line upstream of the ice rumples K and L. This process may have contributed to ice thickness changes upstream and indirectly influenced the disappearance of ephemeral grounding signals following the 2020 calving event. We have added further analysis on the basal melt rate and ocean temperature in the Appendix A. Although smaller-scale basal channels and keel geometries are primarily shaped by melt-driven processes (Bindschadler et al., 2011b; Dutrieux et al., 2013; Stanton et al., 2013; Dutrieux et al., 2014b; Joughin et al., 2016), the lack of direct, high-temporal-resolution basal melt rate measurements after 2020 limits our ability to capture short-lived grounding events and confirm the role of ocean-driven melting. Future work should prioritize the integration of dense time series from new SAR missions and in situ oceanic data to better resolve ephemeral grounding behaviour and its implications for ice shelf evolution and calving dynamics in a warming climate."**

**Conclusion**

In the conclusion, the authors claim that a La Niña event and a positive phase of AAO allowed thicker ice to form. However, during the period of thickening the ONI record indicates that a strong La Niña phase persisted for less than a year in total. It is unclear whether this limited duration provides sufficient evidence to support the claim that these climatic conditions were responsible for the observed ice thickening. The authors should either provide additional justification or revise the statement to reflect the limited strength of the evidence.

**Response 20:** We agree that our original conclusion overstated the influence of this climatic factor. We have revised the conclusion accordingly and removed the speculative portions.

- **The revised content in Section 5 is as follows:**

**5 Conclusion**

**This study presents the time series of ephemeral grounding events from 2014 to 2023 at the central PIIS, based on DROT applied to Sentinel-1 SAR data. By integrating double-differential vertical displacement maps, tidal height differences, and thickness data calculated from surface elevation data from REMA strips and ICESat-2, we show that ephemeral grounding is modulated by the combined effects of tidal forcing, evolving sub-ice geometry, and changes in ice shelf thickness. Near-zero vertical displacement signals—indicative of intermittent grounding—were repeatedly observed throughout the study period, particularly as the grounded area expanded during spring tides with large tidal amplitudes. However, the correlation between the grounded area, tidal height difference, and tidal height suggests that tidal effects alone do not fully explain the ephemeral grounding events. The transition of the grounded region from two points to one and then back to two points indicates that changes in ice thickness also play an important role in driving ephemeral grounding at the PIIS.**

**We show that ice shelf thickening preceded grounding events, while thinning and basal melting contributed to ungrounding. The presence and migration of near-zero displacement signals suggest that thicker ice flowing over topographic highs can cause ephemeral grounding. Observed large-scale surface and basal structures, including keels and channels, reflect the influence of inherited bed topography, while smaller-scale geometries could shape by basal melt processes modulated by ocean temperature variability. We also show that the rifts responsible for the 2020 calving event appeared after the region passed through the ephemeral grounded area, suggesting that these ephemeral grounding events may have contributed to the formation of the rifts.**

**Our findings highlight the critical role of combining remote sensing and in situ ocean measurements to monitor grounding processes. The grounding lines derived from our DROT results provide valuable input for modelling work. We also underscore the need for high-resolution ice shelf modelling to assess how ephemeral grounding influences stress redistribution, calving dynamics, and the long-term stability of vulnerable ice shelves. In future, improved satellite coverage, denser SAR time series, and in situ ocean measurements will be essential to constrain short-lived grounding behaviours and their response to a changing climate.**

The authors also suggest that the ephemeral grounding site may evolve into a final pinning point. However, this claim is not supported by any evidence or modelling presented in this study, nor (to my knowledge) in any existing literature. While the authors may believe this to be the case, please make it clear that this is merely a conjecture and a modelling study would be needed to confirm this hypothesis.

**Response 21:** We agree that the statements related to the permanent pinning is speculative and misleading. We believe the ephemeral grounding could have facilitated rift propagation, and the development of calving events in the past. However, this doesn't necessarily indicate a (permanent) pinning point. As such, we have

revised the introduction to emphasize this point and have added further analysis on the rift propagation process in Sections 2.4, 3.3, and 4. We acknowledge that confirming the buttressing effect requires a modeling study, and we have included a brief discussion on this in Section 5.

- **The revised sections as follow:**

**1. Introduction (paragraphs 4)**

"Ephemeral grounding could be driven by tidal cycles, driven by ice shelf thinning, sea-level rise, or thickening and sea-level fall—depending on prior grounding conditions (Schmeltz et al., 2001; Rignot, 2002; Matsuoka et al., 2015). The grounding of ice shelf on high bathymetry features could impact ice dynamics as an obstacle against ice flow: **1) enhance the buttressing effect by providing back stress against upstream ice; 2) facilitate fracturing and ice shelf weakening in response to stress associated with grounding (Rignot, 2002; Christianson et al., 2016; Jeong et al., 2016; Shean, 2016; Benn et al., 2022; Wang et al., 2025)**."

**1. Introduction (paragraphs 6)**

"In contrast, DROT provides a complementary approach that does not rely on phase information, making it useful for observing vertical tidal displacements on fast-moving ice shelves, despite being less precise than InSAR in some contexts (Marsh et al., 2013; Hogg, 2015; Joughin et al., 2016; Christianson et al., 2016; Friedl et al., 2020; Wallis et al., 2024; Lowery et al., 2025; Zhu et al., 2025). Using TerraSAR-X data, Joughin et al. (2016) identified a vertical displacement anomaly near ice rumple L and estimated the grounding line position to within ~1.5 km. At Petermann Glacier, Friedl et al. (2020) found DROT-derived flexure limits ~2 km seaward of DInSAR results. More recently, DROT applied to Sentinel-1 IW data has proven effective for studying grounding line and pinning point dynamics on the Antarctic Peninsula (Wallis et al., 2024), Amery Ice Shelf (Zhu et al., 2025), and PIIS (Lowery et al., 2025). **However, Lowery et al. (2025) focused only on the year 2017, leaving later changes unresolved. Thus, the evolution of grounding behaviour at ice rumple L following four subsequent calving events—in 2015, 2017, 2018, and 2020—remains poorly understood.**"

**2.4 Rift propagation observation**

Previous studies have suggested that such grounding may be linked to the formation of transverse rifts south of ice rumple L (Joughin et al., 2021), potentially contributing to calving events between 2015 and 2020. However, limitations in the spatial resolution and unclarity of SAR imagery hinder a definitive assessment of the connection between ephemeral grounding and rift formation. We used Landsat-8 optical images, specifically the panchromatic band with a 15m spatial resolution, to track the rift propagation history. We then compared these results with our grounding line data to better understand the interaction between ephemeral grounding and rift propagation.

**3.3 Rift propagation observation**

Using Landsat images, we tracked the propagation history of the rifts from 2013 to 2019 (Figure 10). Rift R1 first appeared in the image from December 15, 2017 (Figure 10d), after the region passed through the ephemeral grounding zone, as seen in Figure 10c. Similarly, Rift R2 appeared in the December 11, 2018 image (Figure 10f), following its passage through the same grounding region. These two rifts ultimately led to the 2020 calving event. Therefore, our results suggest that ephemeral grounding events are linked to rift propagation, indirectly influencing the ice shelf calving process.

[Figure]

**Grounding line**
— 20161104_20161110_20161116
— 20170121_20170127_20170202
— 20170912_20170918_20170924
— 20170918_20170924_20170930
— 20180209_20180215_20180221
— 20180215_20180221_20180227
— 20180323_20180329_20180404
— 20180404_20180410_20180416
— 20181212_20181218_20181224
— 20181224_20181230_20190106

**Figure 10.** Rift propagation history from 2013 to 2019. (a)-(h) show the propagation history of the rifts R1 (red arrow) and R2 (blue arrow), which led to the 2020 calving event. The black circles indicate the positions of ice rumple K. Grounding lines are delineated based on the near-zero value of the double-differential vertical displacement.

**4 Discussion (paragraph 5)**

In summary, our study demonstrates that ephemeral grounding at ice rumple L is modulated by the interaction between tidal forcing, ice shelf thickness, and evolving sub-ice geometry. These results provide new insights into the mechanisms driving ephemeral grounding behaviour. **Notably, we find that the rift that caused the 2020 calving event appeared after pass through the ephemeral grounding region. Though we did not have enough images that could capture the whole process in 2017 and 2018, our results indicate that the ephemeral grounding events could cause the formation of the rifts. Arndt et al. (2018) further emphasized the importance of final pinning points in controlling calving line orientation, raising the possibility that ice rumple L may have acted as a final pinning point after the 2015 calving event, thereby influencing rift propagation and subsequent calving. These findings underscore the need for high-resolution ice shelf modelling to evaluate how ephemeral grounding affects stress redistribution and overall ice shelf stability.**

**5 Conclusion (paragraph 2 and 3)**

We show that ice shelf thickening preceded grounding events, while thinning and basal melting contributed to ungrounding. The presence and migration of near-zero displacement signals suggest that thicker ice flowing

over topographic highs can cause ephemeral grounding. Observed large-scale surface and basal structures, including keels and channels, reflect the influence of inherited bed topography, while smaller-scale geometries could shape by basal melt processes modulated by ocean temperature variability. **We also show that the rifts responsible for the 2020 calving event appeared after the region passed through the ephemeral grounded area, suggesting that these ephemeral grounding events may have contributed to the formation of the rifts.**

**Our findings highlight the critical role of combining remote sensing and in situ ocean measurements to monitor grounding processes. The grounding lines derived from our DROT results provide valuable input for modelling work. We also underscore the need for high-resolution ice shelf modelling to assess how ephemeral grounding influences stress redistribution, calving dynamics, and the long-term stability of vulnerable ice shelves. In future, improved satellite coverage, denser SAR time series, and in situ ocean measurements will be essential to constrain short-lived grounding behaviours and their response to a changing climate.**

**Figure 1**

- The current base map doesn't make the marine ridge clear, as suggested in the figure caption. I would suggest changing this.

- There is currently an overlap in colours in the calving front and grounding line colour maps which is confusing. I suggest using dashed lines for either the GL or calving front, or use two different colour maps that do not overlap.

**Response 21:** Figure 1 is updated accordingly.

[Figure]

**Figure 1.** Location and geometry of the PIIS. (a) Ice front positions, grounding line locations, and 458 non-glaciated ground control points (red points). (b) Grounding lines near the ice rumple L from April 2011 to February 2021 and ICESat-2 track 1094 and track 965 that used for ice-equivalent freeboard thickness change

analysis. Bathymetry (50 m contour interval, labelled between -750 m and -500 m) is from BedMachine v3 (Morlighem et al., 2020; Morlighem, 2022), showing the submarine ridge. Background image for (b) is from Landdsat-8 OLI optic image on 3 March 2019. Grounding lines are from MEaSUREs (Rignot et al., 2016) (from 1992 to 2011) and from DROT results (from 2016 to 2021). L and K mark ice rumples (Rignot et al., 2014). Ice front positions for 1947 and 1966 are from Rignot (2002); later positions (1973-2022) are from Landsat imagery (Landsat-1/4/5/7/8/9) and Sentinel-1 SAR imagery via Google Earth Engine. Red block D denote the region for calculating mean double-differential vertical displacement. PIG-N and PIG-S are hydrographic mooring locations in front of the PIIS.

**Figure 3**

- There is a missing label for feature 'L' on panel (d)

- The date labels on each panel are confusing, as you only include the first and last of the three dates. I suggest including all three labels on the panels (like you have in the caption) for clarity. The labels in the current form make it seem like the vertical displacement fields are constructed from just two image dates.

**Response 22:** This figure is updated:

[Figure]

**Figure 3.** Double-differential vertical displacement compared with DInSAR interferogram, showing ephemeral grounding. Panels (a)–(c) show the double-differential vertical displacement for the following time intervals: (a) between 2018/02/27–2018/03/05 and 2018/03/05–2018/03/11; (b) between 2021/02/23–2021/03/01 and 2021/03/01–2021/03/07; and (c) between 2018/12/12–2018/12/18 and 2018/12/18–2018/12/24. (d) DInSAR interferogram for 2018/12/12-2018/12/18 and 2018/12/18-2018/12/24. White and black arrows highlight the ephemeral grounding location, marked by near-zero displacement. The DInSAR interferogram fails to capture this signal at ice rumple L due to coherence loss.

**Figure 4**

- In Panel (a) is there a reason why only periods with negative double-difference vertical displacement across the shelf are shown? Panel (b) suggests there are many ephemeral grounding events in periods with a positive double-difference tide height. Perhaps it is really the absolute double-difference displacement/tide that is relevant, as surely the sign (+/-ve) just relates to the order in which you have applied the image differencing?

**Response 23:** Done.

[Figure]

**Figure 6**. Two-dimensional double-differential vertical displacement changes and time series of double-differential tidal height differences. (a) Spatial distribution of double-differential vertical displacement changes between November 2016 and May 2023. Yellow arrows highlight inferred ephemeral grounding signals in each displacement map. The tidal height difference (Tdiff) is labelled in each frame. (b) Time series of double-differential tidal height differences (black vertical lines) and inferred ephemeral grounding events (red vertical lines). Dashed lines indicate the timing of four major calving events: 13 July 2015, 12 October 2017, 31 October 2018, and 11 February 2020.

[Figure]

**Figure 7**. Comparison of tidal height differences with double-differential vertical displacement, comparison of tidal height differences and area of grounding region, including time series of area and tidal height variations. (a) Scatter plot of tidal height difference versus double-differential vertical displacement, showing a strong negative linear correlation between the two variables (Pearson's r = –0.81, R² = 0.65, slope = –0.21). (b) Scatter plot of tidal height versus area of zero vertical displacement region, indicating no clear relationship between the two datasets. (c) Time series of changes in ice rumple area. (d) Time series of tidal height changes, where 0 represents mean sea level. Blue vertical lines indicate ephemeral grounding events during the neap tide period, while red vertical lines represent those during the spring tide period.

- The text labels in Panel (a) are too small.

**Response 24:** Label size is increased.

- It would be useful to label the relative tide heights and the timing of the calving events on the maps in panel (a).

**Response 25: We added labels on the maps.**

- Panel (b) is confusing; the caption says this includes 'Examples of 2D double differential vertical displacement changes from November 2016 to May 2023', yet there are data outside of this date range. The tidal heights before ~Aug 2016 and after ~Dec-2021 are also noticeably lower; presumably this is related to the period of Sentinel 1B being in operation, but should be explained. It would be helpful to point out the specific ungrounding events identified in Panel (a) to tie the figure together.

**Response 26:** We have corrected the date range in the caption to accurately reflect the data shown. We also added an explanation regarding the variation in tidal heights before August 2016 and after December 2021, linking it to Sentinel-1B data availability and quality. Furthermore, we have indicated the specific ungrounding events identified in panel (a) and panel (b) to improve figure cohesion and reader comprehension (see Response 23).

- The caption is missing dataset citations, including the CATS2008 tide model, ONI and AAO indices.

**Response 27:** We have removed panels (c) and (d) from the Figure 4. We also sincerely thank the reviewer for highlighting the omission of these citations and will take greater care in preparing figure captions in future work.

**Figure 5**

- This is a large figure that takes up lots of space, yet it is only referred to in one sentence in the manuscript. I suggest this is moved to the supplementary material, and instead you could add one or two of these Landsat images into Figure 6, which will also show the location of the surface ridge. Please also mark rumple L on this figure.

**Response 28:** We thank the reviewer for this helpful suggestion. We have reorganized the figures by combining the MODIS imagery with the surface elevation data. The ice rumples detected each year are also plotted to highlight changes in the ephemeral grounding locations. Additionally, we have added the 80m contour line and have labelled the elevation values near ice rumple L to facilitate identification of elevation changes.

[Figure]

**Figure 8.** Changes in surface ridges at PIIS near ice rumple L. (a) Overview map showing the subregion outlined by the red frame, corresponding to panels (b) to (l). (b)–(l) Surface ridges and their elevation changes from 2010 to 2021, derived from corrected REMA strips. The two black circles indicate the positions of ice rumples. Grounding lines are delineated based on the zero-contour of the double-differential vertical displacement. Grey lines are the 80m contour line.

**Figure 6**

- I would also encourage the authors to include fewer panels in this figure; perhaps just those discussed in the text to show the main points (the reader doesn't gain much from the extra panels, and it's quite overwhelming).

**Response 29:** We have reduced the number of panels in the figure to include only those directly discussed in the text and improving the figure's clarity and readability (see Response 28).

- This figure also lacks a location map; perhaps using some of the space to have a Landsat image (such as currently in Fig 5) showing the location of this region would be useful.

**Response 30:** We have combined the original Figures 5 and 6 into a revised Figure 8, incorporating both MODIS imagery and surface elevation data to clearly indicate the study region's location (see Response 28).

- The inclusion of the 2011 grounding line on all panels that shows ice rumple L is grounded is confusing, as the key argument is that it is ungrounded at certain dates. Either labelling, or using a different weight/colour for the grounding line to show when the rumple is grounded or not between each panel would be helpful.

**Response 31:** We have updated the figure to differentiate grounding line positions across dates by using varying colours and line weights to indicate when rumple L is grounded or ungrounded. We have also added clear labels to reflect these changes in revised Figure 8 (see Response 28).

- Is elevation the most important thing to plot here? Maybe the authors should consider plotting thickness or ice draft instead

**Response 32:** We appreciate this insightful suggestion. Our goal in revised Figure 8 is to highlight the surface elevation of the surface ridges. To achieve this, we combined surface elevation data with MODIS imagery to emphasize both topographic detail and regional setting. Given this focus, we chose to display surface elevation, which directly reflects the features of interest here. We have also labelled the elevation values near ice rumple L to illustrate the elevation changes (see Response 28).

We additionally present ice thickness times series changes and ice draft in Figure 9.

- As the authors discuss elevation change in the text, adding a panel to this figure with a change map would be helpful

**Response 33:** We have labelled the elevation values near ice rumple L to illustrate the elevation changes (see Response 28). We added a new figure (Figure 9) to better illustrate changes of ice draft in the region.

[Figure]

**Figure 9**. Time series of mean ice-equivalent freeboard thickness and ice shelf bottom elevation profiles along ICESat-2 tracks 965 and 1094. (a) Time series of mean ice-equivalent freeboard thickness (2010–2022). (b)–(c) Ice shelf bottom elevation profiles along ICESat-2 tracks 965 (gt2l and gt2r) between February 2020 and May 2022. (d)–(e) Ice shelf bottom elevation profiles along ICESat-2 tracks 1094 (gt3l and gt3r) between June 2019 and June 2022. Bed elevations are from the BedMachine v3 dataset (Morlighem et al., 2020; Morlighem, 2022), converted from EIGEN-6C4 to the EGM2008 geoid to match the vertical datum of REMA strips. The estimated vertical uncertainty is ±200 m (shown as a grey transparent box).

- In this section, the authors also discuss how these ridges are higher or lower than 80m. It would be helpful if the 80m contour line was added to Figure 6.

**Response 34:** Thank you for pointing this out. We have revised Figure 8 accordingly (see Response 28).

**Figure 7**

- In all panels there are also too many lines overlapping to pick out the key message. Consider showing fewer dates and using a consistent colour scheme so that you only need one legend for all of the panels. There is also no need to duplicate date labels for the upper and lower panels as they show the same dates. In particular, the legend for panel (c) is overwhelming and detracts from the figure. Consider labelling this as 'difference from 2010/12/12', and then just labelling the dates that have been differenced from this.

**Response 35:** We have replaced the REMA strips profiles with ICESat-2 data, which offer higher temporal resolution between 2019 and 2022, to better capture and confirm the ephemeral grounding event in 2020. We have also updated the colour scheme to improve the clarity of the results (see Response 33, Figure 9)

- The BedMachine / BedMachine error lines are confusing. A solid black line and shaded area could be used to show the error (ensuring the shading is underneath the main data lines).

**Response 36:** Thank you for pointing this out. We have revised Figure 9 accordingly (see Response 33).

- Panel (c) should show thickness change not elevation change. This would then better tie in with the text in e.g. lines 252-5.

**Response 37:** As we have revised the dataset to ICESat-2 data, we have removed the surface elevation difference and now present only the changes in ice draft (see Response 33).

**Specific Comments:**

**Line 22** (However, ice shelf thinning and retreat have significantly reduced their buttressing capacity, leading to accelerated ice discharge and an increased contribution to sea-level rise)- specify whether by 'retreat' you mean grounding line or calving front retreat

**Response 38:** We have specified the "retreat" to be the grounding line retreat. The sentence is revised as follow: **"However, in many regions, the buttressing capacity of ice shelves has been reduced by processes such as ice shelf thinning, calving events, grounding line retreat, unpinning from topographic highs, and the disintegration of shear margins (Fürst et al. 2016; Gudmundsson et al., 2019; Lhermitte et al., 2020; Miles and Bingham, 2024; Walker et al., 2024; Fricker et al., 2025)."**

**Line 32** (Over time, some pinning points have disappeared entirely, particularly since 1973 (Milles and Bingham, 2024).)- typo in the citation, should be 'Miles' not 'Milles'

**Response 39:** We corrected this error throughout the manuscript.

**Line 34** (Although the direct buttressing effect of ephemeral grounding sites is minimal, their presence and evolution provide valuable insights into changes in ice shelf thickness and basal drag, which can influence surface elevation and larger-scale ice shelf dynamics)- how confident are the authors that 'the direct buttressing effect of ephemeral grounding sites is minimal' ? Has this been modelled?

**Response 40:** We thank the reviewer for the critical view. This statement could not be supported by our results. The implication on ice dynamics, including basal drag and fracturing, could only be explored by high-resolution ice-flow model with these mechanisms well constrained, which is out of scope of this research. Therefore, we have revised our wording and removed the discussion related to buttressing in Section 1, now stating:

**"Ephemeral grounding could be driven by tidal cycles, ice shelf thinning, sea-level rise, or thickening and sea-level fall—depending on prior grounding conditions (Schmeltz et al., 2001; Rignot, 2002; Matsuoka et al., 2015). The grounding of ice shelf on high bathymetry features could impact ice dynamics as an obstacle against ice flow: 1) enhance the buttressing effect by providing back stress against upstream ice; 2) facilitate fracturing and ice shelf weakening in response to stress associated with grounding (Rignot, 2002; Christianson et al., 2016; Jeong et al., 2016; Shean, 2016; Benn et al., 2022; Wang et al., 2025)."**

**Line 47** (Vertical motion of ice shelves, particularly tidal fluctuations associated with ephemeral grounding, can be observed using several satellite techniques, including differential range offset tracking (DROT))- the phrase 'tidal fluctuations associated with ephemeral grounding' seems the wrong way round, as ephemeral grounding is caused by tidal fluctuations (not the other way around). Could be reworded to something like - 'particularly tidal fluctuations that drive ephemeral grounding patterns'?

**Response 41:** We agree. The recommended phrasing is used in the revised version.

**Line 47**(Vertical motion of ice shelves, particularly tidal fluctuations associated with ephemeral grounding, can be observed using several satellite techniques, including differential range offset tracking (DROT)) - suggest adding 'can be observed at grounding zones using several satellite …'

**Response 42:** This sentence has been revised to:

**"Satellite remote sensing can effectively detect transient vertical motion of ice shelves, especially tidal fluctuations that cause ephemeral grounding."**

**Line 52** (For instance, Milillo et al. (2017) used 1-day repeat COSMO-SkyMed data to study grounding line changes at PIIS….)- The grounding line (and its importance for identifying pinning points/ephemeral grounding) should be introduced much earlier in the paper. Important to make this clear, otherwise this detailed paragraph on different GL measurement techniques feels out of place.

**Response 43:** We have moved the explanation of the grounding line and its relevance to ice shelf dynamics and ephemeral grounding earlier in the Introduction to better contextualize the discussion on measurement techniques in this paragraph (see Response 5).

**Line 59-60** (According to Friedl et al. (2020), DROT-derived grounding line positions (tidal flexure limit) were ~2 km seaward of DInSAR and ~2 km landward of H positions, where is the landward limit of stable hydrostatic equilibrium.)- This sentence is currently unclear and requires grammatical edits. It is also somewhat misleading as it implies that Friedl et al. (2020) found that DROT GLs are in general 2 km seaward of DInSAR, whereas they only tested this at a single location (Petermann Glacier) and provide considerable discussion on the larger associated errors of DROT. DInSAR GL also in theory gives the tidal flexure limit. Consider rewriting this sentence to clarify these points.

**Response 44:** We appreciate this clarification. We have revised the sentence to:

**"At Petermann Glacier, Friedl et al. (2020) found DROT-derived flexure limits ~2 km seaward of DInSAR results."**

**Line 60** (To address the knowledge gap regarding the relationship between changes of ice rumple L and ice shelf dynamics at PIIS…)- This knowledge gap should be explained more clearly earlier - perhaps at the end of the paragraph at line 46.

**Response 45:** Thank you. We have revised the paragraph to highlight the knowledge gap regarding ephemeral grounding detection and its relevance to ice shelf dynamics, providing a smoother transition into the discussion of techniques (see response 5).

**Line 62** (…particularly in the period following four recent calving events and an observed increase in ice velocity from 2014 to 2023…) - Again, the information about the velocity increases should come earlier. This is the first time the reader is hearing about this yet it is a key motivation of this study

**Response 46:** We agree that this is a key motivating factor that should be introduced sooner. We have moved the discussion of observed ice velocity increases to an earlier section in the Introduction to better frame the study's objectives (see response 5).

Line 63 (…we analyze double-differential vertical displacement derived from Sentinel-1 SAR data…)- It would be useful to state here that it is because of these reasons stated above that you are therefore using DROT instead of DInSAR in your analysis. It would be useful to then quantify the impact this has on the accuracy of your results compared to using DInSAR.

**Response 47:** We have added a statement explaining that DROT is used in this study due to its broader temporal coverage and ability to detect short-lived vertical displacements, despite its lower spatial resolution compared to DInSAR. We also now briefly discuss the trade-offs in accuracy and the implications for interpreting ephemeral grounding patterns.

The revised content is as follows:

**"In contrast, DROT provides a complementary approach that does not rely on phase information, making it useful for observing vertical tidal displacements on fast-moving ice shelves, despite being less precise than InSAR in some contexts (Marsh et al., 2013; Hogg, 2015; Joughin et al., 2016; Christianson et al., 2016; Friedl et al., 2020; Wallis et al., 2024; Lowery et al., 2025; Zhu et al., 2025)."**

Line 88 (…we first accepted pixels with an NCC greater than 0.05...)- the authors have used the acronym NCC without defining it.

**Response 48:** We have now defined the acronym NCC as "normal cross-correlation coefficient" at its first mention in the manuscript, as follows:

**"First, we retained only pixels with a normalized cross-correlation value greater than 0.05, which also used by Solgaard et al. (2021) to ensure reliable displacement measurements."**

Line 124 (…where the area around Ice rumple L exhibited near-zero displacement...)- be consistent in use of capitalisation of 'ice rumple L' / 'Ice rumple L' throughout. I would suggest no capitalisation is needed.

**Response 49:** We have revised the manuscript accordingly.

Line 134-5 (…were co-registered through the code Basal melt rates Using REMA and Google Earth Engine (BURGEE) developed by Zinck et al. (2023a, 2023b))- suggest using quotation marks for this acronym.

**Response 50:** We have added quotation marks around the acronym when first introduced to improve clarity.

**"The correction and co-registration procedures were implemented using the "Basal melt rates Using REMA and Google Earth Engine (BURGEE)" processing framework developed by Zinck et al. (2023a, 2023b)."**

Line 141 (Using uncorrected REMA DSM data, we found surface elevation 140 changes at PIIS exceeding 30 m (2010-2022); therefore, we tested elevation differences in the DEM mosaic, setting 100 m as the outlier filter criterion for REMA DSM strips and CryoSat-2 elevations.)- please clarify what is meant by 'we tested elevation differences in the DEM mosaic'.

**Response 51:** We agree that the original wording was unclear. We have revised the sentence to clarify that we assessed elevation differences between overlapping DEM strips in the mosaic to evaluate their consistency and identify any systematic offsets, as follows:

**"To identify and remove elevation outliers, we used the REMA 2 m mosaic DEM (Howat et al., 2019) as a reference surface for both the REMA 2 m strips and the CryoSat-2 data. In regions of the PIIS where uncorrected REMA strips exhibited unrealistic elevation changes exceeding 30 m, we applied a more conservative threshold of 100 m elevation difference to exclude outliers."**

Line 143 (We replaced the FES2004 ocean tide model provided by the ESA with the CATS2008 ocean tide model to provide a more accurate ocean loading tide correction value)- CATS2008 tide model is missing a citation here; should use Howard et al. (2019). Also please clarify the version (the most up-to-date is CATS2008_v2023).

**Response 52:** We acknowledge that our original citation was outdated. In the revised manuscript, we have updated the citation for the CATS2008 tidal model to Howard et al. (2024), as the update of the original CATS2008 tide model (Howard et al., 2019, https://doi.org/10.15784/601235). We also clarified that the model used is the most recent version, CATS2008_v2023. These updates are now reflected in Section 2.1, as shown below:

**"Ephemeral grounding events were compared with double-differential tidal height time series (Figure 4(b)), derived from the CATS2008_v2023 ocean tide model (Howard et al., 2024) using Tide Model Driver 3.0 (Greene et al., 2023) at (-75.186576°S, -100.617021°W)."**

Howard, S. L., Greene, C. A., Padman, L., Erofeeva, S., & Sutterley, T. CATS2008_v2023: Circum-Antarctic Tidal Simulation 2008, Version 2023, [Dataset], U.S. Antarctic Program (USAP) Data Center, https://doi.org/10.15784/601772., 2024

Line 145 (For other corrections, such as solid earth tides, geocentric polar tides, and dry and wet tropospheric and ionospheric effects, we used the data provided by the ESA, as in Zinck et al. (2023a). The erroneous elevation datasets where the interferometric cross-track location failed are filtered based on the quality flags provided by the ESA.)- it is unclear in this paragraph the source of the corrections and the dataset they are being applied to. Does 'data provided by the ESA' mean that these are the corrections provided in the CryoSat-2 dataset, which have then been applied to the REMA DSM strips? This should be clarified, with appropriate citation given to the original dataset. Note that the acronym ESA should be defined, and no 'the' is needed preceding 'ESA'.

**Response 53:** We have defined 'ESA' as the European Space Agency and removed the preceding article 'the' accordingly. The correction methods applied to the REMA strips differ from those used for the CryoSat-2 data. We have revised the paragraph as follows:

**"Dynamic and static corrections were applied to both the REMA strips and the CryoSat-2 dataset to bring all elevations into a consistent reference frame, following the methodology described by Zinck et al. (2023a). For REMA, the corrected surface elevation ($h_{corr}$) was calculated as:**

$$h_{corr} = h_{Data} - \Delta h_{Geoid} - \alpha(\Delta h_T + \Delta h_{MDT} + \Delta h_{IBE}) \tag{6}$$

**where $h_{Data}$ is the uncorrected surface elevation, $\Delta h_{Geoid}$ is the geoid offset from EGM2008 (Pavlis et al., 2012), $\Delta h_T$ is the tidal height from the CATS2008_v2023 ocean tide model (6-hour intervals, ~3 km resolution), $\Delta h_{MDT}$ is the mean dynamic topography from the DTU15MDT dataset (Andersen et al., 2015), and $\Delta h_{IBE}$ is the inverse barometer effect based on 6-hourly NCEP/NCAR sea-level pressure residuals (Kalnay et al., 1996), referenced to a mean sea level pressure of 1013 hPa. Tidal and barometric corrections were applied based on the acquisition time of the first stereo image in each DEM strip. The stereo image pairs used to generate the DEMs are typically acquired within a short time interval—usually within minutes to a few hours. Therefore, applying tidal and inverse barometric effect**

(IBE) corrections based on the acquisition time of the first image introduces only minimal temporal bias. The coefficient α ensures a smooth transition between grounded and floating ice, varying from 0 to 1 with distance from the floating ice edge to the grounding line (Shean et al., 2019), as defined by the ASAID product (Bindschadler et al., 2011a):

$$\alpha(l) = \begin{cases} 0, & l \leq 0\,km \\ \dfrac{1}{3}, & 0\,km < l \leq 3\,km \\ 1, & l > 3\,km \end{cases} \tag{7}$$

The ASAID grounding line product serves as an input to the BURGEE framework and is the same dataset used in Zink et al. (2023a).

CryoSat-2 data were similarly corrected using the same tide model and additional fields from the Level 2 SARIn product (Howard et al., 2019; Zhang et al., 2020). Erroneous elevation measurements resulting from failed interferometric cross-track positioning were excluded based on quality flags provided by European Space Agency."

Line 146-7 (We applied the dynamic and static corrections mentioned in Zinck et al. (2023a) to both the REMA strips and the CryoSat-2 elevations to bring all elevations into the same reference frame regardless of sea level variations.) - This sentence is unclear. Is this referring to the CryoSat-2 data used for the co-registration? Please clarify.

Response 54: Yes, this sentence refers to the use of CryoSat-2 elevation data for co-registering the REMA strips. We have revised this section and moved the description to the beginning of Section 2.2 for improved clarity. The updated paragraph is as follows:

"Elevation data from the CryoSat-2 Baseline-D Level 2 SARIn product (Meloni et al., 2019), spanning July 2010 to June 2022, were used to correct and co-register REMA Version 4.1 2 m resolution DSM strips acquired between October 2010 and December 2022 (Howat et al., 2022b)."

Line 155 (The grounding line product of the ASAID was used to define α.)- please define ASAID, provide the appropriate citation (Bindschadler & Choi, 2011), and explain how it was used to define α (i.e. was the distance between the grounding line and hydrostatic point used to determine the width to apply a smooth transition from grounded to floating ice?). Also note that the ASAID grounding line marks the break-in-surface slope rather than the tidal flexure limit (as it was derived using optical imagery + altimetry), so this choice should be justified (as opposed to, say, the MEaSUREs DInSAR-derived grounding line product that marks the tidal flexure limit).

Response 55: We have now defined ASAID as the Antarctic Surface Accumulation and Ice Discharge project and cited Bindschadler & Choi (2011; https://doi.org/10.5194/tc-5-569-2011) in the revised manuscript.

We used the ASAID grounding line product as the input to the BURGEE framework because it provides a continent-wide, consistent dataset that is compatible with the hydrostatic correction method employed in our analysis. Specifically, the distance between the ASAID grounding line and the nearest hydrostatic point was used to define the transition width (α), allowing for a smooth interpolation between grounded and floating ice conditions, as required in the BURGEE methodology. We acknowledge that the ASAID grounding line reflects the break-in-surface slope derived from optical imagery and altimetry, and therefore may not coincide with the tidal flexure limit. However, we chose ASAID to ensure consistency with prior studies using the BURGEE framework, such as Zink et al. (2023a).

In our analysis, we used the MEaSUREs DInSAR-derived grounding line (which captures the tidal flexure limit) for comparison when examining the location of the ice rumple, but not for hydrostatic correction. We have clarified this rationale in the revised text.

We have revised the paragraph as follows:

**"The coefficient α ensures a smooth transition between grounded and floating ice, varying from 0 to 1 with distance from the floating ice edge to the grounding line (Shean et al., 2019), as defined by the** Antarctic Surface Accumulation and Ice Discharge **(ASAID) product (Bindschadler et al., 2011a):**

$$\alpha(l) = \begin{cases} 0, l \leq 0km \\ \dfrac{1}{3}, 0km < l \leq 3km \\ 1, l > 3km \end{cases} \tag{7}$$

**The ASAID grounding line product serves as an input to the BURGEE framework and is the same dataset used in Zink et al. (2023a)."**

Line 156 (The correction for the tides and the inverse barometer effect were based on the acquisition time of the first stereo image.)- Please comment on how the choice to apply the tidal and IBE corrections based on the acquisition time of the first stereo image may or may not affect the results here.

**Response 56:** Thank you for raising this important point. We have added a statement to the manuscript addressing this issue.

**"Tidal and barometric corrections were applied based on the acquisition time of the first stereo image in each DEM strip. The stereo image pairs used to generate the DEMs are typically acquired within a short time interval—usually within minutes to a few hours. Therefore, applying tidal and inverse barometric effect (IBE) corrections based on the acquisition time of the first image introduces only minimal temporal bias. "**

Line 162 (To include smaller but good-quality REMA strips to obtain a more complete time series of freeboard ice thickness, we set the latitude and longitude criteria for the CryoSat-2 distribution to be at least 5 km for both directions.)- What does 'CryoSat-2 distribution' mean here? Please clarify this sentence.

**Response 57:** Thank you for pointing out the ambiguity in the phrase "CryoSat-2 distribution."

We have revised the paragraph and added Figure 5 to describe the co-registration criteria as follows:

**"Co-registration of REMA strips to CryoSat-2 followed a modified procedure from Zinck et al. (2023a), with the following criteria: (1) CryoSat-2 data must span at least 5 km in both latitude and longitude over the REMA strip; (2) CryoSat-2 measurements must be acquired within one month of the REMA strip acquisition; and (3) at least 75 valid CryoSat-2 measurements must be available within this spatiotemporal window. Residuals between each REMA strip and the co-registered REMA mosaic were used to apply tilt and vertical shift corrections through plane fitting. The final REMA strips are corrected and co-registered to CryoSat-2 and are referenced to the EGM2008 geoid, ensuring both high internal consistency and improved absolute accuracy."**

[Figure]

**Figure 5. Processing steps for correction of REMA, CryoSat-2, and ICESat-2 data.**

Line 167 (To evaluate the accuracy of the REMA DSM strips, we used four REMA DSM strips from 2019 to 2021 with nearly contemporaneous ICESat-2 elevations…)- Please provide some more detail on the validation using ICESat-2.

**Response 58:** We appreciate this suggestion. We have now added details in the manuscript. We have revised the paragraph as follows:

**"To assess the accuracy of the corrected REMA strips, we compared three strips from 2019–2021 with nearly contemporaneous ICESat-2 ATL06 data (Smith et al., 2019; Smith et al., 2023). The ICESat-2 elevations were converted to heights relative to the instantaneous sea surface by referencing them to the EGM2008 geoid and applying corrections for ocean tides and the inverse barometer effect, following Wang et al. (2021). At overlapping locations between the datasets, we calculated the mean elevation difference (REMA minus ICESat-2) and the standard deviation of this bias. As shown in Table 2, the corrected REMA strips exhibited lower standard deviations compared to the uncorrected data,**

**indicating reduced uncertainty. However, a consistent negative mean bias remained, with the corrected REMA elevations appearing systematically lower than those from ICESat-2.”**

Line 179 (The ice-equivalent freeboard thickness and Eulerian thickness changes were calculated through the methods provided in Griggs and Bamber (2011) and Shean et al. (2019). )- More detail should be provided about the methods applied here, so that the reader does not have to rely on reading these two referenced papers. Specifying the equation(s) used here would be useful.

**Response 59:** We appreciate this suggestion and have now added additional methodological details to the manuscript, including the specific equations used to calculate ice-equivalent freeboard thickness. This ensures the analysis is more transparent without relying solely on the referenced studies.

- **The revised Section 2.3 is as follows:**

**“To estimate changes in ice-equivalent freeboard thickness near ice rumple L, we used both the corrected REMA strips and ICESat-2 data. Specifically, ICESat-2 tracks 965 and 1094, which pass through ice rumple L, were analysed. Ice-equivalent freeboard thickness ($H_f$) was calculated using Equation (5), following the methods of Griggs and Bamber (2011) and Shean et al. (2019):**

$$H_f = h_{corr} \left( \frac{\rho_{sw}}{\rho_{sw} - \rho_{is}} \right) - h_{FAC} \qquad (8)$$

**where $h_{corr}$ is the corrected surface elevation, $\rho_{is}$ is the ice density (917 kg/m3), $\rho_{sw}$ is the seawater density (1027 kg/ m3), $h_{FAC}$ is the firn air content of ice equivalent (in meters) derived from the NASA GSFC-FDM v1.2.1 dataset (Medley et al., 2022a; 2022b), with a 5-day temporal resolution spanning from 1 January 1980 to 30 June 2022.”**

Line 182 (We used optical imagery and DSMs to derive surface changes at the PIIS.)- Define acronym

**Response 60:** The acronym “DSMs” has now been replaced as “Digital Surface Model” when first time appeared in the manuscript.

Line 184 (…Surface feature changes at the PIIS were derived from Landsat optical imagery provided by the USGS Earth Explorer and MODIS optical imagery...)- Please specify which Landsat satellite(s) were used, and provide appropriate citation.

**Response 61:** Thank you for the comment. For the optical imagery used to reveal surface ridges, we relied solely on MODIS data. For extracting ice front positions in Figure 1, we used imagery from Landsat 7 and Landsat 8 satellites. This sentence/paragraph as well as citations are added in the revised manuscript:

**Figure 1 caption:**

**“…later positions (1973-2022) are from Landsat imagery (Landsat-1/4/5/7/8/9) and Sentinel-1 SAR imagery via Google Earth Engine.”**

**Section 2.2 (last paragraph):**

**“Surface elevation changes over the PIIS were derived from the corrected REMA strips. Additionally, MODIS optical imagery from the Images of the Antarctic Ice Shelves Version 2 dataset (Scambos et al., 2022), with a spatial resolution of 250 m and spanning from 1 January 2001 to 23 October 2023, was used to identify changes in surface ridges.”**

**Section 2.4:**

**"We used Landsat-8 optical images, specifically the panchromatic band with a 15m spatial resolution, to track the rift propagation history."**

Line 185 (Surface feature changes at the PIIS were derived from Landsat optical imagery provided by the USGS Earth Explorer and MODIS optical imagery provided in the images of the Antarctic Ice Shelves Version 2 dataset (Scambos et al., 2022) at a spatial resolution of 250 m between 1 January 2001 and 23 October 2023.)- Presumably the Landsat data is at a higher resolution (30m)? As currently worded it seems like both Landsat and MODIS are 250 m.

**Response 62:** Thank you for pointing this out. We changed the wording to clarify that the Landsat panchromatic band have a spatial resolution of 15 meters, whereas the MODIS data have a resolution of 250 meters.

**Section 2.2 (last paragraph):**

**"Surface elevation changes over the PIIS were derived from the corrected REMA strips. Additionally, MODIS optical imagery from the Images of the Antarctic Ice Shelves Version 2 dataset (Scambos et al., 2022), with a spatial resolution of 250 m and spanning from 1 January 2001 to 23 October 2023, was used to identify changes in surface ridges."**

**Section 2.4:**

**"We used Landsat-8 optical images, specifically the panchromatic band with a 15m spatial resolution, to track the rift propagation history."**

Line 187 (We used three-month moving mean data from the Oceanic Niño Index (ONI) and year-round monthly mean anomaly data from the Antarctic Oscillation (AAO) index from 2014 to 2023)- This sentence should include citations for the ONI and AAO data.

**Response 63:** We thank the reviewer for noting the missing citations. We'll be more careful with figure captions and the citations in the future.

Line 190 (3.1 Changes in the Double-differential Vertical Displacement)- Please ensure that the precise acquisition time and coincident modelled tide heights for each Sentinel-1 acquisition are provided in the supplementary material.

**Response 64:** We have ensured that the exact acquisition times and coincident modelled tide heights for each Sentinel-1 image used in our analysis are now provided in the supplementary material. The description also included in Section 3.1 is as follows:

**"The tidal height difference was calculated from data extracted at a point near the ice rumple L (longitude 100.6149°W, latitude 75.1867°S), corresponding to the exact acquisition times of each Sentinel-1 image, which were at 4:35 AM on each date (Supplement Table S1)."**

Line 192 (This displacement, ranging from -2 m to 2 m closely matching the double-differential tidal height (Figure 4(b)).) – Typo, should be 'matches'

**Response 65:** Revised.

Line 192-3 (This displacement, ranging from -2 m to 2 m closely matching the double-differential tidal height (Figure 4(b)). Positive displacement anomalies correspond to high tidal phases, while negative anomalies correspond to low tidal phases.) - The way this sentence is written makes it seem that Figure 4(b) shows that the displacement matches the double differential tidal height, which it does not. Perhaps somehow the double difference tide height values on the maps in panel (a) could be labelled to show this.

**Response 66:** We have rewritten this sentence to avoid implying that panel (b) of revised Figure 6 shows a direct match with double-differential tide heights. We now clarify that panel (a) displays the double-differential tide heights, and panel (b) shows the corresponding vertical displacement patterns. We also added labels for the tide height differences in panel (a) to improve clarity (see Response 11 and 23).

The relevant paragraph is now written as:

**"Ephemeral grounding regions, characterized by double-differential vertical displacements close to zero, are influenced by oceanic tidal variations. (Figures 6-7 and Movie S1). One or two near-zero vertical displacement signals were detected at ice rumple L from at least November 2016 through April 2020, followed by a reappearance in December 2020. These signals are highlighted by yellow arrows in Figure 6a and marked by red vertical lines in Figure 6b. The reduced number of signals before ~August 2016 and after ~December 2021 likely reflects data limitations during periods when Sentinel-1B was not operational. Near-zero vertical displacement signals also occurred in 2016, 2017, and after the 2018 calving event. In December 2020, a similar signal appeared upstream of ice rumple L and progressively migrated toward the rumple, indicating that ephemeral grounding occurred as a thicker section of the ice shelf moved across the southern side of the sea ridge."**

Line 195 (Figure 4(a) illustrates this relationship, showing the floating region where negative double-differential vertical displacement indicated in blue. Movie S1 provides a more complete visualization, 195 showing both negative and positive displacement.)- Typo, missing verb

**Response 67:** Revised.

Line 197 (The floating area (exhibiting both negative and positive anomalies is enclosed by the 2011 grounding line, derived using the DInSAR method) - Presuming that the '2011 grounding line' refers to the ASAID grounding line, Bindschadler & Choi (2011) should be cited here. This GL was not derived using the DInSAR method; it was derived from a combination of optical imagery and ICESat laser altimetry, marking the break-in-surface-slope. Additionally, looking at Figure 4 it is not clear that the GL is even marked on the maps in panel (a), so please address this.

**Response 68:** We confirm that the "2011 grounding line" refers to the MEaSUREs product, and have now added a citation to Rignot et al. (2016). This product was based on DInSAR method. In addition, we have ensured that the grounding line is clearly marked on the maps in revised Figure 1, Figure 6(a) and Figure 8.

Line 197-8 (Our results are consistent with Friedl et al. (2020), who found that DROT-derived grounding zones, indicating the landward limit of tidal flexure, were located seaward (up to ~2 km) of those derived from DInSAR.)- Note that the Friedl et al. (2020) results are from one single study at Petermann Glacier. The way this is currently written implies that a more general/widespread relationship between DROT and DInSAR GLs has been shown in this review paper, which it does not. I suggest making this clearer in the text. Also (as commented above) both DROT and DInSAR GLs in theory indicate the landward limit of tidal flexure.

**Response 69:** We agree and have revised the text to reflect that the Friedl et al. (2020) study is based on a single glacier (Petermann Glacier), and does not establish a general relationship between DROT and DInSAR GLs. We have revised the paragraph as follows:

**"Satellite remote sensing can effectively detect transient vertical motion of ice shelves, especially tidal fluctuations that cause ephemeral grounding. Key methods include differential range offset tracking (DROT) (Marsh et al., 2013; Joughin et al., 2016; Christianson et al., 2016; Wallis et al., 2024, 2025; Lowery et al., 2025; Zhu et al., 2025), interferometric synthetic aperture radar (InSAR) (Schmeltz et al., 2001; Rignot, 2002, 2014), and satellite altimetry (Fricker and Padman, 2006). Both DROT and InSAR methods in theory indicate the landward limit of tidal flexure. While InSAR is widely used to map grounding line migration, its effectiveness is limited in fast-flowing areas due to phase aliasing unless very short repeat intervals are available. For instance, Milillo et al. (2017) used 1-day repeat COSMO-SkyMed data to track grounding line changes at PIIS.**

**In contrast, DROT provides a complementary approach that does not rely on phase information, making it useful for observing vertical tidal displacements on fast-moving ice shelves, despite being less precise than InSAR in some contexts (Marsh et al., 2013; Hogg, 2015; Joughin et al., 2016; Christianson et al., 2016; Friedl et al., 2020; Wallis et al., 2024; Lowery et al., 2025; Zhu et al., 2025). Using TerraSAR-X data, Joughin et al. (2016) identified a vertical displacement anomaly near ice rumple L and estimated the grounding line position to within ~1.5 km. At Petermann Glacier, Friedl et al. (2020) found DROT-derived flexure limits ~2 km seaward of DInSAR results. More recently, DROT applied to Sentinel-1 IW data has proven effective for studying grounding line and pinning point dynamics on the Antarctic Peninsula (Wallis et al., 2024), Amery Ice Shelf (Zhu et al., 2025), and PIIS (Lowery et al., 2025). However, Lowery et al. (2025) focused only on the year 2017, leaving later changes unresolved. Thus, the evolution of grounding behaviour at ice rumple L following four subsequent calving events—in 2015, 2017, 2018, and 2020—remains poorly understood."**

Line 198 (Our results are consistent with Friedl et al. (2020), who found that DROT-derived grounding zones, indicating the landward limit of tidal flexure, were located seaward (up to ~2 km) of those derived from DInSAR.) - You should use 'grounding line' here (not 'grounding zone'). Be careful with use of line vs zone throughout.

**Response 70:** Thank you for pointing this out. We agree with you that the original sentence here should use grounding line here. We have reviewed the manuscript to ensure consistent and appropriate use of the terms "grounding line," "grounding zone," and "grounding region." In the manuscript, we use "grounding region" to refer to the area where double-differential vertical displacement values are close to zero in Figure 6(a).

The relevant paragraph is now written as:

**"Ephemeral grounding regions, characterized by double-differential vertical displacements close to zero, are influenced by oceanic tidal variations. (Figures 6-7 and Movie S1)."**

Line 256 (…which allowed for potential re-grounded of sub-ice-shelf keels)- Typo, should be 're-grounding'

**Response 71:** Corrected.

Line 262 (A local surface low/high in grounded ice indicates a topographic low/high in the bed, forming surface troughs/ridges…) - I suggest re-wording to avoid the use of opposites in the sentence (e.g., 'low/high' and 'troughs/ridges').

**Response 72:** We have rewritten this section. The relevant paragraph is now:

**"Figure 8 shows the evolution of surface ridges and their elevations from December 2010 to January 2021. Some ridges higher than 75m were advected from upstream and passed through the area near the ice rumple L (Figure 8b-k). Near ice rumple L (red point in Figure 8), surface elevations remained**

**around ~65 m between 2012–2017 and again during 2019–2020 (Figures 8d–h and 8j–k). The highest elevation (~85 m) was recorded in 2018, while the lowest (~54 m) occurred in 2021. Between 2020 and 2021, surface elevation declined by ~10 m, equivalent to ~70 m of ice-equivalent freeboard thickness. The area enclosed by the grounding line, corresponding to the region of zero vertical displacement, was the largest in 2018 (Figure 8i)."**

Line 278 (In general, positive phases of the SAM are more likely to occur during La Niña) - Please define SAM acronym and provide a citation

**Response 73:** The acronym SAM stands for "South Antarctic Mode". As we removed the discussion on atmospheric forcing, SAM is not mentioned any longer.

Line 285 (As mentioned by Bradley et al. (2022), when the ice front of the PIG retreated from the 2009 position to the 2020 position, melt rates within 10 km of the ice front increased significantly; the melt response to calving was sensitive to the thickness of the gap between the ice shelf and the seabed ridge.) - This sentence suggests that the increase in PIG melt rates during this time were observed, when in fact this is results from a modelling study. This should be made clear in this sentence.

**Response 74:** Thank you for pointing this out. We have removed the section related to basal melting and have carefully revised the description of the relationship between basal melting and ephemeral grounding in Section 4.

**"De Rydt et al. (2014) demonstrate that both the height of the ridge and the gap between the ridge and the ice shelf strongly influence the inflow of warm bottom waters into the cavity, and consequently, the melt rates. The melt rate may influence the ice thickness near to the grounding line upstream than the ice rumples K and L. This process may have contributed to the ice thickness changes upstream and indirect influence the disappearance of ephemeral grounding signals following the 2020 calving event. We have added further analysis on the basal melt rate and ocean temperature in the Appendix A. Although smaller-scale basal channels and keel geometries are primarily shaped by melt-driven processes (Bindschadler et al., 2011b; Dutrieux et al., 2013; Stanton et al., 2013; Dutrieux et al., 2014b; Joughin et al., 2016), the lack of direct, high-temporal-resolution basal melt rate measurements after 2020 limits our ability to capture short-lived grounding events and confirm the role of ocean-driven melting. Future work should prioritize the integration of dense time series from new SAR missions and in situ oceanic data to better resolve ephemeral grounding behaviour and its implications for ice shelf evolution and calving dynamics in a warming climate."**

Line 325 (Ground line products are available from Rignot et al. (2016), Floricioiu et al. 325 (2021), and Mohajerani et al. (2021).) - Typo, should be 'Grounding line products'. Also missing a reference for the ASAID grounding line product - Bindschadler & Choi (2011).

**Response 75:** The typo has been corrected to "Grounding line products," and we have added the missing reference for the ASAID grounding line dataset as follows: "**Grounding line products are available from Bindschadler et al. (2011a), Rignot et al. (2016), and this study (Qian et al., 2025a).**"

---

## Referee Report (RR1)

**Summary**

The authors have gone to great lengths to respond to the comments made previously, and the manuscript is much improved as a result. The authors have now justified most of their claims, and the presentation of their work is much clearer.

There remain many grammatical errors across the manuscript that need to be corrected.

**General comments on the results and discussion**

The authors contradict themselves in lines 215-217 ('Ephemeral grounding region, characterised by double-differential vertical displacements close to zero, shows significant correlation with oceanic tidal variations') and lines 235-237 ('Figure 7b reveals no clear correlation between tidal height and grounding region area').

- From 7b it looks like there is little correlation. Perhaps worth doing a simple correlation between the two to test this.
- Can you colour the points in 7b to match the spring and neap tides in Figure 7d, which might make the difference in the 2 categories clearer
- Can you explain the groundings during negative tide heights?

It is hard to interpret the elevation change at L in Figure 8. Perhaps simply showing a time series of elevation in the vicinity of L would be easier to interpret, especially as the authors don't make use of the high-frequency elevation variations visible in this figure. I suggest this time series could even be overlaid on Figure 9b.

The authors fail to reference Joughin et al. 2021 at multiple points throughout the manuscript. This paper also showed rifts propagating from the ephemeral grounding point in the centre of PIG.

The authors discussion regarding ice shelf basal melt rates remains unsatisfactory. Firstly, the lack of melt observations after 2020 doesn't limit the ability to observe grounding events (lines 342-345). The melt data presented shows a decrease in melt rates on the order of 10m/yr in 2015. If the ice is in hydrostatic equilibrium, this would be seen as a ~1m change in the surface elevation. Whereas the IceSat-2 observations presented suggest up to a 70m change in thickness and the elevation data shows a ~20m change. It is clear from these approximate numbers that there must be another process involved in the ephemeral grounding here.

Further, the authors don't discuss the potential role of the thick column of ice that was advected downstream in the ephemeral grounding. They do mention this in their introduction (lines 135-137).

The authors discuss melt rates and ocean mooring data within the context of ice shelf thinning in Appendix A. In its current form, it is unclear what this section contributes to the main narrative. While a rigorous analysis could potentially reveal a meaningful link between ocean conditions, basal melt rates and ephemeral grounding, the present version lacks sufficient clarity and detail. I recommend either substantially refining this section with a more thorough and transparent analysis or removing it altogether. In particular:

- What depths in the ocean mooring profile have the authors used to construct the time series?
- The authors claim that the decrease in melt in 2015 corresponds with decreased temperatures in the mooring (lines 394-396). To me, it looks like the cooling occurs after the peak in reduced melting.
- Joughin et al., 2021 showed that there was a 12% acceleration of the ice shelf between 2017 and 2020. It is likely this has contributed to some of the observed thinning through divergence.

**Specific Comments**

Line 1 abstract: 'The evolution of ephemeral grounding in ice shelf can...' should read 'The evolution of ephemeral grounding of an ice shelf can...'

Lines 8-9: Please provide a citation.

Lines 23-24: 'Despite the grounding line retreat, the Pine Island Ice Shelf (PIIS) was observed to maintain intermittent contact with the bathymetric high when thick ice column being advected from...' should read 'Despite the grounding line retreat, the Pine Island Ice Shelf (PIIS) was observed to maintain intermittent contact with the bathymetric high when a thick ice column was advected from....'

Line 136: 'The grounding of ice shelf' should read 'The grounding of an ice shelf...'

Line 137: remove 'as an obstacle against ice flow' and replace with 'by'

Line 205: 'Our results reveal recurring of ephemeral grounding...' should read 'our results reveal recurring ephemeral grounding...'

Line 323: 'These signals disappeared during the 2020-2021 thinning period but reappeared in December 2020'. This doesn't make sense. Please check the dates.

Lines 330-331: 'Notably, we find the rift that...' Joughin et al., (2021) suggested this. Should be cited.

Line 337: De Rydt et al., (2014) results don't seem particularly relevant to anything discussed in the following paragraph.

Line 338: Warm waters at the bottom of the ocean here should be referred to as Circumpolar Deep Water.

Lines 358-360: 'Observed large-scale surface and basal structures... basal melt modulated by ocean temperature variability'. This sentence is confused and certainly doesn't belong in the conclusion.

Figure 7 :
- Colour the points in b depending on the spring or neap tide

Figure 8:
- Here, the reader is interested in elevation change over time. Showing the elevation anomaly from some reference might make the author's point clearer.
- Furthermore, a simple time series might be easier to interpret.

Figure 9:
- I don't think we learn anything extra from having the across transect profiles from both the right and left ICESat-2 beams
- A single colour ramp that spans c-f would help the reader compare observations on different tracks
- Could the areas along the transect that Sentinel-1 detects grounding be shown?

---

## Author Response (AR2)

**Response to the review of "Ephemeral grounding on the Pine Island Ice Shelf, West Antarctica, from 2014 to 2023," submitted to The Cryosphere.**

**Editor:**

We sincerely thank Editor and Reviewers for the thoughtful review and constructive feedback on the figures. In response, we have improved the visualization of all figures to enhance clarity and presentation.

We recognize that the Appendix section had a weak connection to the main narrative. To maintain a clear and focused presentation centered on our most robust results, we decided to remove this section.

In this document, we provide a detailed, point-by-point response to all comments. The reviewer's comments are shown in black, and our responses are shown in blue, with proposed new text in bold. Some sentences were underlined to emphasize key content closely related to the comments.

Dear authors,

Many thanks for the update manuscript. As you can see in their reports, that referees still have identified some outstanding issues. Most of these are about the figures, but some also about the results and interpretation. I invite you to submit a revised manuscript, which I will review.

Referee 2 provided a list of comments on the figures. Please implement all the suggested changes and add explanation where requested by R2. Likewise, implement all the suggestions made by R1 concerning figures. If you decide to keep both the right and left ICESat-2 beams in figure 9, please motivate your choice in the rebuttal. Please also show the areas along the transect that Sentinel-1 detects grounding, if data is available there.

**Response 1:** We have removed Figures 9c and 9e from the original Figure 9, as suggested by R1. In the revised Figure 9c, we added the ephemeral grounding sites on each track to show the locations detected by ICESat-2.

[Figure]

**Figure 9. Time series of mean ice-equivalent freeboard thickness and ice shelf bottom elevation profiles along ICESat-2 tracks 965 and 1094. (a) ICESat-2 tracks 965 and 1094 that used for ice-equivalent freeboard thickness change analysis and grounding lines near the ice rumple L from April 2011 to February 2021. Background is from Landsat-8 OLI optical image on 3 March 2019. (b) Time series of mean ice-equivalent freeboard thickness (2010–2022). Mean ice-equivalent freeboard thickness from ICESat-2 was calculated along tracks 965 and 1094 between 75.15°S and 75.05°S, representing the average of measurements from both the strong and weak ICESat-2 beams. REMA thickness values were sampled at the same locations as the ICESat-2 tracks. (c) Ice shelf bottom elevation profiles along ICESat-2 track 965 gt2r between February 2020 and May 2022. (d) Ice shelf bottom elevation profiles along ICESat-2 track 1094 gt3r between June 2019 and June 2022. Bed elevations are from the BedMachine v3 dataset (Morlighem et al., 2020; Morlighem, 2022), converted from EIGEN-6C4 to the EGM2008 geoid to match the vertical datum of REMA strips. The estimated vertical uncertainty is ±200 m (shown as a grey transparent box). The potential actual bed elevation is marked by a red dashed line.**

Regarding the first comment of R1, I agree that this sentence is unclear: "Ephemeral grounding region, characterized by double-differential vertical displacements close to zero, shows significant correlation with oceanic tidal variations (Figures 6-7 and Movie S1)." Specifically, it is unclear what is correlated with the ocean tidal variations, since 'Ephemeral ground region' is not a quantity that can be correlated to something else.

**Response 2:** Thank you and R1 for pointing this out. Based on our results, we found out that there is no relationship between the area of grounding region and tidal variation. Thus, the sentence on lines 215–217 has been revised to: **"Figures 6-7 and Movie S1 show the two-dimensional double-differential vertical displacement changes and time series of double-differential tidal height differences."**

On line 235-237, add the correlation value. Related to the point above, please p-values for whenever you give correlation values.
**Response 3:** Revised. We have revised the content in Section 3.1 paragraph 2:

 "……**Figure 7 identifies 80 ephemeral grounding events between September 2016 and October 2021, including 43 during spring tides (red points) and 37 during neap tides (blue points). As shown in Figures 6a and 7a, positive vertical-displacement anomalies generally coincide with negative tidal-height differences, and vice versa, indicating a strong inverse linear relationship between these variables (r = −0.80, p = 2.41 × 10-19 < 0.05, R² = 0.65). In contrast, Figure 7b shows no significant relationship between tidal-height difference and area of grounding region (r = −0.02, p = 0.887 > 0.05, R² = 0.00026). During spring tides, only three grounding events exceeded an area of 100 km², while all other events remained below this threshold. No significant linear relationship is observed between tidal-height difference and area of grounding region during spring tides (r = −0.11, p = 0.484 > 0.05, R² = 0.012). Similarly, during neap tides, area of grounding region ranges from 0 to 90 km² and show no significant dependence on tidal-height difference (r = 0.07, p = 0.694 > 0.05, R² = 0.004)..…** "

Include references to Joughin, 2021 where appropriate.

**Response 4:** We have added the reference to Joughin et al. (2021) in Section 3.3, last sentence:

**"……Together, these observations support Joughin et al. (2021) in suggesting that ephemeral grounding events facilitate rift propagation and thereby indirectly influence the calving process of the ice shelf."**

We also included it in the Discussion (paragraph 4):

**"……Our findings support Joughin et al. (2021) in suggesting that ephemeral grounding is linked to ice-shelf rift propagation……"**

R1 is correct that the lack of melt observations after 2020 doesn't limit the ability to observe grounding events, since this is based on satellite observations. Please adjust.

**Response 5:** We have removed this sentence.

Furthermore, I agree that the discussion of the melt rates can be strengthened by adding a more thorough and transparent analysis, and a discussion of other processes that can explain the discrepancies between the melt rates and elevation changes.

**Response 6:** We agree that the discussion of melt rates could be strengthened with a more thorough and transparent analysis, including consideration of other processes that might explain the discrepancies between melt rates and elevation changes. However, to maintain a clear and focused presentation centred on our most robust results, we have decided to remove this section.

Line 339: "The melt rate may influence the ice thickness near to the grounding line upstream than the ice rumples K and L": Is there a word missing before 'than' - more/less than, for example? This sentence is unclear to me, please rephrase.

**Response 7:** We apologize for the lack of clarity. We meant to say that the melt rate may influence ice thickness near the grounding line along the main trunk. Since we have removed the content related to melt rate, this part may no longer be visible in the revised manuscript.

Fig A: add the locations of the PIG-S and PIG-N station in the Landsat image.

**Response 8:** Since we have removed the content related to melt rate, this part may no longer be visible in the revised manuscript.

In the caption, make clear over which area the 'Time series of mean ice-equivalent freeboard thickness' are calculated.

**Response 9:** Revised. The description has been added to the caption of Figure 9b.

**Figure 9b. Time series of mean ice-equivalent freeboard thickness (2010–2022). Mean ice-equivalent freeboard thickness from ICESat-2 was calculated along tracks 965 and 1094 between 75.15°S and 75.05°S, representing the average of measurements from both the strong and weak ICESat-2 beams. REMA thickness values were sampled at the same locations as the ICESat-2 tracks.**

In all the figures containing ICESat-2 and REMA data, make sure that the figure legend is correct (there is no ICESat-2 data before 2018).

**Response 10:** Revised. See Response 1.

In the abstract, add a short explanation of what 'ephemeral grounding' is, for non-experts.

**Response 11:** Revised. We have added a brief explanation of ephemeral grounding in the first sentence of the abstract.

**"Ephemeral grounding refers to the intermittent contact between an ice shelf and elevated seafloor features……"**

**Response to the review of "Ephemeral grounding on the Pine Island Ice Shelf, West Antarctica, from 2014 to 2023," submitted to The Cryosphere.**

**Reviewer #1:**

We sincerely thank Reviewer 1 for the thoughtful review and constructive feedback on the figures. In response, we have improved the visualization of all figures to enhance clarity and presentation.

We recognize that the Appendix section had a weak connection to the main narrative. To maintain a clear and focused presentation centered on our most robust results, we decided to remove this section.

In this document, we provide a detailed, point-by-point response to all comments. The reviewer's comments are shown in black, and our responses are shown in blue, with proposed new text in **bold**. Some sentences were underlined to emphasize key content closely related to the comments.

**Summary**

The authors have gone to great lengths to respond to the comments made previously, and the manuscript is much improved as a result. The authors have now justified most of their claims, and the presentation of their work is much clearer.

There remain many grammatical errors across the manuscript that need to be corrected.

We appreciate you bringing these errors to our attention, and the necessary corrections have been made.

**General comments on the results and discussion**

The authors contradict themselves in lines 215-217 ('Ephemeral grounding region, characterised by double-differential vertical displacements close to zero, shows significant correlation with oceanic tidal variations') and lines 235-237 ('Figure 7b reveals no clear correlation between tidal height and grounding region area').

**Response 1:** Thank you for pointing this out. The sentence on lines 215–217 has been revised to: **"Figures 6-7 and Movie S1 show the two-dimensional double-differential vertical displacement changes and time series of double-differential tidal height differences."**

- From 7b it looks like there is little correlation. Perhaps worth doing a simple correlation between the two to test this.

**Response 2:** We performed a simple correlation analysis and found a weak inverse linear relationship. Accordingly, we have revised the content describing Figure 7.

**Section 3.1 paragraphs 2 and 3:**

**"Figure 7 identifies 80 ephemeral grounding events between September 2016 and October 2021, including 43 during spring tides (red points) and 37 during neap tides (blue points). As shown in Figures 6a and 7a, positive vertical-displacement anomalies generally coincide with negative tidal-height differences, and vice versa, indicating a strong inverse linear relationship between these variables (r = −0.80, p = 2.41 × 10⁻³¹ < 0.05, R² = 0.65). In contrast, Figure 7b shows no significant relationship between tidal-height difference and grounding-region area (r = −0.02, p = 0.887 > 0.05, R² = 0.00026). During spring tides, only three grounding events exceeded an area of 100 km², while all other events remained below this threshold. No significant linear relationship is observed between tidal-height difference and**

**grounding-region area during spring tides (r = −0.11, p = 0.484 > 0.05, R² = 0.012). Similarly, during neap tides, grounding areas range from 0 to 90 km² and show no significant dependence on tidal-height difference (r = 0.07, p = 0.694 > 0.05, R² = 0.004).**

**Figures 7c and 7d further show no significant relationship between tidal height and area of grounding, indicating that area variability is not solely governed by tidal forcing. Notably, area of grounding region increased from December 2016 to February 2019 and decreased thereafter. When combined with Figure 6a, where near-zero double-differential vertical-displacement signals suggest the upstream advection of thicker ice, these observations indicate that ice-dynamical processes likely play a substantial role in driving ephemeral grounding."**

[Figure]

**Figure 7. Comparison of tidal height differences with double-differential vertical displacement, comparison of tidal height differences and area of grounding region, including time series of area and tidal height variations. (a) Scatter plot of tidal height difference versus double-differential vertical displacement, showing a strong negative linear correlation between the two variables (r = −0.80, p = 2.41×10⁻¹⁹ < 0.05, R² = 0.65). (b) Scatter plot of tidal height versus area of zero vertical displacement region, indicating weak relationship between the two datasets (r = −0.02, p = 0.887 > 0.05, R² = 0.00026). (c) Time series of changes in ice rumple area. (d) Time series of tidal height changes, where 0 represents**

**mean sea level. In all panels, blue vertical lines or points indicate ephemeral grounding events during the neap tide period, while red vertical lines or points represent those during the spring tide period.**

- Can you colour the points in 7b to match the spring and neap tides in Figure 7d, which might make the difference in the 2 categories clearer

**Response 3:** Revised. See Response 2.

- Can you explain the groundings during negative tide heights?

**Response 4:** In Figure 7d, time series of tidal height changes are shown, where 0 represents mean sea level. Negative tidal heights indicate that the water level is below the mean sea level. During these periods, the ice shelf can contact with the bed, leading to ephemeral grounding. Grounding at negative tide heights occurs because the lower water level reduces buoyancy, allowing the ice bottom to touch the seafloor.

It is hard to interpret the elevation change at L in Figure 8. Perhaps simply showing a time series of elevation in the vicinity of L would be easier to interpret, especially as the authors don't make use of the high-frequency elevation variations visible in this figure. I suggest this time series could even be overlaid on Figure 9b.

**Response 5:** We have removed the elevation label and included the ice thickness changes at L in Figure 9b, as suggested.

[Figure]

**Figure 8. Changes in surface ridges at PIIS near ice rumple L. (a) Overview map showing the subregion outlined by the red frame, corresponding to panels (b) to (l). (b)–(l) Surface ridges and their elevation changes from 2010 to 2021, derived from corrected REMA strips. The two black circles indicate the positions of ice rumples. Grounding lines are delineated based on the zero-contour of the double-differential vertical displacement. Grey lines are the 80m contour line. The red point in panel (b) marks the location where the thickness time series near Rumple L was extracted in Figure 9b.**

[Figure]

**Figure 9. Time series of mean ice-equivalent freeboard thickness and ice shelf bottom elevation profiles along ICESat-2 tracks 965 and 1094. (a) ICESat-2 tracks 965 and 1094 that used for ice-equivalent freeboard thickness change analysis and grounding lines near the ice rumple L from April 2011 to February 2021. Background is from Landsat-8 OLI optical image on 3 March 2019. (b) Time series of mean ice-equivalent freeboard thickness (2010–2022), representing the average of measurements from both the strong and weak ICESat-2 beams. Mean ice-equivalent thickness from ICESat-2 was calculated along tracks 965 and 1094 between 75.15°S and 75.05°S. REMA thickness values were sampled at the same locations as the ICESat-2 tracks. (c) Ice shelf bottom elevation profiles along ICESat-2 track 965 gt2r between February 2020 and May 2022. (d) Ice shelf bottom elevation profiles along ICESat-2 track 1094 gt3r between June 2019 and June 2022. Bed elevations are from the BedMachine v3 dataset (Morlighem et al., 2020; Morlighem, 2022), converted from EIGEN-6C4 to the EGM2008 geoid to match the vertical datum of REMA strips. The estimated vertical uncertainty is ±200 m (shown as a grey transparent box). The potential actual bed elevation is marked by a red dashed line.**

The authors fail to reference Joughin et al. 2021 at multiple points throughout the manuscript. This paper also showed rifts propagating from the ephemeral grounding point in the centre of PIG.

**Response 5:** Thank you for pointing this out. We have added the reference to Joughin et al. (2021) in Section 3.3, last sentence:

**"…Together, these observations support Joughin et al. (2021) in suggesting that ephemeral grounding events facilitate rift propagation and thereby indirectly influence the calving process of the ice shelf."**

We also included it in the Discussion (paragraph 4):

**"…Our findings support Joughin et al. (2021) in suggesting that ephemeral grounding is linked to ice-shelf rift propagation…"**

The authors discussion regarding ice shelf basal melt rates remains unsatisfactory. Firstly, the lack of melt observations after 2020 doesn't limit the ability to observe grounding events (lines 342-345). The melt data presented shows a decrease in melt rates on the order of 10m/yr in 2015. If the ice is in hydrostatic equilibrium, this would be seen as a ~1m change in the surface elevation. Whereas the IceSat-2 observations presented suggest up to a 70m change in thickness and the elevation data shows a ~20m change. It is clear from these approximate numbers that there must be another process involved in the ephemeral grounding here.

**Response 6:** Thank you for pointing this out. We have deleted the sentences in lines 342-345.

Further, the authors don't discuss the potential role of the thick column of ice that was advected downstream in the ephemeral grounding. They do mention this in their introduction (lines 135-137).

**Response 7:** Thank you for pointing this out. We have revised the sentence to: **"Ephemeral grounding could be driven by tidal cycles, ice shelf thinning or thickening, sea-level rise, sea-level fall, and the downstream advection of thicker ice column—depending on prior grounding conditions (Schmeltz et al., 2001; Rignot, 2002; Matsuoka et al., 2015).".**

The authors discuss melt rates and ocean mooring data within the context of ice shelf thinning in Appendix A. In its current form, it is unclear what this section contributes to the main narrative. While a rigorous analysis could potentially reveal a meaningful link between ocean conditions, basal melt rates and ephemeral grounding, the present version lacks sufficient clarity and detail. I recommend either substantially refining this section with a more thorough and transparent analysis or removing it altogether. In particular:

**Response 8:** Thank you for pointing this out. We have deleted the content relate to the basal melting to maintain a clear and focused presentation centered on our most robust results.

- What depths in the ocean mooring profile have the authors used to construct the time series?

**Response 9:** The depth we used to construct the time series is at 648 m to 657 m at the PIG-N and at 665 m to 690 m at the PIG-S to show the temperature changes near where the top of the mCDW is (~600m). Since we have removed the content related to basal melting, this change may not be visible in the revised manuscript.

- The authors claim that the decrease in melt in 2015 corresponds with decreased temperatures in the mooring (lines 394-396). To me, it looks like the cooling occurs after the peak in reduced melting.

**Response 10:** Thank you for pointing this out. We have plotted the time series of basal melting, ocean temperature, and ice thickness together for comparison (Figure R1). As you noted, changes in ocean temperature do not align with variations in basal melting. Thus, we delete the related content.

[Figure]

Figure R1. Time series of basal melting, ocean temperature, and ice thickness.

- Joughin et al., 2021 showed that there was a 12% acceleration of the ice shelf between 2017 and 2020. It is likely this has contributed to some of the observed thinning through divergence.

**Response 11:** We agree ice-shelf acceleration between 2017 and 2020 likely contributed to some of the observed thinning through divergence. However, the processes driving ice-shelf thinning are complex. To maintain a clear narrative, we have chosen to focus on the robust results—such as thinning, thickening, and rift propagation—while a detailed analysis of the causes of thinning is beyond the scope of this study.

**Specific Comments**

Comments related to grammar or word choice:

Line 1 abstract: 'The evolution of ephemeral grounding in ice shelf can...' should read 'The evolution of ephemeral grounding of an ice shelf can...'

Lines 23-24: 'Despite the grounding line retreat, the Pine Island Ice Shelf (PIIS) was observed to maintain intermittent contact with the bathymetric high when thick ice column being advected from...' should read 'Despite the grounding line retreat, the Pine Island Ice Shelf (PIIS) was observed to maintain intermittent contact with the bathymetric high when a thick ice column was advected from....'

Line 136: 'The grounding of ice shelf' should read 'The grounding of an ice shelf...'

Line 137: remove 'as an obstacle against ice flow' and replace with 'by'

Line 205: 'Our results reveal recurring of ephemeral grounding...' should read 'our results reveal recurring ephemeral grounding...'

**Response 12:** All comments related to grammar or word choice have been revised as suggested.

Lines 8-9: Please provide a citation.

**Response 13:** Revised to **"A prominent example of these dynamics can be seen in the Amundsen Sea sector of West Antarctica, which accounts for over 31% of the continent's total ice loss (Smith et al.,**

**2020). Within this sector, the Pine Island Glacier (PIG) basin alone contributed approximately 3.0 mm to global sea-level rise between 1979 and 2017 (Rignot et al., 2019).”**

Reference:

Smith, B., Fricker, H. A., Gardner, A. S., Medley, B., Nilsson, J., Paolo, F. S., et al.: Pervasive ice sheet mass loss reflects competing ocean and atmosphere processes, Science, 368, 1239–1242, https://doi.org/10.1126/science.aaz5845, 2020.

Line 323: 'These signals disappeared during the 2020-2021 thinning period but reappeared in December 2020'. This doesn't make sense. Please check the dates.

**Response 14**:This sentence has been revised to: **"These signals disappeared between March and December 2020, during the ice shelf thinning period from 2020 to 2021."**

Lines 330-331: 'Notably, we find the rift that…' Joughin et al., (2021) suggested this. Should be cited.

**Response 15**:This sentence has been revised to: "**Notably, we find the rift that caused the 2020 calving event appeared after pass through the ephemeral grounding region. Our findings support Joughin et al. (2021) in suggesting that ephemeral grounding is linked to ice-shelf rift propagation…….**".

Line 337: De Rydt et al., (2014) results don't seem particularly relevant to anything discussed in the following paragraph.

**Response 16**:Removed.

Line 338: Warm waters at the bottom of the ocean here should be referred to as Circumpolar Deep Water.

**Response 17**:Thank you for pointing this out. Since we have removed the content related to basal melting, this change may not be visible in the revised manuscript.

Lines 358-360: 'Observed large-scale surface and basal structures… basal melt modulated by ocean temperature variability'. This sentence is confused and certainly doesn't belong in the conclusion.

**Response 18**:Removed.

Figure 7 : - Colour the points in b depending on the spring or neap tide

**Response 19**:Revised. See Response 2.

Figure 8:

- Here, the reader is interested in elevation change over time. Showing the elevation anomaly from some reference might make the author's point clearer. Furthermore, a simple time series might be easier to interpret.

**Response 20**:Revised. See Response 5.

Figure 9:

- I don't think we learn anything extra from having the across transect profiles from both the right and left ICESat-2 beams

**Response 21**:Revised. We have removed the original Figures 9c and 9e, which contained results from the weak beams (see Response 5).

- A single colour ramp that spans c-f would help the reader compare observations on different tracks

**Response 22:** We decided to retain the color ramp to highlight the ephemeral grounding locations along the ICESat-2 tracks in Figure 9a. Since the two ICESat-2 tracks correspond to different dates, using different colors helps to make this distinction clearer.

- Could the areas along the transect that Sentinel-1 detects grounding be shown?

**Response 23:** Yes. We plotted the grounding regions in Figure 9c as dots that cover the ICESat-2 tracks, using the same colors as the lines in Figure 9d.

**Response to the review of "Ephemeral grounding on the Pine Island Ice Shelf, West Antarctica, from 2014 to 2023," submitted to The Cryosphere.**

Reviewer #2:

We sincerely thank Reviewer 2 for the thoughtful review and constructive feedback on the figures. In response, we have improved the visualization of all figures to enhance clarity and presentation.

In this document, we provide a detailed, point-by-point response to all comments. The reviewer's comments are shown in black, and our responses are shown in blue, with proposed new text in **bold**.

Thanks for providing substantial revisions on the manuscript and I think the overall quality has improved a lot. Please find my comments below.

Line 292: It should be 27 November 2018 according to Figure 10g.

**Response 1:** After rechecking the date of the Landsat-8 image, we confirm that the image used is from 11 December 2018. The date originally labeled in Figure 10g was incorrect. We have now corrected it.

In Figure 7, why does the number of ephemeral grounding events marked by blue and red vertical lines in 7d not match the number of red vertical lines representing the ice rumple area in 7c? The area of grounding zone should exceed zero if there is a grounding event.

**Response 2:** We originally applied the MATLAB function **graythresh** to determine the threshold for identifying the grounding-line region. However, this function only operates on positive values, causing incorrect extraction for double-differential vertical displacement maps containing negative values. Thus, we correct the script and produced correct result.

Previously, the automatically detected grounding areas were smaller than those identified manually from the 2D double-differential displacement maps (i.e., the white "spots"). In the revised approach, we first extract the ephemeral grounding regions using the Otsu method, then compare the results with the 2D displacement maps and remove noise or ambiguous detections. Using this method, we identify 80 ephemeral grounding events during the observation period. We have also revised the relevant figure and ensured the number of vertical lines in Figures 7c and 7d is consistent.

[Figure]

**Figure 7. Comparison of tidal height differences with double-differential vertical displacement, comparison of tidal height differences and area of grounding region, including time series of area and tidal height variations. (a) Scatter plot of tidal height difference versus double-differential vertical displacement, showing a strong negative linear correlation between the two variables (r = –0.80, p = 2.41×10⁻¹⁹ < 0.05, R² = 0.65). (b) Scatter plot of tidal height versus area of zero vertical displacement region, indicating weak relationship between the two datasets (r = –0.02, p = 0.887 > 0.05, R² = 0.00026). (c) Time series of changes in ice rumple area. (d) Time series of tidal height changes, where 0 represents mean sea level. In all panels, blue vertical lines or points indicate ephemeral grounding events during the neap tide period, while red vertical lines or points represent those during the spring tide period.**

In Figure 8, please remove the labels 'Surface elevation xxx' from the plots and put them somewhere else, currently they overlap the surface features and make the figures rather messy.

**Response 3:** Revised. We moved the surface-change results to Figure 9b and replaced elevation with thickness.

[Figure]

**Figure 8. Changes in surface ridges at PIIS near ice rumple L. (a) Overview map showing the subregion outlined by the red frame, corresponding to panels (b) to (l). (b)–(l) Surface ridges and their elevation changes from 2010 to 2021, derived from corrected REMA strips. The two black circles indicate the positions of ice rumples. Grounding lines are delineated based on the zero-contour of the double-differential vertical displacement. Grey lines are the 80m contour line. The red point in panel (b) marks the location where the thickness time series near Rumple L was extracted in Figure 9b.**

In Figure 9b, it is impossible to quickly distinguish ice thickness measured from these two different ICESat-2 tracks due to color choices. Please change to a different color scheme, and I suggest making line plots for each different ground track measurement. Does 'REMA DEM gt2l' mean the REMA DEM sampled at gt2l ground locations?

**Response 4:** Yes, "REMA DEM gt2l" refers to REMA DEM values sampled at the gt2l ground locations. We have revised both the description and the labels. We also updated the color scheme and changed the dot plot to a line plot with markers, averaging values from both tracks (e.g., Track 965 represents the mean of gt2l and gt2r; Track 1094 represents the mean of gt3l and gt3r). This modification improves clarity.

[Figure]

**Figure 9. Time series of mean ice-equivalent freeboard thickness and ice shelf bottom elevation profiles along ICESat-2 tracks 965 and 1094. (a) ICESat-2 tracks 965 and 1094 that used for ice-equivalent freeboard thickness change analysis and grounding lines near the ice rumple L from April 2011 to February 2021. Background is from Landdsat-8 OLI optic image on 3 March 2019. (b) Time series of mean ice-equivalent freeboard thickness (2010–2022). Mean ice-equivalent freeboard thickness from ICESat-2 was calculated along tracks 965 and 1094 between 75.15°S and 75.05°S, representing the average of measurements from both the strong and weak ICESat-2 beams. REMA thickness values were sampled at the same locations as the ICESat-2 tracks. (c) Ice shelf bottom elevation profiles along ICESat-2 track 965 gt2r between February 2020 and May 2022. (d) Ice shelf bottom elevation profiles along ICESat-2 track 1094 gt3r between June 2019 and June 2022. Bed elevations are from the BedMachine v3 dataset (Morlighem et al., 2020; Morlighem, 2022), converted from EIGEN-6C4 to the EGM2008 geoid to match the vertical datum of REMA strips. The estimated vertical uncertainty is ±200 m (shown as a grey transparent box). The potential actual bed elevation is marked by a red dashed line.**

Figure 10:

• Please explain how you tracked the rift propagation history – I would like to know how you managed to put R1 and R2 arrows in Figures 10b,c,d, because there aren't any visible surface features in those locations from the images.

**Response 5: We manually identified the rift. We have revised the figure to highlight the rift more clearly.**

[Figure]

**Figure 10. Rift propagation history from 2013 to 2019. (a) Overview map showing the positions of panels (b) to (k). The background image is a Landsat-8 panchromatic image from 4 December 2013. (b)-(k) show the propagation history of the rifts R1, R2, and R3 (black or white arrow), which led to the 2020 calving event. The black circles indicate the positions of ice rumple L. Grounding lines are delineated based on the near-zero value of the double-differential vertical displacement.**

• In the figures, if an image does not have any rift formed, consider removing the labels. Also it should be 'not appeared', not 'no appeared'

**Response 6: Revised. See Response 5.**

• In Figures 10a, 10e and 10g, it's difficult to identify the rifts from these plots. Please make a high-resolution zoom-in map to show the rift patterns in detail

**Response 7: Revised. See Response 5.**

---

## Author Response (AR3)

**Response to the review of "Ephemeral grounding on the Pine Island Ice Shelf, West Antarctica, from 2014 to 2023," submitted to The Cryosphere.**

**Editor:**

We sincerely thank Editor for the review and feedbacks. In this document, we provide a point-by-point response to two comments. The editor's comments are shown in black, and our responses are shown in blue.

Dear authors,

Many thanks for updating your manuscript. It is now almost ready for publication, pending two changes:

- In the description of Figure 9: "tracks 965 and 1094 that WERE used..."

**Response 1:** Revised.

- In figure 7c and 2, the numbers of blue and red lines now match. However, they do not line up. For example, fig 7c shows 7 grounding events (3 blue, 4 red) in Dec 2020/Jan 2021, but in fig. 7d, they appear to occur in Nov 2020. Please check your code and ensure the grounding events are plotted on the right date in both figures.

**Response 2:** Thank you for pointing this out. We checked the code and found out that the axis for Figure 7c was correct and for Figure 7d was incorrect. Thus, we replotted them and corrected this error.

Thanks
Bert
Thanks.